# Targeting adipocyte ESRRA promotes osteogenesis and vascular formation in adipocyte-rich bone marrow

Tongling Huang [1], Zhaocheng Lu [1], Zihui Wang [1,2], Lixin Cheng [3], Lu Gao [1,2], Jun Gao [1], Ning Zhang [4], Chang-An Geng[5], Xiaoli Zhao [1], Huaiyu Wang [1], Chi-Wai Wong[6], Kelvin W. K. Yeung [7], Haobo Pan [1], William Weijia Lu[8] & Min Guan [1,2] ✉

Excessive bone marrow adipocytes (BMAds) accumulation often occurs under diverse pathophysiological conditions associated with bone deterioration. Estrogen-related receptor α (ESRRA) is a key regulator responding to metabolic stress. Here, we show that adipocyte-specific ESRRA deficiency preserves osteogenesis and vascular formation in adipocyte-rich bone marrow upon estrogen deficiency or obesity. Mechanistically, adipocyte ESRRA interferes with E2/ESR1 signaling resulting in transcriptional repression of secreted phosphoprotein 1 (*Spp1*); yet positively modulates leptin expression by binding to its promoter. ESRRA abrogation results in enhanced SPP1 and decreased leptin secretion from both visceral adipocytes and BMAds, concertedly dictating bone marrow stromal stem cell fate commitment and restoring type H vessel formation, constituting a feed-forward loop for bone formation. Pharmacological inhibition of ESRRA protects obese mice against bone loss and high marrow adiposity. Thus, our findings highlight a therapeutic approach via targeting adipocyte ESRRA to preserve bone formation especially in detrimental adipocyte-rich bone milieu.

Mammalian bone marrow consists of multiple cell types such as adipocytes, osteoblasts, osteoclasts, stromal cells and vascular cells. The interactions of these cells provide a critical regulatory milieu for the differentiation of bone marrow skeletal (also known as stromal or mesenchymal) stem cells (BMSCs) and other lineage cells, in turn, maintaining a complex homeostatic system for remodeling and regeneration in bone microenvironment[1,2]. It is well-accepted that bone marrow adipocytes (BMAds) originate from BMSCs which also give rise to osteoblasts and osteocytes[3]. Marrow adipose tissue

(MAT) accounts for approximately 10% of the total fat mass and about 70% of the whole bone marrow volume in healthy adult humans[4]. However, excessive BMAds accumulation often occurs in diverse clinical conditions including obesity, diabetes, anorexia nervosa, glucocorticoid treatment, radiotherapy, menopause and ageing, most of which are concomitant with bone deterioration[5]. MAT expansion may exaggerate the detrimental bone micro-environment and worsen disease progression. However, the precise mechanisms of excessive BMAds particularly in response to

[1]Research Center for Human Tissues and Organs Degeneration, Institute of Biomedicine and Biotechnology, Shenzhen Institute of Advanced Technology, Chinese Academy of Sciences, Shenzhen, China. [2]University of Chinese Academy of Sciences, Beijing, China. [3]Guangdong Provincial Clinical Research Center for Geriatrics, Shenzhen Clinical Research Center for Geriatrics, Shenzhen People's Hospital, Shenzhen, China. [4]Neuroscience Center, Shantou University Medical College, Shantou, China. [5]State Key Laboratory of Phytochemistry and Plant Resources in West China, Kunming Institute of Botany, Chinese Academy of Sciences, Kunming, China. [6]Guangzhou Huazhen Biosciences, Guangzhou, China. [7]Department of Orthopaedics and Traumatology, Li Ka Shing Faculty of Medicine, The University of Hong Kong, Hong Kong, China. [8]Faculty of Pharmaceutical Sciences, Shenzhen Institute of Advanced Technology, Chinese Academy of Sciences, Shenzhen, China. ✉e-mail: min.guan@siat.ac.cn

pathophysiological conditions in mediating bone-resident cellular communication remain elusive.

Adipose tissue is central to the regulation of whole-body energy homeostasis. Similar to white adipose tissue (WAT), normal MAT exhibits energy storage function and can release free fatty acid (FFA) to support exercise-stimulated bone formation[6]. However, MAT expands rapidly in response to caloric restriction indicating that MAT is distinct from peripheral adipose tissues. Moreover, a high-fat diet (HFD) also rapidly increases MAT expansion through circulating leptin acting on Leptin receptor expressing (LepR⁺) BMSCs to promote adipogenesis and inhibit osteogenesis of BMSCs[3,7]. Thus, it is plausible that aberrant signaling within bone microenvironment as a result of pathophysiological conditions drives BMSC lineage allocation toward committed adipogenic progenitors at the expense of osteoprogenitors, resulting in bone deterioration. Recent studies have described that accumulated BMAds contribute to circulating signal factors such as DPP4, RANKL and several well-known adipokines such as adiponectin and leptin in response to obesity, ageing or caloric restriction[2,8–11]. These secreted factors from either peripheral WAT or MAT might further impact on bone homeostasis.

BMAds are located in the bone milieu and in close contact with vascular and hematopoietic tissues. Osteogenesis is tightly coupled to bone angiogenesis; especially, a specific capillary endothelial cell (EC) subtype termed type H blood vessel in the metaphysis is linked to bone marrow vasculature at the metaphyseal-diaphyseal interface[12]. Senescence of BMAds triggered by glucocorticoid causes secondary senescence of surrounding vascular cells and osteoblasts leading to bone defect[13]. Abnormal type H vessels are also observed in aged individuals and postmenopausal women suffering from osteoporosis which is frequently accompanied by augments in marrow adiposity[14,15]. These evidences largely reflect that BMAds, osteoblasts and vessels are engaged in a reciprocal relationship maintaining bone microenvironment homeostasis. In fact, adipocyte-rich marrow is believed to impose a dominant negative effect on hematopoiesis after irradiation[16]. However, BMAds arising from adiponectin recombinase (Adipoq-Cre) labeled progenitors have recently been identified as the source of stem cell factor (SCF or KitL) which is required for normal haematopoiesis in young adult mice. Moreover, *Scf* transcripts are highly enriched in BMAds of human bone marrow from young donors, suggesting that BMAds play essential roles in normal marrow microenvironment[17]. Recent single cell RNA-seq based evidence further proposes that a subpopulation of Adipoq-labeled marrow adipocytes is critical for supporting marrow vasculature[18]. Thus, there are context dependent functions of BMAds as a niche component maintaining bone marrow microenvironment.

Orphan nuclear receptor estrogen-related receptor α (ESRRA; also known as ERRα or NR3B1) is an essential transcription factor governing energy balance and metabolism[19]. Specifically, ESRRA is required in stress-induced response to fasting, calorie restriction, cold exposure or overnutrition[20–22]. We previously found that liver-specific ESRRA knockout results in decreased levels of blood lipids due to blunted hepatic triglyceride-rich very low-density lipoprotein secretion in fasting mice[23]. ESRRA is also characterized as a key mediator involved in regulating diurnal metabolic rhythm-dependent intestinal lipid absorption, coupling cell metabolism and differentiation of kidney proximal tubule or controlling liver metabolism in response to insulin action and resistance[24–26]. We generated adipocyte-specific *Esrra* knockout mice by using the Adipoq-Cre which labels most BMAds and also peripheral adipose depots[2,11,17]. Here, we show that adipocyte ESRRA deficiency protects mice against bone loss and high marrow adiposity induced by either a HFD or ovariectomy surgery (OVX) with little effect on peripheral WAT. Using in vitro and in vivo approaches, we establish that adipocyte ESRRA inhibition promotes bone formation and reforms bone vasculature by oppositely regulating the expression and secretion of

leptin and secreted phosphoprotein 1 (SPP1, also known as osteopontin).

## Results

### Conditional adipocyte deletion of ESRRA enhances bone formation and counteracts marrow fat accumulation in DIO osteopenia mice

We utilized a conditional knockout strategy by intercrossing *Esrra*^fl/fl mice with Adipoq-Cre to generate adipocyte-specific *Esrra* knockout mice (*Esrra*^fl/fl; *AdipoqCre*, denoted *Esrra*^AKO) (Supplementary Fig. 1a, b). *Esrra*^AKO mice at the age of 10 weeks developed normally with no significant differences in growth, body weight, WAT phenotype and rectal temperature compared to *Esrra*^fl/fl littermates (Supplementary Fig. 1c–f). MicroCT analysis of distal femurs of *Esrra*^AKO male mice showed no changes in trabecular bone volume per total volume (BV/TV), trabecular bone number (Tb.N) and trabecular bone thickness (Tb.Th) compared to control littermates; in addition, no changes were detected in plasma concentrations of procollagen type 1 N-terminal propeptide (P1NP, a bone formation marker) (Supplementary Fig. 1g–i). These data suggest that *Esrra* deletion in adipocytes did not affect the phenotypes of WAT and bone in adult mice under normal physiological condition.

Obesity is believed to induce excessive adipocyte accumulation not only in peripheral WAT but also in bone marrow cavities in rodents and humans, contributing to detrimental disturbances in bone remodeling or bone regeneration[2,27,28]. Since ESRRA is responsive to stress-induced metabolic challenges including overfeeding[21,26], we queried whether adipocyte-specific ablation of ESRRA is resistant to diet-induced obesity (DIO) or involved in fat-bone axis. Compared to *Esrra*^fl/fl NCD (normal chow diet) mice, *Esrra*^fl/fl DIO mice exhibited obese metabolic phenotypes including increased body weight, WAT mass and plasma levels of TG and FFA after a long-term HFD feeding for 16 weeks (Fig. 1a–d). Moreover, long-term HFD resulted in a significant increase in the circulating levels of inflammatory factors TNFa and IL6 in *Esrra*^fl/fl DIO mice, as well as dramatically augmented leptin level which is proportional to fat stores and considered a mature adipocyte marker (Fig. 1e). Compared to *Esrra*^fl/fl DIO mice, *Esrra*^AKO DIO mice are similar in these metabolic cues and food intake, as well as glucose disposal rates upon an oral glucose tolerance test (Fig. 1b–d and Supplementary Fig. 2a–c). However, circulating lepin level was significantly reduced in *Esrra*^AKO DIO mice while the levels of TNFa and IL6 remained unchanged compared to *Esrra*^fl/fl DIO mice (Fig. 1e). These evidences indicated that adipocyte ESRRA deficiency modulates leptin level without disturbing general metabolism in the context of HFD-induced obesity.

Notably, the trabecular bone mass in *Esrra*^AKO DIO mice was greater than that of *Esrra*^fl/fl DIO mice which displayed osteopenia in comparison with NCD controls (Fig. 1f, g). *Esrra*^AKO DIO mice had significant increases in BV/TV, Tb.N and Tb.Th; whereas Tb.Sp was reduced (Fig. 1g). In addition, *Esrra*^AKO DIO mice had more osteoblasts (Ob.N/BS and Ob.S/BS), indicating an enhancement of bone formation confirmed by significant increases in trabecular mineral apposition rate (MAR) and bone formation rate (BFR/BS) using calcein double labeling (Fig. 1h, i). However, bone histomorphometry analysis of femoral metaphyses displayed no difference in bone resorption between the two DIO groups revealed by TRAP staining, quantification of osteoclast number (Oc.N/BS) and proportion of bone surface with active osteoclasts (Oc.S/BS) (Fig. 1j). We detected no changes in cortical bone including the midshaft cortical BV/TV and cortical bone thickness (Ct. Th) (Supplementary Fig. 2d, e). Consistently, plasma concentrations of P1NP were apparently elevated in *Esrra*^AKO DIO mice but collagen type I c-telopeptide (CTX-1, a bone resorption marker) remained unchanged, suggestive of increased net bone formation in *Esrra*^AKO DIO mice (Fig. 1k, l). As expected, HFD promoted MAT expansion substantially in *Esrra*^fl/fl DIO mice in comparison with NCD mice (Fig. 1m, n). However, histological examination in femur sections

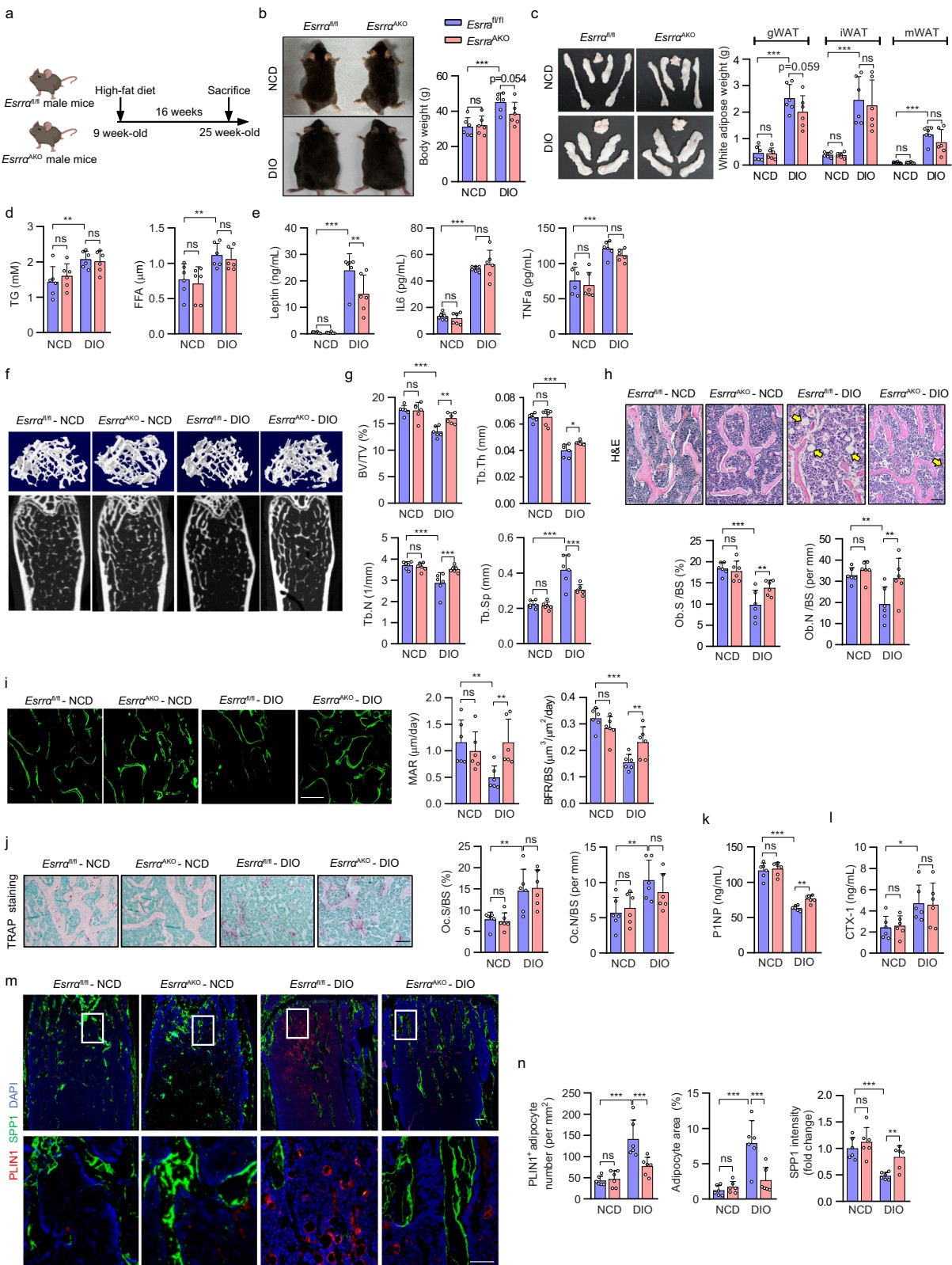

by immunofluorescence staining of perilipin 1 (PLIN1), an adipocyte marker, and SPP1, a bone matrix protein marker, revealed that loss of ESRRA dramatically restrained HFD-induce MAT expansion while promoting bone matrix formation (Fig. 1m, n). Collectively, these data demonstrated that adipocyte ESRRA ablation counteracts DIO-associated bone loss and high marrow adiposity by favoring bone formation.

## Loss of *Esrrα* in adipocytes attenuates bone loss through favoring bone formation and inhibiting marrow adiposity in OVX-induced osteoporosis

There is a high coincidence rate of bone loss and excessive fat deposition including MAT expansion in postmenopausal women[29,30]. Similarly, estrogen deprivation also leads to fat-bone imbalance in rodents[31]. To investigate whether lack of *Esrrα* in adipocytes affects

**Fig. 1 | Adipocyte-specific ESRRA ablation augments bone formation and inhibits MAT expansion in DIO osteopenia mice. a** Schematic diagram illustrating the experimental procedure for diet-induced obesity (DIO) mice model. 9-week-old *Esrra*^fl/fl and *Esrra*^AKO male mice were fed a normal chow diet (NCD) or high-fat diet (HFD) for 16 weeks. (Schematic created with BioRender.com. Agreement number: WP26KB8FER). **b** Representative pictures and body weights of NCD and DIO mice. **c** Representative images and weight analysis of white adipose tissue (WAT) depots, including gonadal WAT (gWAT), inguinal WAT (iWAT), and mesentery WAT (mWAT). **d** Plasma triglyceride (TG) and free fatty acid (FFA) levels. **e** Plasma leptin, IL-6 and TNFa levels. **f** Representative micro-CT images of the distal femoral trabecular bone. **g** Quantitative analysis of bone volume/tissue volume ratio (BV/TV), trabecular thickness (Tb. Th), trabecular number (Tb. N) and trabecular separation (Tb. Sp). **h** H&E staining of femur sections (scale bar: 100 μm). Yellow arrows indicate the bone marrow adipocytes. Osteoblast surface to bone surface ratio (Ob.S/BS) and osteoblast number to bone surface ratio (Ob.N/BS) are shown on the right panel. **i** Calcein double labeling of trabecular bone (scale bar: 50 μm). Mineral apposition rate (MAR) and bone formation rate (BFR/BS) were determined as graphs. **j** TRAP staining of femur sections with quantitative analysis of Oc.S/BS and Oc.N/BS. TRAP-positive purple spots indicate multinucleated osteoclasts (scale bar: 100 μm). Plasma P1NP (**k**) and CTX1 (**l**) levels. **m** PLIN1 positive marrow adipocytes (PLIN1⁺, red) and SPP1 (green) immunofluorescence staining in femur sections (scale bar: 100 μm). The box in the upper showing the metaphysis region near growth plate is represented at higher magnification in the bottom (scale bar: 50 μm). **n** The number and area of adipocytes in the femur marrow per tissue area and the quantification of SPP1 fluorescence intensity were measured from femur sections in (**m**). Data are shown as mean ± SD ($n = 6$ mice per group). \*$p < 0.05$, \*\*$p < 0.01$ and \*\*\*$p < 0.001$. Statistical analysis is performed using two-way ANOVA with Fisher's LSD post hoc analysis. Source data are provided as a Source Data file.

bone homeostasis under hormonal deficiency, ovariectomized (OVX) female mice were utilized to mimic estrogen deficiency-related osteoporosis. As expected, 18-wk-old female mice at 8 weeks after OVX gained body weight with excessive WAT deposition in *Esrra*^fl/fl controls (Fig. 2a–c and Supplementary Fig. 3a). However, there are no differences in body weight, WAT phenotype, blood lipids and glucose between the two genotypes in sham or OVX groups (Fig. 2b, c and Supplementary Fig. 3a–d). These evidences implied that adipocyte ESRRA abrogation did not alter WAT development upon estrogen deprivation. Similar to *Esrra*^AKO DIO mice, the increase in circulating leptin induced by OVX was significantly reduced in *Esrra*^AKO OVX mice despite no differences in peripheral WAT mass, suggesting a regulatory role of ESRRA on leptin expression (Fig. 2d).

MicroCT analysis of the femur trabecular bone showed that the disruption in bone remodeling induced by OVX measured by BV/TV, Tb.N, Tb.Sp and Tb.Th was significantly ameliorated in *Esrra*^AKO mice in comparison to *Esrra*^fl/fl controls (Fig. 2e, f). No differences in bone resorption and cortical bone parameters were found between *Esrra*^AKO OVX mice and the corresponding controls (Fig. 2g and Supplementary Fig. 3e, f). However, osteoblast number and osteoblast surface were noticeably elevated in *Esrra*^AKO OVX mice due to accelerated bone formation (Fig. 2h). Consistently, blood levels of P1NP and CTX-1 further confirmed the increased net bone formation in *Esrra*^AKO mice following OVX (Fig. 2i, j) evidenced by increases in MAR and BFR (Fig. 2k). As expected, OVX significantly increased MAT accumulation accompanied with bone loss *Esrra*^fl/fl mice (Fig. 2h, l). Despite unchanged extramedullary WAT phenotype in *Esrra*^AKO OVX mice, MAT expansion induced by estrogen deprivation was substantially eliminated in mutant OVX mice (Fig. 2b, h, l). *Esrra*^AKO OVX mice displayed marked declines in the number and size of marrow adipocytes while having more osteoprogenitors around trabecular bone (Fig. 2h, l, m). These evidences reveal that adipocyte-specific ESRRA deficiency has a major impact on facilitating bone formation and inhibiting marrow adipogenesis under estrogen deficiency.

## Adipocyte-specific ESRRA ablation leads to increased SPP1 expression

The fact that lack of ESRRA in adipocytes apparently reduces marrow adiposity in DIO and OVX mice drew our attention to BMAds. Given that BMAds originate from BMSCs present in the bone marrow, ESRRA expression was confirmed to be abrogated in *Esrra*^AKO BMAds lineage cells at indicated days after adipogenic induction, but not during BMSCs osteogenesis. (Fig. 3a and Supplementary Fig. 4a). Then, isolated BMSCs from *Esrra*^fl/fl and *Esrra*^AKO mice were differentiated for 14 days in adipogenic or osteogenic induction medium, but no significant differences were observed (Fig. 3b, c). These data suggest that Adipoq-Cre-driven ablation of *Esrra* did not directly change the differentiation capability of BMSCs under normal physiological condition which is consistent with in vivo evidences. Treatment of rosiglitazone (Rosi) or

pioglitazone can dramatically induce the expression of peroxisome proliferator activated receptor gamma (PPARG) target genes and exaggerate MAT accumulation in rodents or humans[13,18,32]. Therefore, we induced BMSCs differentiation by Rosi to mimic adipocyte expansion under pathological conditions and observed a reduction in adipogenesis of BMSCs from *Esrra*^AKO mice (Fig. 3c). These evidences indicated that loss of ESRRA during adipocyte expansion under pathological conditions might exert a secondary or paracrine effect on heterogeneous BMSCs differentiation.

To explore the transcriptional role of ESRRA in regulating BMAds autonomous genes, bulk RNA-seq transcriptome analysis was performed in BMAds lineage cells at 7 days upon adipogenic induction (Fig. 3b). A total of 4217 genes were quantified, of which 1977 and 1448 were respectively up- and down-regulated in the ESRRA knockdown cells (Fig. 3d). As expected, the expression profile of genes associated with energy metabolism, including oxidative phosphorylation, tricarboxylic acid cycle, lipid metabolism and glycolysis were modestly affected (Supplementary Fig. 4b–e). By using Gene Ontology (GO) and Kyoto Encyclopedia of Genes and Genomes (KEGG) analysis, differentially expressed genes were found to be enriched in pathways related to extracellular environment and cellular response, indicating the involvement in cell signaling and interactions (Fig. 3e, f). Given that multiple secreted factors from BMAds are well-known functional regulators[33], priority was given to 17 differentially expressed secreted factors (Fig. 3g). Notably, SPP1 was most robustly upregulated among the top-ranked secreted factors by ESRRA ablation (Fig. 3g, h and Supplementary Fig. 4f, g). SPP1 is a well-known matrix protein by binding to multiple integrins through a highly conserved arginine-glycine-aspartic acid (RGD) motif[34] and involved in multiple pathways including the extracellular region, integrin binding, ECM-receptor interaction, and focal adhesion (Fig. 3e, f). Furthermore, qRT-PCR profiling and western blot analysis confirmed that SPP1 expression was indeed enhanced dramatically by ESRRA abrogation in fully differentiated BMAds (Fig. 3i, j). In contrast, the expression levels of leptin and other adipogenic marker genes such as *Pparg*, *Cebpa* and *Fabp4* were down-regulated (Fig. 3i, j and Supplementary Fig. 4h). Taken together, these ex vivo data reveal that ESRRA ablation in BMAds led to altered expression of a panel of secreted factors, especially a significant augment of SPP1 expression.

## Adipocyte ESRRA negatively regulates the transcription of *Spp1* by interfering with E2/ESR1 signaling

We next sought to dissect the mechanism by which ESRRA negatively regulates SPP1 expression in adipocytes. Compared to sham, circulating SPP1 was significantly decreased in OVX mice but was partially rescued in *Esrra*^AKO OVX mice (Fig. 4a). This ESRRA-mediated SPP1 production and secretion may not only arise from BMAds but also derive from peripheral WAT. We detected local SPP1 expression in gWAT with leptin as a WAT-adipocyte marker by using

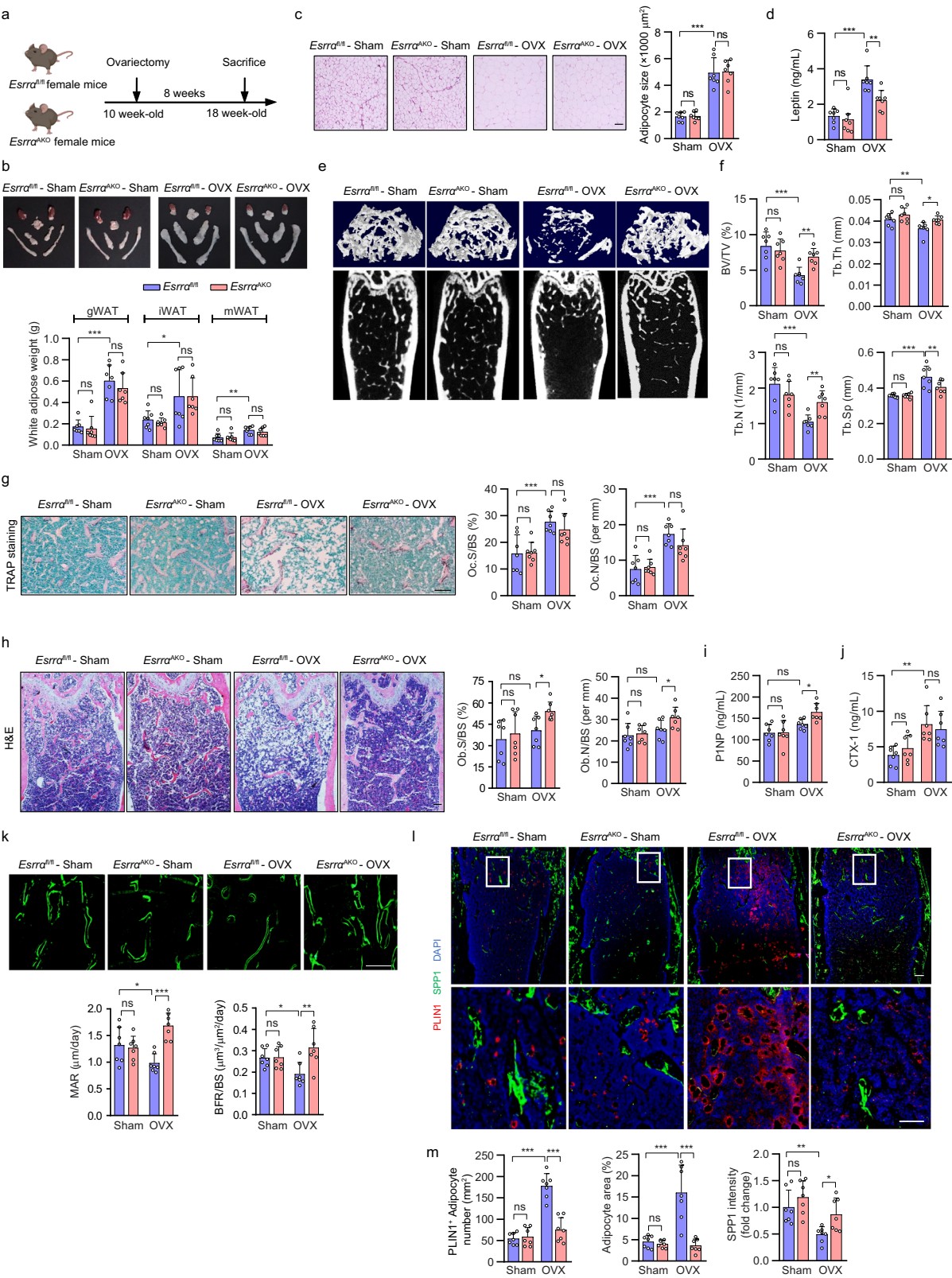

immunofluorescence double staining. The results showed that ESRRA knockdown increased SPP1 expression in gWAT adipocytes from OVX mice, concomitant with decreased leptin expression confirmed by qRT-PCR and western blot analysis (Fig. 4b–d). No changes were observed in other adipocyte-related genes including *Adipoq*, *Pparg*, *Cebpa* and *Fabp4* (Fig. 4c). Furthermore, SPP1 expression were remarkably reduced in gWAT adipocytes of female mice 4 and 8 weeks

after OVX, suggesting estrogen might be essential for adipocyte SPP1 expression (Fig. 4d and Supplementary Fig. 5a, b). These results clearly revealed that in addition to BMAds, WAT-adipocyte is also a source of SPP1. Importantly, a reduction in adipocyte SPP1 expression due to estrogen deficiency can be reversed by ESRRA ablation.

Although ESRRA does not bind 17β-estradiol (E2), ESRRA can modulate estrogen receptor alpha (ESR1)-mediated response of shared

**Fig. 2 | ESRRA deficiency in adipocytes favors bone formation and attenuates marrow adiposity in OVX-induced osteoporosis. a** Schematic diagram illustrating the experimental procedure for ovariectomy (OVX)-induced osteoporosis mice model. 10-week-old *Esrra*^fl/fl^ and *Esrra*^AKO^ female mice underwent either sham or OVX operation for 8 weeks (schematic created with BioRender.com. Agreement number: Il26KB8K2T). **b** Representative images and weights of adipose depots. **c** Representative images and adipocytes size analysis from H&E-stained gWAT sections (scale bar: 50 μm). **d** Plasma leptin levels. Micro-CT images of distal femurs in sham and OVX mice (**e**) with morphometric analysis of BV/TV, Tb.N, Tb.Th, and Tb.Sp (**f**). **g** Representative TRAP-stained images and quantification of Oc.S/BS and Oc.N/BS in distal femoral metaphysis regions from sham and OVX mice (scale bar: 100 μm). **h** Representative H&E-stained images and quantification of Ob.S/BS and Ob.N/BS (scale bar: 100 μm). Plasma P1NP (**i**) and CTX1 (**j**) levels. **k** Calcein double labeling with quantitative analysis of MAR and BFR/BS (scale bar: 50 μm). **l, m** Immunofluorescence co-staining and quantification of PLIN1^+^ bone marrow adipocytes (red) and SPP1 (green) of femur sections from sham and OVX mice. Scale bar: upper panel, 100 μm; lower panel, 50 μm. Data are shown as mean ± SD ($n = 7$ mice per group). *$p < 0.05$, **$p < 0.01$ and ***$p < 0.001$. Statistical analysis is performed using two-way ANOVA with Fisher's LSD post hoc analysis. Source data are provided as a Source Data file.

target genes in an E2 dependent way[35]. *Spp1* promoter does not contain any canonical ESR1 response element (ERE) with an invert repeat of AGGTCA half-sites. Nonetheless, E2/ESR1 can regulate a single half-site preceded by a trinucleotide sequence TCA, which is also an ESRRA response element (ERRE)-like sequence[36]. Three such sites (S1–S3) can be found on the *Spp1* promoter (Fig. 4e). We found that ESR1 induced *Spp1* promoter activity, which was further enhanced by E2, in 3T3-L1 preadipocytes upon adipogenic differentiation for 2 days (Fig. 4f and Supplementary Fig. 5c). In contrast, ESRRA overexpression suppressed the E2/ESR1-driven *Spp1* promoter activation in a dose-dependent manner, demonstrating a role of ESRRA in attenuating the E2/ESR1-dependent transcriptional activation (Fig. 4f). However, this attenuation was diminished by using a truncated ESRRA protein without its DNA-binding domain (DBD), confirming the repressive effect of ESRRA on *Spp1* transcriptional activity is through binding to its promoter (Fig. 4g). To further evaluate this interference, we next determined whether ESRRA knockdown would result in an induction of E2/ESR1 transactivation on the *Spp1* promoter. By chromatin immunoprecipitation (ChIP) assay, ESRRA abrogation led to enhanced recruitment of ESR1 to the *Spp1* promoter, especially region 2 which was more responsive to E2 stimulation (Fig. 4h). To validate whether region 2 is responsible for the transcriptional regulation, we constructed a truncated *Spp1* promoter without region 2 and found that it was no longer responsive to E2/ESR1 transactivation and ESRRA-mediated repression effect (Fig. 4i). Moreover, the directly interruption effect of ESRRA on E2/ESR1 modulation was further confirmed by ChIP assay, evidenced by increased occupancy of ESRRA to the *Spp1* promoter upon ESRRA overexpression while decreased enrichment by ESR1 or E2 treatment (Fig. 4j). Consistently, mRNA expression of *Spp1* was induced by E2 but suppressed by an adenovirus expressing ESRRA in fully differentiated BMAds (Fig. 4k and Supplementary Fig. 5d). Similar evidences were recaptured in 3T3-L1 mature adipocytes as well as human BMSCs-derived BMAds (Fig. 4l, m and Supplementary Fig. 5e, f). Collectively, our data demonstrated that ESRRA negatively modulates SPP1 expression through interfering with E2/ESR1 signaling in adipocytes (Fig. 4n).

## Adipocyte ESRRA ablation results in an increase in SPP1 secretion facilitating type H vessel formation in osteoporotic mice

SPP1 is well known to exert a proangiogenic effect mediating neovascularization in the early stage of bone healing[37,38]. Notably, global knockout of SPP1 causes a reduction in type H vessels in mice long bone[39]. Therefore, it is possible that soluble form of SPP1 released from adipocytes might contribute to type H vessel formation. We performed CD31 and Endomucin (EMCN) double immunofluorescence staining in the metaphysis of distal femur to verify type H blood vessel formation. Consistent with previous studies[14,40], OVX caused a significant reduction of type H blood vessels and adjacent Osterix^+^ osteoprogenitor cells in *Esrra*^fl/fl^ mice compared to sham (Fig. 5a–d). In contrast, the abundance of type H vessels and osteoprogenitors were markedly preserved in *Esrra*^AKO^ mice following OVX (Fig. 5a–d). As a characteristic feature of metaphyseal type H vessels, the organization

and length of columnar vessels exhibited pronounced improvement in mutant mice (Fig. 5a, b). Similarly, *Esrra*^fl/fl^ DIO mice exhibited apparently distorted type H vessels; however, this distortion was minimal in *Esrra*^AKO^ DIO mice associated with an increase of osteoprogenitor cells (Supplementary Fig. 6a–d). These results suggest that adipocyte ESRRA ablation rescues deteriorated type H vessels under pathological osteoporotic conditions while promoting perivascular osteoprogenitors accretion.

As a negatively charged non-collagenous matrix glycoprotein, soluble SPP1 has a high affinity for calcium allowing it to anchor in bone environment[41]. To determine whether BMAd- or WAT-derived soluble SPP1 contributed to the formation of type H blood vessels, we collected conditioned medium (CM) from both sources and performed co-culture assays to evaluate the efficacy of secreted SPP1 in inducting angiogenesis of microvascular endothelial cells (ECs) (Fig. 5e). We first detected the concentrations of SPP1 from CM and confirmed that adipocyte ESRRA abrogation enhanced SPP1 secretion (Fig. 5f, g). Transwell migration assay revealed that ECs cultured in either BMAds CM or gWAT CM from *Esrra*^AKO^ mice displayed an enhanced ability to migrate relative to *Esrra*^fl/fl^ controls (Fig. 5h, i). Additionally, Matrigel tube formation assay confirmed that CM from both sources of mutant mice promoted the formation of capillary-like network structures (Fig. 5j–m). Importantly, recombinant SPP1 (rSPP1) rescued ECs migration and tube formation in two *Esrra*^fl/fl^ CM groups (Fig. 5h–m). On the contrary, addition of anti-SPP1 neutralizing antibody (SPP1 NAb) in two *Esrra*^AKO^ CM groups hampered the angiogenic processes (Fig. 5h–m). These data indicated that soluble SPP1 released by both sources of ESRRA-deficient adipocytes was responsible for ECs migration and vascular formation. Taken together, our findings illustrated a potent protective effect of adipocyte ESRRA ablation on bone angiogenesis in OVX and DIO mice through elevated soluble SPP1 production. It is thus plausible that enhanced bone formation in *Esrra*^AKO^ mice is in part facilitated by the sustained bone marrow vascularization, especially under osteoporotic condition induced by estrogen derivation or overfeeding.

## Adipocyte ESRRA positively regulates transcriptional expression of *Leptin* by binding to the *Leptin* promoter

Compared to DIO or OVX controls, ESRRA deficient mice displayed declined levels of circulating leptin despite no differences in peripheral WAT mass, as well as repressed expression in both WAT- adipocytes and BMAds (Figs. 1e, 2d, 3j, 4b–d and Supplementary Fig. 7), suggesting that *Lep* may be a target gene of ESRRA. Thus, we evaluated whether ESRRA may directly regulate the expression of *Lep* by binding to four putative ERREs S1–S4 on the promoter (Fig. 6a). We first established that this wild-type *Lep* promoter is sufficient to confer, in a dose-dependent manner, responsiveness to ESRRA and its co-activator PPARGC1A (Fig. 6b). Moreover, the regulation on *Lep* promoter was dose-dependently suppressed by a well-studied ESRRA specific antagonist Compound 29 (C29) (Fig. 6c). Importantly, deletion and mutation analysis of these putative ERREs revealed that S1 and S3 ERREs were primarily responsible for the ESRRA-driven *Lep* expression (Fig. 6d). We next investigated if ESRRA is physically bound to these

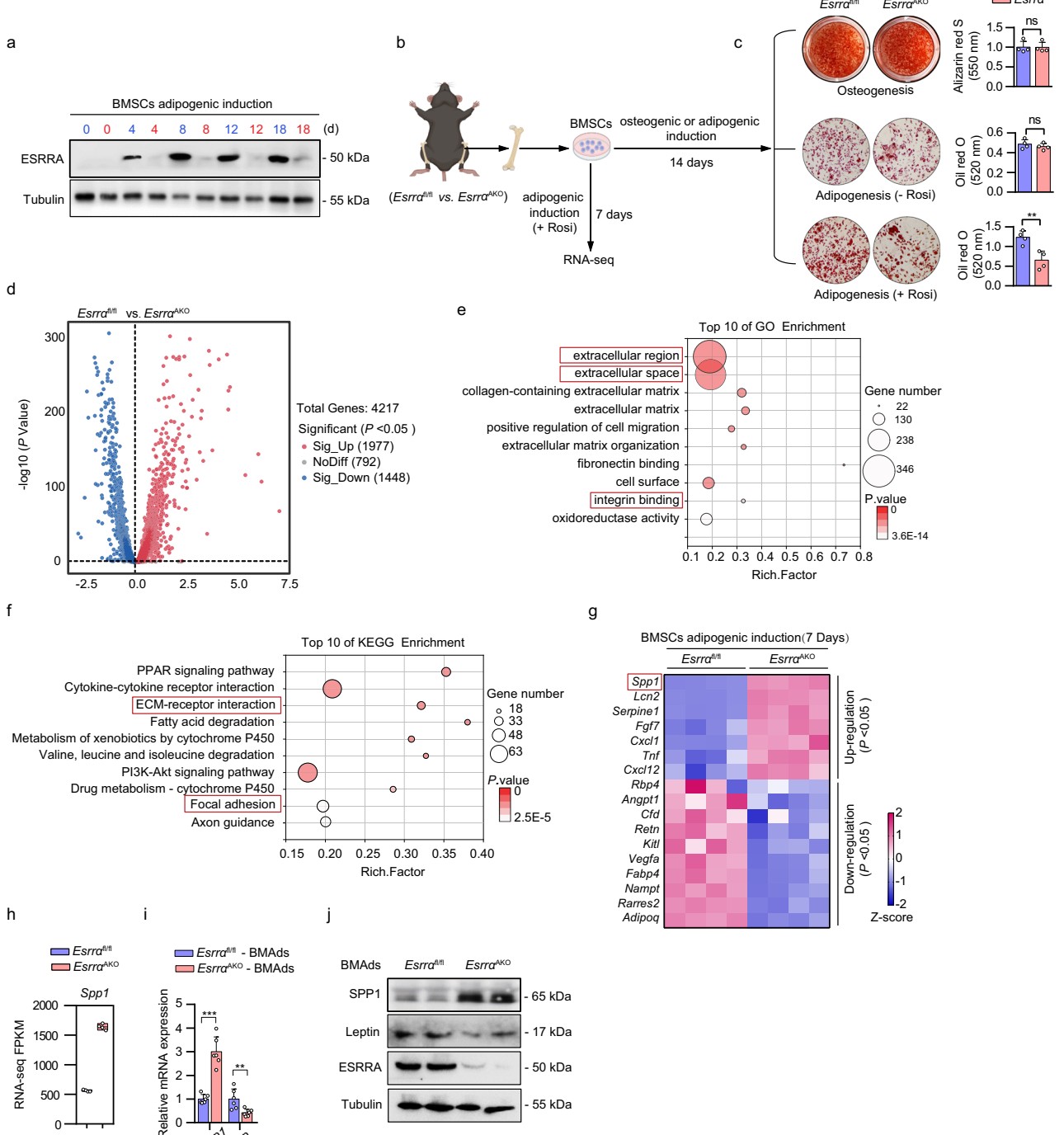

**Fig. 3 | Loss of ESRRA results in enhanced SPP1 expression in adipocytes.**
**a** Protein expression levels of ESRRA were evaluated in BMSCs upon adipogenic induction for indicated days, comparing *Esrra*[fl/fl] mice (blue font) with *Esrra*[AKO] mice (red font). **b** Schematic representation of the experimental design. BMSCs were isolated from *Esrra*[fl/fl] and *Esrra*[AKO] mice and subsequently subjected to either adipogenic or osteogenic induction for indicated days (schematic created with BioRender.com. Agreement number: LW26KBAHRA). **c** Representative images and quantitative analyses of alizarin red S staining and oil red O staining following the indicated induction. *n* = 4 biologically independent experiments. Rosi, rosiglitazone. **d** Volcano plot of transcriptional profiling between BMSCs-derived BMAds lineage cells from *Esrra*[fl/fl] and *Esrra*[AKO] mice. Differentially expressed genes were identified using DESeq2 analysis (*p* < 0.05). *n* = 4 biologically independent samples.

Gene Ontology (GO) (**e**) and Kyoto Encyclopedia of Genes and Genomes (KEGG) (**f**) pathway enrichment analyses of all differentially expressed genes by RNA-seq (top 10 according to adjusted *p* value). **g** Heatmap depicting selected genes related to secreted factors (*p* < 0.05). *n* = 4 biologically independent samples. **h** Boxplot showing the transcript expression value (FPKM) of *Spp1* based on RNA-seq data. Data are represented as box and whiskers with bars representing maximum and minimum values and with median highlighted as a line. *n* = 4 biologically independent samples. Validation of SPP1 and leptin expression were performed by qRT-PCR (**i**) and western blotting analysis (**j**) in BMAds that were fully differentiated for 14 days. *n* = 6 biologically independent samples. All the data are shown as mean ± SD. **p* < 0.01 and ***p* < 0.001. Statistical analysis is performed using unpaired two-tailed Student's *t* test. Source data are provided as a Source Data file.

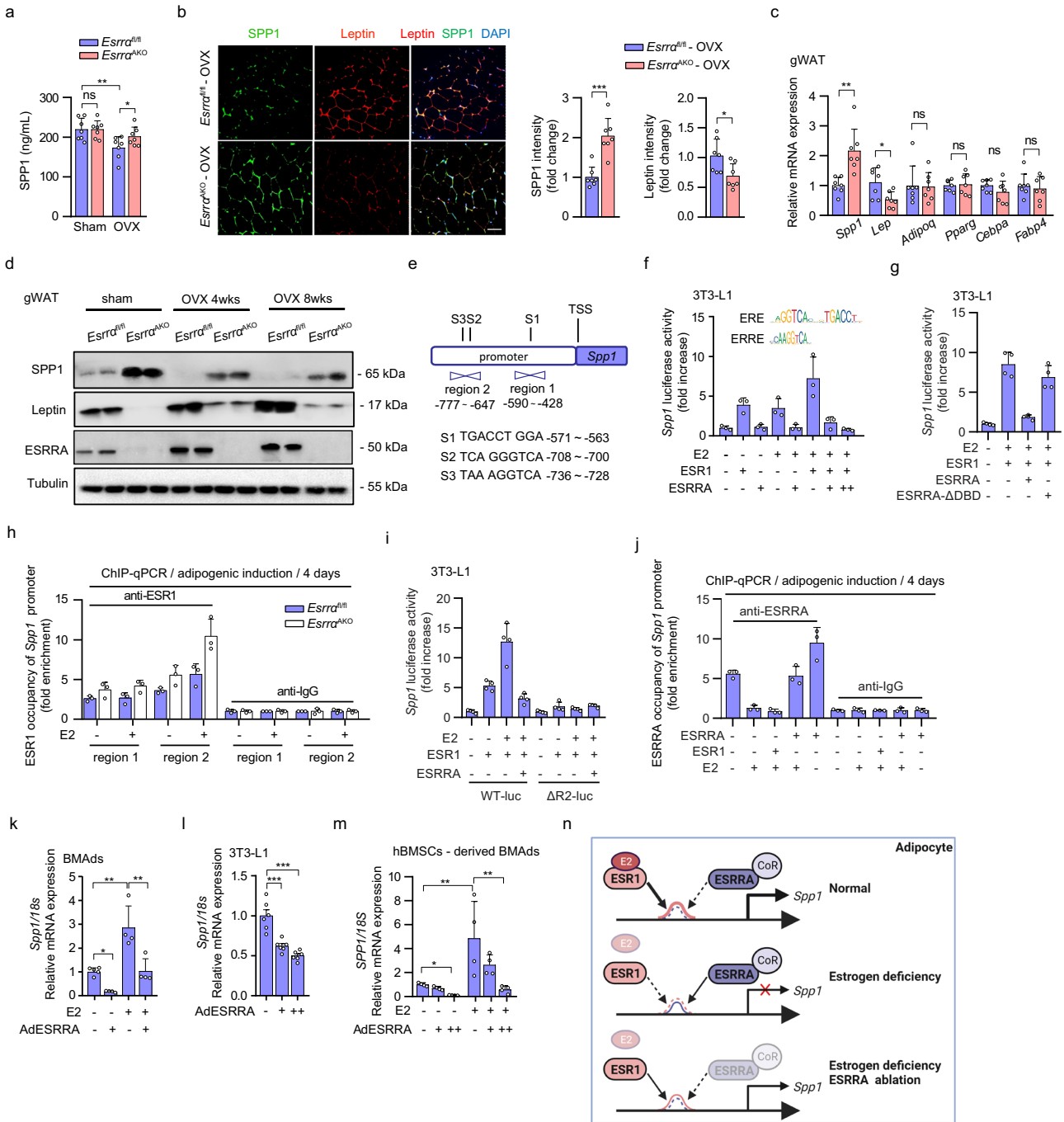

**Fig. 4 | ESRRA represses SPP1 transcriptional expression via interrupting with E2/ESR1 signaling in adipocytes. a** Plasma SPP1 levels of *Esrra*^fl/fl and *Esrra*^AKO mice at 8 weeks post-OVX or sham operation. *n* = 7 mice per group. **b** Representative images and analysis of SPP1 and leptin co-staining of gWAT from OVX mice studied in (**a**) (scale bar: 100 μm). *n* = 7 mice per group. **c** mRNA expression of *Spp1*, *Leptin*, *Adipoq*, *Pparg*, *Cebpa* and *Fabp4* of gWAT from OVX mice. *n* = 7 mice per group. **d** Protein levels of SPP1, leptin and ESRRA of gWAT from *Esrra*^fl/fl and *Esrra*^AKO mice at 4 and 8 weeks post-OVX or sham operation. **e** Schematic diagram displays the potential binding sites of ESR1 within the *Spp1* promoter, including S1, S2 and S3. Fragments for ChIP assay shown as region 1 (R1) and region 2 (R2). **f** Luciferase reporter activities of the *Spp1* promoter in adipogenesis induced 3T3-L1 cells transfected with *Esrra* or *Esr1* expressing plasmids in the presence of E2 or not. *n* = 3 biologically independent experiments. The consensus sequence binding motifs for ESR1 response element (ERE) and ESRRA response element (ERRE) are presented. **g** Luciferase reporter activities of the *Spp1* promoter regulated by E2/ESR1 in the presence of wild-type (WT) or DNA-binding domain-deleted ESRRA construct (ESRRA-ΔDBD). *n* = 4 biologically independent experiments. **h** ChIP assay with ESR1

antibody in BMSCs from *Esrra*^fl/fl and *Esrra*^AKO mice after adipogenic induction for 4 days along with or without E2. *n* = 3 biologically independent experiments. **i** Luciferase reporter activities of the R2 deleted-*Spp1* promoter (ΔR2-luc) as compared to *Spp1* promoter (WT-luc). *n* = 4 biologically independent experiments. **j** Enrichment of ESRRA in R2 of *Spp1* promoter in adipogenesis induced 3T3-L1 cells with the indicated treatments. *n* = 3 biologically independent experiments. *Spp1* mRNA in murine BMAds (**k**), matured 3T3-L1 adipocytes (**l**) or human BMSCs-derived BMAds (**m**) infected with adenovirus expressing ESRRA or GFP with E2 treatment for 2 days. *n* = 4, 6, 4 biologically independent experiments, respectively. **n** Diagram illustrating the mechanism of ESRRA-regulated repression of *Spp1* transcriptional expression via interfering with E2/ESR1 signaling in adipocytes (schematic created with BioRender.com. Agreement number: BH26KF823M). Data are shown as mean ± SD. **p* < 0.05, ***p* < 0.01 and ****p* < 0.001. Statistical analysis is performed using unpaired two-tailed Student's *t* test (**c**), one-way ANOVA followed by Bonferroni's post hoc tests (**k**, **l**, **m**). Source data are provided as a Source Data file.

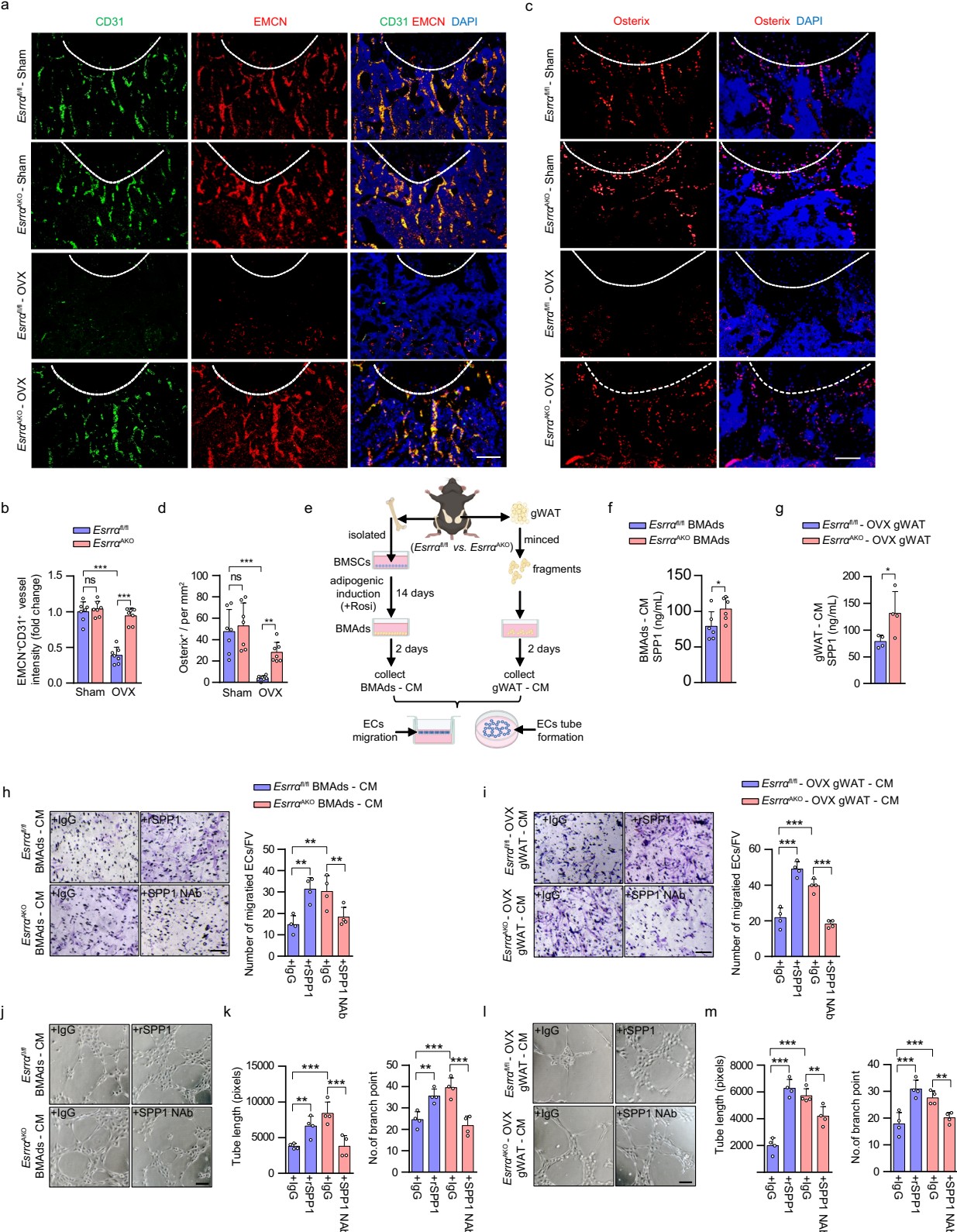

ERREs by ChIP assays using primers encompassing regions 1–3 (Fig. 6a). 3T3-L1 underwent adipogenic differentiation for 4 days to induce endogenous ESRRA and leptin expression. ChIP-qPCR analysis confirmed the binding of endogenous ESRRA to regions 1 and 2 that contains S1 and S3 respectively; however, C29 reduced said ESRRA occupancy (Fig. 6e). Furthermore, augmented ESRRA occupancy was

measured in regions 1 and 2 in 3T3-L1 overexpressing ESRRA (Fig. 6f). In line with these observations, leptin expression was increased in a dose-dependent manner by an adenovirus overexpressing ESRRA in matured 3T3-L1 adipocyte as well as in BMAds differentiated from murine or human BMSCs (Fig. 6g–i). In summary, our data strongly established that *Lep* is a bona fide ESRRA target gene in adipocytes.

**Fig. 5 | Adipocyte ESRRA deficiency rescues type H vessel formation under osteoporotic condition by facilitating SPP1-induced angiogenesis.**
**a** Representative images of metaphyseal type H vessels near growth plate immunostained for Endomucin (EMCN, red) and CD31 (green) in distal femurs of *Esrra*fl/fl and *Esrra*AKO mice following sham and OVX. DAPI (blue) is used for counterstaining of nuclei (scale bar: 100 μm). **b** Quantification of CD31+ EMCN+ type H vessel intensity per mm². *n* = 7 mice per group. **c** Immunostaining of Osterix (red) with DAPI (blue) in the metaphysis of distal femurs of *Esrra*fl/fl and *Esrra*AKO mice following sham and OVX (scale bar: 50 μm). **d** Quantification of Osterix+ cells in bone marrow per mm². *n* = 7 mice per group. **e** Schematic diagram showing the procedure of the conditioned medium (CM) preparation, tube formation assay and cell migration assay (schematic created with BioRender.com. Agreement number: VY26KB8O73). The concentrations of soluble SPP1 in BMAds-CM (**f**) or gWAT-CM (**g**) prepared from *Esrra*fl/fl and *Esrra*AKO OVX mice were measured by ELISA. *n* = 6, 4 biologically

independent samples, respectively. Microvascular endothelial cells (ECs) migration in response to BMAds-CM (**h**) or gWAT-CM (**i**) with the addition of 0.5 μg/ml recombinant SPP1 (rSPP1), 1 μg/ml neutralizing SPP1 antibody (SPP1 Nab), or an equal volume of IgG for 24 h was followed by quantification and presentation of representative images from four independent experiments (scale bars: 100 μm). *n* = 4 biologically independent experiments. **j–m** Matrigel tube formation assay was performed using ECs and BMAds-CM (**j**) or gWAT-CM (**l**) with the indicated treatments for 4 h (scale bar: 100 μm). The length of the tubes and the number of branch points (**k**, **m**) per field were analyzed. *n* = 4 biologically independent experiments. Data are shown as mean ± SD. *p < 0.05, **p < 0.01 and ***p < 0.001. Statistical analysis is performed using unpaired two-tailed Student's *t* test (**f**, **g**) and two-way ANOVA with Fisher's LSD post hoc analysis (**b**, **d**, **h**, **i**, **k**, **m**). Source data are provided as a Source Data file.

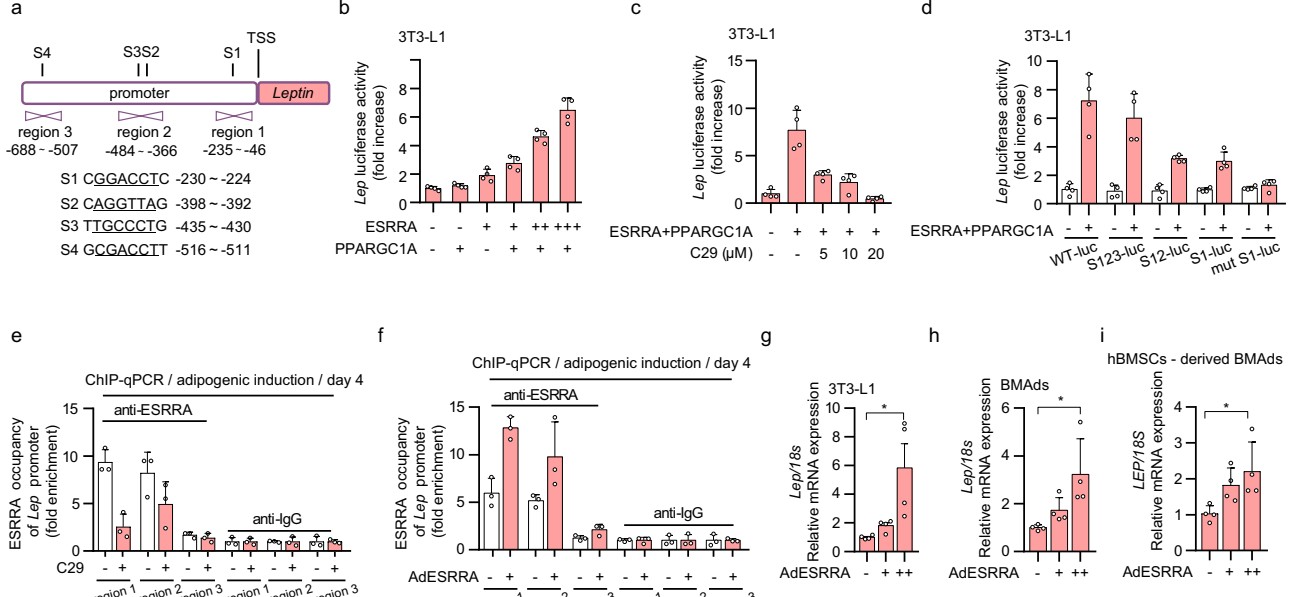

**Fig. 6 | Adipocyte ESRRA positively regulates *Leptin* transcriptional expression by binding to the *Leptin* promoter. a** Schematic diagram illustrating the putative binding sites of the ERRE on the mouse *Leptin* promoter, with four ERREs sequences highlighted by underlined nucleotides (S1, S2, S3 and S4). Positions of qRT-PCR primers on the *Leptin* promoter used in ChIP are shown as region 1, region 2 and region 3. **b** Luciferase reporter activities of the *Leptin* promoter in 3T3-L1 transiently transfected with *Esrra*, *Ppargc1a* or control plasmid. *n* = 4 biologically independent experiments. **c** Effects of compound 29 (C29) on ESRRA and PPARGC1A in the regulation of the *Leptin* promoter by using luciferase reporter assays. 3T3-L1 cells were transfected with *Esrra* and *Ppargc1a* expressing plasmids and treated with the indicated doses of C29 for 24 h. *n* = 4 biologically independent experiments. **d** Effects of ESRRA on different constructs of *Leptin* promoter

activities were tested in 3T3-L1 cells transfected with WT-luc, S123-luc, S12-luc, S1-luc, or mut S1-luc. *n* = 4 biologically independent experiments. ChIP assay with ESRRA antibody or IgG in 3T3-L1 cells after adipogenic induction for 4 days along with treatment of 10 μM C29 or DMSO (**e**), or with an infection of adenovirus expressing ESRRA or control GFP (**f**). *n* = 3 biologically independent experiments. The mRNA levels of *Leptin* in matured 3T3-L1 adipocytes (**g**), murine BMAds (**h**) and human BMSCs-derived BMAds (**i**). After 14 days of adipogenic induction, matured adipocytes were infected with adenovirus expressing ESRRA or control GFP for another 2 days. *n* = 4 biologically independent experiments. All data in this figure are represented as mean ± SD. *p < 0.05. Statistical analysis is performed using one-way ANOVA followed by Bonferroni's post hoc tests (**g**–**i**). Source data are provided as a Source Data file.

## ESRRA ablation drives a reduction of leptin and an increase of SPP1 released from adipocytes synergistically directing BMSCs lineage commitment

Leptin released from adipocytes acts on LepR+ BMSCs dictating fate commitment toward adipogenesis[3,7]. On the other hand, SPP1 directs osteogenic differentiation of BMSCs mainly in an extracellular manner via binding to integrin β1[34]. Altered BMSCs lineage commitment maybe behind the reduced BMAds and enhanced bone formation seen in *Esrra*AKO mice. To assess the effects of secreted factors from distinct adipocytes on MSCs fate determination, we first collected CM from either BMAds or minced gWAT depots isolated from *Esrra*fl/fl and *Esrra*AKO OVX mice (Fig. 7a). BMSCs from wild-type adult mice were then cultured in CM supplemented with osteogenic/adipogenic mixed induction medium as reported[42] (Fig. 7a). The concentration of leptin

was dramatically lower in the gWAT-derived CM from *Esrra*AKO OVX mice (Fig. 7b); whereas, SPP1 level was higher (Fig. 5g). In wild-type BMSCs treated with said gWAT-CM, mRNA levels of well-established osteogenic markers, *Runx2*, *Sp7* and *Bglap* were significantly up-regulated, revealing a shift toward committed osteoblast progenitors (Fig. 7c). On the contrary, adipogenic markers *Pparg*, *Cebpa* and *Fabp4* expression were significantly down-regulated, indicating a reduction in adipogenic differentiation (Fig. 7c). As expected, BMSCs treated with gWAT-CM from *Esrra*AKO OVX mice exhibited a profound increase in osteogenic capability in contrast to a significant decrease in adipogenic potential (Fig. 7d, e). Compared to IgG control, when induction medium containing gWAT-CM from *Esrra*fl/fl OVX mice was supplemented with Allo-aca, a specific leptin receptor antagonist peptide that blocks leptin/LepR signaling, BMSCs preferentially differentiated into

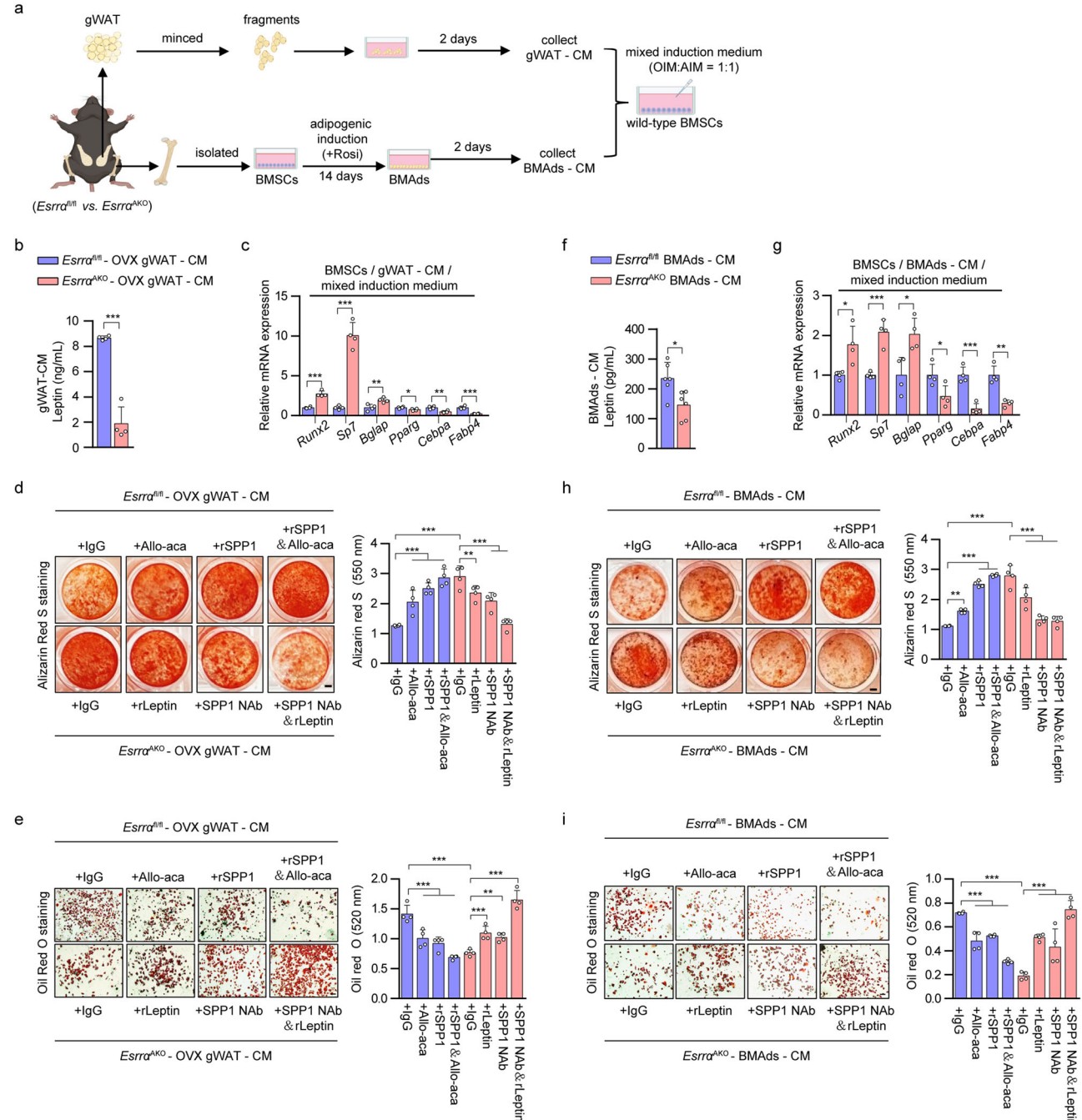

**Fig. 7 | ESRRA-ablated adipocytes secrete increased SPP1 and decreased leptin synergistically dictating BMSCs lineage commitment toward osteogenesis.** **a** Schematic diagram showing the procedure of the conditioned medium (CM) preparation from cultured BMAds or minced gWAT; and wild-type BMSCs were differentiated in osteogenic/adipogenic mixed induction medium (OIM:AIM = 1:1) supplemented with the indicated CM (schematic created with BioRender.com. Agreement number: ZP26KB8SN4). **b** The concentrations of soluble leptin in gWAT-CM prepared from *Esrra*^fl/fl^ and *Esrra*^AKO^ OVX mice were measured by ELISA. *n* = 4 biologically independent samples. **c** mRNA levels of osteogenesis markers *Runx2, Sp7, Bglap,* as well as adipogenic markers *Pparg, Cebpa, Fabp4* in wild-type BMSCs cultured with mixed induction medium and indicated gWAT-CM for 14 days. *n* = 4 biologically independent experiments. Representative images and quantification of alizarin red S staining (**d**) and oil red O staining (**e**) of BMSCs cultured as in (**c**) with an addition of gWAT-CM for 14 days, in the presence of rSPP1, SPP1 NAb,

recombinant leptin (rLeptin), leptin receptor antagonist Allo-aca or IgG as indicated. *n* = 4 biologically independent experiments. Scale bar: 2 mm (**d**); scale bar: 100 μm (**e**). **f** The concentrations of soluble leptin in BMAds-CM as prepared from (**a**). *n* = 6 biologically independent samples. **g** mRNA levels of indicated genes in wild-type BMSCs cultured in mixed induction medium supplemented with the indicated BMAds-CM for 14 days. *n* = 6 biologically independent experiments. Representative images and quantification of alizarin red S staining (**h**) and oil red O staining (**i**) of BMSCs cultured as in (**g**) with the indicated treatments for 14 days. The experiments were conducted according to the procedure shown in (**a**–**e**). *n* = 4 biologically independent experiments. Scale bar: 2 mm (**h**); scale bar: 100 μm (**i**). Data are shown as mean ± SD. *$p < 0.05$, **$p < 0.01$ and ***$p < 0.001$. Statistical analysis is performed using unpaired two-tailed Student's *t* test (**b, c, f, g**) and two-way ANOVA with Fisher's LSD post hoc analysis (**d, h, e, i**). Source data are provided as a Source Data file.

osteoblasts at the expense of adipocytes differentiation (Fig. 7d, e). Additionally, rSPP1 facilitated osteogenesis and inhibited adipogenesis, which was even more profound in combination with Allo-aca, demonstrating their concerted effects (Fig. 7d, e). Conversely, when recombinant leptin (rLeptin) or SPP1 NAb was added to gWAT-CM from *Esrra*[AKO] OVX mice, osteogenic differentiation of BMSCs was reduced; whereas, adipogenic differentiation was enhanced (Fig. 7d, e). Similar results were obtained using BMAds-CM, suggesting the presence of a paracrine effect of BMAds on surrounding BMSCs (Fig. 7f–i). Collectively, these results delineated that the loss of ESRRA in adipocytes alters the production and secretion of leptin and SPP1, acting on BMSCs through either a paracrine or endocrine manner, promoting osteogenic differentiation at the expense of adipogenic differentiation; thereby, shifting the lineage commitment of BMSCs.

### Pharmacological ESRRA inhibition protects bone loss and impedes MAT expansion in DIO mice

Next, we employed a specific ESRRA antagonist C29 to assess the pharmacological effect of ESRRA inhibition on adipocytes (Fig. 8a). In mature 3T3-L1 adipocytes, C29 dose-dependently suppressed leptin but induced SPP1 expression while preserving adipogenesis shown by Oil Red O staining (Fig. 8b, c). Moreover, we recaptured similar results in C29-treated BMAds differentiated from murine or human BMSCs; namely, C29 dose-dependently altered leptin and SPP1 production in an opposite manner (Fig. 8d, e and Supplementary Fig. 8a–e). These evidences confirmed the expression of leptin and SPP1 was dictated by ESRRA but not fate switching. These data proved that in vitro C29 treatment can efficiently inhibit ESRRA activity in adipocytes, raising the prospect of achieving augmented therapeutic efficacy by simultaneously targeting SPP1 and leptin pathways.

To further assess the pharmacological effects of C29 in vivo, we treated DIO male mice with C29 or vehicle for 4 weeks (Fig. 8f). Consistent with our previous report[23], C29 did not alter body weight and visceral fat mass. In comparison to NCD vehicle mice, circulating levels of Lepin, TNFa and IL6 were dramatically enhanced in DIO vehicle mice which were significantly reduced by oral gavage of C29, indicating repressed systemic inflammation due to reduced ESRRA activity (Fig. 8g). Of note, DIO mice had significant bone loss evident from μCT analysis and quantitative bone parameters which was rescued by C29 (Fig. 8h, i). Consistent with a recent study[27], HFD decreased cortical thickness by 7% and BV/TV by 10% both were restored by C29 (Fig. 8j). This protective effect may be in part due to decreased bone resorption shown by a reduction of TRAP staining and blood CTX-1 concentration (Fig. 8k, l). This result is also consistent with ours and other previous reports that pharmacological inhibition of ESRRA directly blocks osteoclastogenesis and bone resorption[43]. Moreover, SPP1 immunofluorescence staining and plasma P1NP level revealed more osteoblasts and bone formation in C29-treated than vehicle DIO mice (Fig. 8m, n). Remarkably, BMAT expansion in response to HFD was diminished by C29 (Fig. 8o, p). These findings illustrate a potential therapeutic option by inhibiting ESRRA function in adipocytes for treating osteopenia associated with high marrow adiposity.

### Discussion

Bone is an active metabolic organ within which different cell types communicate with each other fine-tuning homeostasis. Marrow adipocytes enable bone-forming cells to harness essential metabolic fuels but also render them susceptible to microenvironment damages in bone pathologic niche. Here, we show that adipocyte-specific ESRRA knockout rescues bone loss and type H vessel distortion as well as inhibits marrow adiposity in mice induced by OVX or DIO. Our present analysis identifies a novel function of ESRRA in adipocytes, namely, ESRRA oppositely regulates the expression of leptin and SPP1. Our findings demonstrated that blockade of ESRRA in adipocytes substantially restricted MAT expansion and promoted bone formation by

altering the expression of secreted factors such as leptin, SPP1 and possibly other factors which play synergistic roles on BMSCs fate determination and vascular endothelial cells angiogenesis in an adipocyte-rich bone milieu (Fig. 9). Our results point to new avenues for treating bone disorders especially in clinical conditions associated with high marrow adiposity by chemically targeting ESRRA.

Unlike peripheral WAT and brown adipose tissue (BAT), the pathophysiological role of MAT is largely unexplored. Patients with congenital generalized lipodystrophy (CGL) who lack MAT such as CGL1 and CGL2 develop pathological osteosclerosis, while patients with CGL3 and CGL4 who retain MAT fail to develop high bone density, suggesting that MAT is necessary for bone homeostasis[44]. Distinct from peripheral adipocytes, MAT can change not only size but also BMAds number in response to nutritional cues and hormonal signals. As a precursor reservoir for BMAds, MSCs are highly enriched in the bone marrow and undergo a dynamic differentiation process throughout life[3,32]. Of note, leptin plays an important role in regulating adipogenesis and osteogenesis through acting on LepR+ MSCs in the bone milieu[7]. MAT can reach approximately 30% of total fat mass under pathological conditions[8,45] and may also contribute to leptin contents. However, the endocrine or paracrine effect of MAT-derived leptin is underestimated and most studies on energy balance only focus on leptin produced from peritoneal WAT.

Leptin was originally identified as a circulating hormone involved in feeding behavior and energy homeostasis. The role of leptin in bone formation is heavily debated as evidences for central nervous system and peripheral pathways have both been presented[46,47]. However, analysis of bone phenotypes in these studies are complicated by the fact observed in genetic mutant *ob/ob* and *db/db* mice with global defects of *Lep* and *LepR*, respectively. *ob/ob* and *db/db* mice exhibit severe metabolic disorders, including not only obesity and diabetes but also muscle hypoplasia, hypogonadism and hypercortisolism, accompanied with hypometabolic and hypothermic states, risk factors that can independently influence bone formation. Different from *ob/ob* and *db/db* mice with the abolished leptin/LepR signaling, obese mice induced by a diet of 35% fat develop hyperleptinemia, irrespective of genotype, which are similar to most humans with obesity. Exogenous leptin replacement by either a peripheral or central pathway in leptin-deficient *ob/ob* mice has been reported to play a positive influence in bone formation suggesting that leptin is essential for normal bone homeostasis and might play multifaceted roles not only in bone turnover and metabolic regulation but also under the different scenarios such as leptin-deficiency and hyperleptinemia[48,49]. Leptin has been observed to act on LepR+ BMSCs mediating diet-induced fat accumulation in adult bone marrow at the expense of osteogenesis[7]. Noticeably, high expression of leptin by human bone marrow adipocytes has been observed in primary culture and that human femur MAT adipocytes secrete high levels of leptin[50,51]. Importantly, LepR+ BMSCs were shown to be more abundant in humans with obesity subjects associated with an enrichment of the molecular signature of adipocyte progenitor cells[52]. These observations implicate BMAds-derived leptin may play a paracrine function on surrounding cells such as LepR+ stromal cells. In our study, we identified that leptin released from BMAds imposed similar effects on directing BMSCs fate determination as leptin derived from peritoneal gWAT. Our findings reinforce the negative influence of excess leptin on bone homeostasis in a paracrine manner.

Triggered by excessive leptin/LepR signaling, BMAds expansion may further inflate at the expense of osteoblasts since they arise from common progenitors. Indeed, our data show that circulating level of leptin was significantly enhanced in mice upon estrogen deficiency or overfeeding accompanied with expanded adipocytes and diminished Osterix+ osteoprogenitors which were rescued by ESRRA ablation in adipocytes. But peripheral WAT pads changed little by ESRRA inhibition indicating that BMSCs lineage allocation, as a highly dynamic process, is more responsive to local leptin signaling in the bone

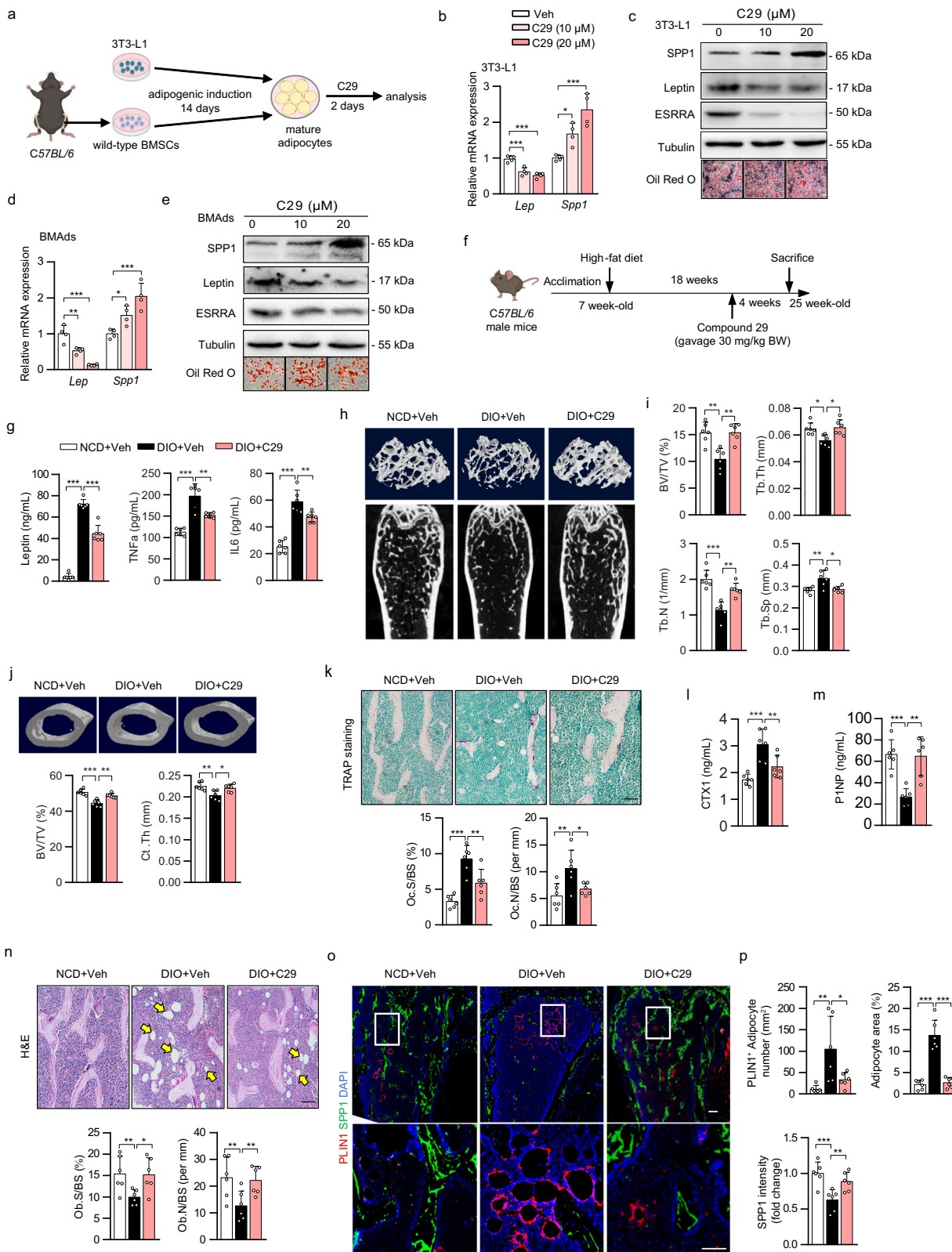

microenvironment. Although we have technical difficulties in deciphering the respective contribution of leptin from distinct sources, our findings from ex vivo adipogenic differentiation culture and conditioned medium assays strongly suggest that ESRRA inhibition in mature BMAds performed essential roles by reducing leptin levels and its local signaling in the bone marrow. Consistent with this concept, recently two independent studies reported that an increased local leptin/LepR response driven by expanded hypodermal adipocytes leads to a chronic wound; and augmented leptin level in malignant tumor microenvironment facilitates benign-to-malignant transition through acting on LepR-expressing cancer stem cells[53,54]. Thus, targeting adipocyte ESRRA may also have implications to restricting the leptin/LepR signaling pathway required for certain pathological progression.

**Fig. 8 | Pharmacological inhibition of ESRRA protects DIO mice from bone loss and MAT expansion. a** Experimental design. Wild-type BMSCs were isolated from C57BL/6 mice and differentiated into BMAds, and 3T3-L1 preadipocytes were cultured in adipogenic medium for 14 days. Mature adipocytes were subsequently treated with C29 for an additional 2 days (schematic created with BioRender.com. Agreement number: IH26KB8VKA). The mRNA and protein levels of leptin and SPP1 in mature 3T3-L1 adipocytes (**b, c**) or BMAds (**d, e**). In vitro experiments were repeated four times. **f** Schematic diagram showing the experimental design for pharmacological treatments in DIO mice. Seven-week-old C57BL/6 mice were fed either a NCD or HFD for 18 weeks, and received either vehicle or C29 (30 mg/kg/body weight) every day during the last 4 weeks (schematic created with BioRender.com. Agreement number: WH26KBA0SE). **g** Plasma leptin, TNFa and IL6 levels.
**h, i** Representative micro-CT images and histomorphometry analysis of BV/TV, Tb.N, Tb.Th and Tb.Sp at the distal femurs. **j** Representative micro-CT images of middle-

segment of cortical bone and histomorphometry analysis of cortical bone volume/tissue volume ratio (BV/TV) and cortical thickness (Ct.Th). **k** Representative images of TRAP-stained femoral sections (scale bar: 100 μm). Quantitative assessment of trabecular Oc.S/BS and Oc.N/BS based on TRAP-stained sections. Plasma CTX-1 (**l**) and P1NP (**m**) levels. **n** Representative images of H&E-stained femur sections (scale bar: 100 μm). Quantitative assessment of trabecular Ob.S/BS and Ob.N/BS based on H&E-stained sections. **o** Representative PLIN1 and SPP1 immunostaining in femoral sections. Scale bar: upper panel, 100 μm; lower panel, 50 μm. **p** Quantification of PLIN1⁺ adipocyte number and of SPP1 fluorescence intensity. Six mice per group were used in all animal experiments. Data are shown as mean ± SD. *$p < 0.05$, **$p < 0.01$ and ***$p < 0.001$. Statistical analysis is performed using one-way ANOVA followed by Bonferroni's post hoc tests (**b–d**) and two-way ANOVA with Fisher's LSD post hoc analysis (**g, i–n, p**). Source data are provided as a Source Data file.

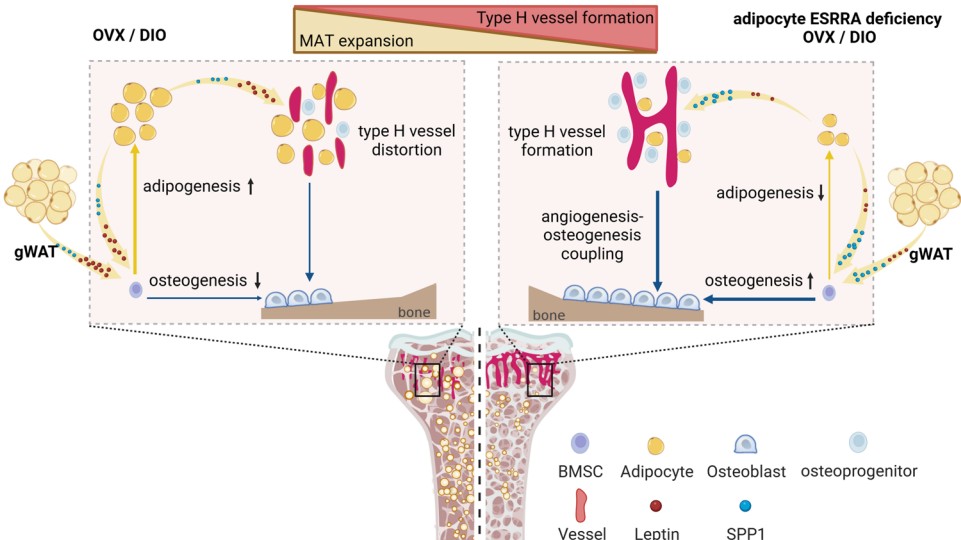

**Fig. 9 | Schematic diagram showing adipocyte ESRRA deficiency preserves osteogenesis and vascular formation in adipocyte-rich bone marrow via oppositely modulating lepin and SPP1.** Estrogen deficiency or high-fat diet-induced obesity results in excessive bone marrow adipocytes and distorted type H

vessel. Adipocyte ESRRA deficiency preserves bone formation and counteracts high marrow adiposity by decreased leptin and enhanced SPP1 secretion, dictating BMSCs fate commitment toward osteogenesis and promoting vessel formation (schematic created with BioRender.com. Agreement number: QL26KB8YYQ).

In the skeletal system, vasculature is not only a conduit system for nutrients and oxygen, but also fundamental for bone formation and haematopoiesis[12,40]. Abundance of type H vessels is also considered as an important indicator of osteopenia in human subjects[14,15]. In line with previous findings, we observed that OVX mice have declined type H vessels and diminished perivascular Osterix⁺ progenitors. Type H vasculature in DIO mice was also apparently distorted and sparse. The functional properties of bone endothelial cells require cell-matrix signaling interactions frequently mediated by endothelial integrin β1 and its binding partners such as matrix glycoprotein SPP1. Loss of endothelial integrin β1 or global *Spp1* knockout results in a disorganized vasculature, that is, shorter type H vessel columns, highly branched metaphyseal capillaries and lack of a straight columnar organization[39]. We noticed that the length and organization of type H vessel columns were significantly improved by adipocyte-specific ESRRA abrogation, accompanied with a notable reduction of BMAds number and size around type H vessels below the growth plate. We revealed that SPP1 can be produced by BMAds in which *Spp1* is downregulated during adipocyte expansion, especially upon estrogen deprivation. However, ESRRA interferes with E2/ESR1-dependent regulation of *Spp1* expression as a result of their recognition of common binding sites. The repression activity of ESRRA on *Spp1* promoter might be attributed to corepressor receptor-interacting protein 140 (RIP140) recruited by ESRRA in adipocytes where RIP140 is highly

expressed[55]. Thus, SPP1 production from ESRRA-deficient adipocytes may account for bone vasculature reconstruction. Previous studies reported that ESRRA regulates SPP1 expression as a transcriptional activator or repressor in a cell context-dependent manner[56–58]. ESRRA positively regulates SPP1 gene in mouse osteoblastic (MC3T3-E1) cell lines or in non-osteoblastic (HeLa) human cell line[56,57]. On the other hand, ESRRA also exerts repression effect on *SPP1* promoter transactivation in human osteosarcoma cell lines SaOs-2 and U2-OS by cross-talking with another nuclear receptor NR4A2[58]. Most of these studies were performed in cell lines in vitro. Here, we revealed that ESRRA acts as a repressor on Spp1 promoter in adipocytes by interrupting with E2/ESR1 signaling and further confirmed the bona fide binding sites by luciferase assays, as well as by ChIP assays in ESRRA-deficiency primary cells which may confer a truly physiological response. However, SPP1 is also involved in a wide variety of physiological and pathological conditions, including inflammatory processes and even cancer which need to be considered in the therapeutically option[59–61].

The direct regulatory role of ESRRA on bone cells remains obscure. Global ESRRA knockout mice displayed no changes in bone mass in male mice but a protective effect on aged bone in female mice[62–64]; whereas other literature reported high bone mass in knockout male mice due to suppressed osteoclastogenesis and bone resorption[65,66]. Moreover, several studies support a positive regulatory role for ESRRA in osteoblast differentiation in vitro by using human or

murine MSCs[67,68]. On the contrary; the absence of ESRRA in mice calvaria cells was demonstrated to promote osteoblast differentiation[62,63]. These discrepancies might be due to differences in age, gender or mice strain. Furthermore, ESRRA regulates numerous genes in different tissues such as intestine, kidney and liver that are involved in bone metabolism, of which the multifaceted roles on bone need to be clarified in a cell or tissue-specific context. Moreover, ESRRA also mediates adaptive thermogenesis of BAT[69,70]. Whether ESRRA deficiency affects bone metabolism through regulating BAT or the beige-like characteristics of MAT especially in cold stress condition need to be further investigated. On the other hand, emerging evidences showed that thermoneutral housing has a broad influence in regulating bone turnover, metabolic and behavioral responses in mice which might better model human physiology and disease[71–73]. Whether adipocyte-ESRRA deficiency mice can be modulated by thermoneutral housing should be paid attention to in future investigations.

BMAds and their precursors LepR $^+$ stromal cells in adult bone marrow, are the major sources of NGF, KitL and VEGF contributing to sustain nerves and promote haematopoietic and vascular regeneration after myeloablation[17,74]. These emerging evidences suggest marrow adipocytes homeostasis are essential, especially in impaired bone marrow environment after irradiation or chemotherapy. Furthermore, the absence of marrow adipocytes induced by global reduction in c-kit receptor function and m-kit ligand function in mice result in exacerbated bone loss, demonstrating that normal marrow adipocytes and hematopoietic lineage cells are required for maintaining bone homeostasis[75–77]. It needs to be determined whether these factors play essential roles in the expanded MAT induced by different pathological conditions. On the other hand, MAT volume varies between age, gender and species. Specifically, fold changes of MAT in rodents models due to metabolic diseases such as obesity generally exceeds that observed in humans[9]. Excessive MAT correlation with bone, vascularization, hematopoiesis or metabolism in humans in these pathophysiological conditions need to be further investigated. Moreover, although we observed adipocyte ESRRA inhibition leads to a significantly augmentation of osteoprogenitors associated with improved type H vessel formation in adipocyte-rich bone marrow, this study is limited without using elderly OVX mice or aged female mice at over 18 months of age to mirror bone turnover in elderly postmenopausal women and preclinical testing of potential therapies for osteoporosis. Hence, considering ageing and species differences, whether adipocyte ESRRA deficiency is protective against bone loss in aged mice or non-human primates is to be further addressed in estrogen-deficiency condition associated with ageing, which is often accompanied by not only accelerated bone resorption but also decreased bone formation due to declined number in BMSCs pool and decreased capability in osteogenic differentiation.

In addition to its pro-angiogenic effect, SPP1 has also been attributed multiple functional roles in promoting MSC migration and adhesion, and facilitating osteogenic differentiation while retarding adipogenic differentiation through its RGD-integrin binding sequence in the soluble form. Recently, a BMSC-secreted factor KIAA1199 has been identified and KIAA1199 blockade facilitates bone formation and regeneration by promoting SPP1/integrin β1-mediated osteogenic differentiation[78]. Importantly, SPP1 as a negatively charged matrix glycoprotein, has a high affinity to calcium which allows it to anchor to and retain in bone environment through the circulation system[41]. On the other hand, systemically administered leptin in rodents and primates has also been reported to be localized and retained in the bone marrow[79]. Hence, leptin and SPP1 may prefer to exert functional roles in the bone microenvironment. Thus, adipocyte expansion is detrimental to bone and vasculature, and further exacerbates BMAds accumulation at the expense of osteoblasts, forming a vicious feed forward loop for bone deterioration. Targeting adipocyte ESRRA might impede the disease-promoting alterations in the adipocyte-rich

bone microenvironment via concomitantly modulating two pathways leptin and SPP1.

## Methods

### Adipocyte-specific *Esrra* knockout mice generation

Homozygous *Esrra*flox/flox mice were described previously[23]. Both ends of the exon 2 of *Esrra* gene are flanked by loxP sites that are oriented in the same direction. The Adipoq-Cre mice [B6; FVB-Tg(Adipoq-cre)1Evdr/J] (JAX Strain: 010803) were obtained from the National Resource Center for Mutant Mice of China. The Cre/loxP system was used to generate adipocyte-specific *Esrra* knockout mice (Adipoq-Cre; *Esrra*fl/fl, refer as *Esrra*AKO) by consecutive mating of *Esrra*flox/flox mice with AdipoqCre mice. The Cre-negative *Esrra*flox/flox (*Esrra*fl/fl) mice were employed as the control genotype. All mice were maintained in a specific-pathogen-free facility at 24 ± 2 °C with 60% ± 5% humidity under 12 h light/dark cycles which was defined as normal conditions. All procedures used for animals and their care in this study were reviewed and approved by the ethical committee at Shenzhen Institute of Advanced Technology, Chinese Academy of Sciences.

### Animal models and treatments

For DIO, male *Esrra*fl/fl and *Esrra*AKO mice at 9 weeks old were fed either a normal chow diet (NCD) or a high-fat diet (HFD, 60% kcal in fat (about 35% fat); D12492, Research Diets) for up to 16 weeks. After 12 weeks of HFD feeding, oral glucose tolerance tests were performed in mice fasted overnight. Core body temperature was measured by a clinical rectal thermometer (Thermalert model TH-5; Physitemp). Mice were singly housed and food intake was measured daily for 5 days. Blood glucose levels were measured from the tail vein before and at 30, 60, 90 and 120 min after administering glucose (2 g/kg body weight), using the ACCU-CHEK blood glucose meter (ROCHE #06870279001). For OVX, 10-week-old female *Esrra*fl/fl and *Esrra*AKO mice were subjected to either bilateral ovariectomy to mimic postmenopausal osteoporosis or sham operation. Eight weeks post-operation, the mice were euthanized by cervical dislocation, under isoflurane-mediated anesthesia.

Wild-type C57BL/6 mice were purchased from GemPharmatech (Nanjing, China) for pharmacological experiments. 7-week-old male mice were randomized into two groups: NCD and DIO groups. The DIO mice were fed with a high-fat diet for 14 weeks to induce obesity, and they were then divided into two subgroups: the DIO + C29 group and the DIO+Veh group. The mice in DIO + C29 group were administered an ESRRA inverse agonist C29 (ChemPartner #S1039) via oral gavage at a dose of 30 mg/kg body weight daily for 4 weeks. The DIO + Veh group was administered an equal volume of vehicle, which was composed of 10% vitamin E-TPGS, 20% PEG400, and 70% water. Both DIO + Veh group and DIO + C29 group were exposed to 4 additional weeks of HFD feeding and treatments. Mice in the NCD + Veh group were fed a regular chow diet throughout the experiment and were administered the equal volume of vehicle as described in DIO + Veh group by oral gavage for the last 4 weeks. After fasting overnight, all mice were euthanized by cervical dislocation under isoflurane-mediated anesthesia, and then the samples were collected for analysis. Femurs were collected for micro-CT, histological and immunofluorescence analysis. Plasma glucose, triglyceride (TG, Applygen #E1003) or free fatty acid (FFA, WAKO #294-63601) were measured using commercial kits.

### Bone micro-CT analysis

Femurs were fixed in 4% paraformaldehyde for 48 h. Micro-CT analysis of fixed femur was performed using a Skyscan 1176 (Bruker, Kontich, Belgium) at 50 kVp, 450 µA and a resolution of 9 µm. The region of interest (ROI) was then segmented and analyzed using both the DataViewer software and CTAn software. For assessing trabecular bone, the ROI starting point was defined as 0.36 mm from the metaphyseal growth plate and extended 1 mm distal from that start point. Regions 0.5 mm above and 0.5 mm below the mid-diaphysis (50% of

length) were selected to evaluate cortical bone. Three-dimensional model visualization software was applied to display the metaphyseal trabecular bone and the diaphyseal cortical bone.

## Bone histology and histomorphometry

Fixed femurs were decalcified for 3–4 weeks at 4 °C using 0.5 M EDTA (pH 7.2) before being sectioned at 5 μm. The paraffin-embedded femurs were subjected to deparaffinization and rehydration procedures before being stained using hematoxylin and eosin (H&E) kit (Beyotime #C0105S) or tartrate-resistant acid phosphatase (TRAP) staining kit (Solarbio #G1492). Images were obtained using a fluorescent microscope (Olympus #BX53). The quantification of osteoblasts or osteoclasts was then measured and calculated using ImageJ software. For calcein double labeling, 20 mg/kg of calcein (Sigma-Aldrich #C0875) dissolved in 2% sodium bicarbonate solution, was administered to the mice via intraperitoneal injection at 8 and 3 days before euthanasia. Femurs were soaked in 70% ethanol, directly embedded in methyl methacrylate, and sagittally sectioned at 10 μm. Images of calcein double labeling in undecalcified femora sections were acquired on a fluorescent microscope. Dynamic indices of overall bone formation rate (BFR) and mineral apposition rate (MAR) were calibrated by ImageJ software.

## Bone immunofluorescence

For immunofluorescence staining, 5-μm bone sections were soaked in sodium citrate buffer at 65 °C overnight for antigen retrieval. After being permeabilized with 0.2% Triton X-100 and blocked with 5% BSA for 1 h, the sections were then incubated overnight at 4 °C with primary antibodies SPP1 (1:200; Novus Biologicals #AF808), PLIN1 (1: 200; Cell Signaling Technology #9349), Osterix (1:200; Abcam #ab209484), Endomucin (V.7C7) (EMCN,1:50; Santa Cruze #sc-65495), or CD31/PECAM-1 Alexa Fluor 488-conjugated Antibody (1:200; R&D system #FAB3628G). After primary antibody incubation, sections were washed and incubated with appropriate secondary antibodies Alexa Fluor 488-conjugated donkey anti-goat IgG (1:500; Thermo Fisher #A-11055), Alexa Fluor 555-conjugated donkey anti-rabbit IgG (1:500; Abcam #ab150074), or Alexa Fluor 555-conjugated goat anti-rat IgG (1:500; Bioss #bs-0293G-AF555) for 1 h at room temperature. Sections were then washed and nuclei were counterstained with DAPI (Cell Signaling Technology #4083) for 30 min at room temperature. After counterstaining, sections were washed and mounted with anti-fade reagent (Invitrogen #P10144). The images were captured by a fluorescent microscope.

The quantification of type H vessels was performed using ImageJ software. The area around the vessels with a range of 20 mm was lined out and the fluorescence signals based on color recognition within this area were quantified using the "area sum" parameter. The number and area of PLIN1 positive adipocytes were counted manually in each bone marrow section of distal femur by using ImageJ. For some indexes, after the signals were absolutely quantified, relative quantification was performed by taking the numerical value of one group as the reference and calculating the fold changes of other groups.

## White adipose tissue histology and immunofluorescence

gWAT was fixed in 4% paraformaldehyde for 24 h. H&E staining was performed on paraffin-embedded gWAT sections, and the sections were also subjected to immunofluorescent staining using primary antibodies against SPP1 and leptin (1:200; Abcam #ab16227). Adipose cell size and fluorescence intensity were measured using ImageJ software in a semi-automated manner.

## BMSCs isolation and culture

Mouse bone marrow MSCs (BMSCs) were isolated from tibia and femur. Briefly, cells were rinsed out with sterile syringe and then cultured in α-MEM (Gibco #C12571500BT) with 10% FBS (Gibco #10099141C) and 1% penicillin/streptomycin (Hyclone # SV30010). Cells were incubated in 5% $CO_2$ incubators at 37 °C. After 8 h, non-adherent cells in the supernatants were transferred to a new culture dish, and adherent cells were further cultured with fresh medium. Following overnight culture, all non-adherent cells were washed out, and adherent cells were then fed with fresh medium. At 80%–90% confluence, the BMSCs were digested with 0.25% Trypsin (Invitrogen #25200056) and passaged with medium being changed every 48–72 h. Human BMSCs (hBMSCs) were purchased from Cyagen Biosciences Inc. (#HUXMF-01001) and were observed to have a high level of purity, with >70% expressing CD29, CD44, and CD105, and <5% expressing CD34 and CD45 in flow cytometry assays. The maintaining of hBMSCs in growth medium was carried out with the procedure described above. Cells within 3 passages were used for in vitro experiments.

## Adipogenic and osteogenic differentiation

For osteogenic differentiation, BMSCs were cultured in osteogenic induction medium comprised of dexamethasone (100 nM), β-glycerophosphate (10 mM, Sigma # G9422) and ascorbic acid (50 μg/ml, Sigma #A8960). The medium was changed every 3 days for 14 days. Differentiated osteoblasts were stained with 2% alizarin red S (Beyotime #C0138). The amount of mineral content was measured by eluting the alizarin red S with 10% cetylpyridinium chloride and the optical density was measured at OD 550 nm. For adipogenic differentiation of BMSCs, cells were cultured in adipogenic induction medium containing insulin (10 mg/ml, Yeasen #40112ES25), IBMX (0.5 mM, Sigma #I5879), dexamethasone (1 μM, Sigma # D4902), in the presence or absence of 1 μM rosiglitazone (Rosi, Sigma# R2408) for 7 or 14 days according to experimental design. For adipogenic differentiation of 3T3-L1 cells, cells were stimulated with adipogenic induction medium for 14 days. When specified, matured 3T3-L1 adipocytes or BMAds were treated with C29 or vehicle for an additional 2 days. Alternatively, they were infected with adenovirus expressing ESRRA (50 pfu) or GFP for 48 h. Adenoviruses were produced and purified as we described previously[23]. Adipogenic differentiation was assessed by oil red O staining following the manufacturer's instructions (Solarbio #G1262).

## Conditioned medium preparation

To prepare gWAT conditioned medium (CM), gWAT from *Esrra*[fl/fl]-OVX and *Esrra*[AKO]-OVX mice were minced and then incubated in α-MEM for 48 h before collection. To prepare BMAds-CM, BMSCs were isolated from *Esrra*[fl/fl] and *Esrra*[AKO] mice and then subjected to adipogenesis for 14 days to achieve fully differentiated BMAds. The adipogenic induction medium was replaced with fresh α-MEM for an additional 2 days before collection. The collected CM were then centrifuged for 5 min at $1000 \times g$ to remove cell pellets. The resulting supernatant was then stored at −80 °C.

## Mixed differentiation

Before preparing the mixed differentiation medium, gWAT-CM was diluted 10-fold with fresh medium to obtain 10% gWAT-CM. The mixed induction medium was composed of 50% adipogenic medium and 50% osteogenic medium, with an adipogenic to osteogenic induction ratio of 1:1 as reported[42]. Wild-type BMSCs were differentiated in the mixed induction medium supplemented with 10% WAT-CM or BMAds-CM as indicated, in the presence of 0.5 μg/ml recombinant SPP1 protein (rSPP1, Abcam #ab281820), 1 μg/ml SPP1 neutralizing antibody (SPP1 Nab, Novus Biologicals #AF808), 0.5 μg/ml recombinant leptin protein (rLeptin, R&D Systems #490-OB-01M), 250 nM leptin receptor antagonist Allo-aca (MCE #HY-P3212) or an equal volume of vehicle IgG (Cell Signaling Technology #2729). The formation of mineralized nodules was evaluated by alizarin red S staining, and adipocytes were distinguished by oil red O staining.

## Migration and tube formation assay

For transwell migration assays, gWAT-CM or BMAds-CM was added in the lower chamber supplemented with 0.5 µg/ml rSPP1, 1 µg/ml SPP1 Nab or an equal volume of vehicle IgG. Mouse bEnd.3 microvascular endothelial cells (ECs) were seeded into the upper chambers of 24-well plates containing an 8-µm membrane pore size (c) and were incubated for 24 h. The migratory cells that had moved to the underside of the membrane were fixed and stained with 1% crystal violet (Beyotime #C0121). The number of migratory cells was determined by counting the cells that penetrated the membrane in 5 random fields per chamber using an optical microscope.

For tube formation assays, Matrigel (BD Biosciences #354230) was added to 24-well culture plates and polymerized for 1 h at 37 °C. After overnight starvation, ECs were resuspended with gWAT-CM or BMAds-CM and seeded at a density of $2 \times 10^4$ cells/well onto the polymerized Matrigel in plates. Cells were then incubated with 0.5 µg/ml rSPP1 or 1 µg/ml SPP1 Nab at 37 °C for 4 h, and then tube formation was assessed with an inverted microscope. The parameters of tube length and the number of branch points were analyzed using the angiogenesis analysis plugin of the ImageJ software. The number of migratory cells was determined by counting the cells that penetrated the membrane in 5 random fields per chamber.

## Enzyme-linked immunosorbent assay

ELISA was conducted to detect SPP1 (R&D systems #MOST00), leptin (R&D systems #MOB00B), CTX1 (USCN #CEA665Mu), PINP (NOVUS biologicals #NBP2-76466), TNFa (USCN #MEA133Mu) or IL6 (USCN #MEA079Mu) from plasma or CM based on manufacturer's instructions. The concentrations of SPP1 were measured by diluting plasma 300-fold, gWAT-CM 300-fold, BMAds-CM 150-fold and matured 3T3-L1 adipocytes-CM 150-fold.

## Plasmid construction and dual luciferase reporter assay

*Esrra, Ppargc1a* and *Esr1* were cloned into pcDNA4 vector as we previously reported[23]. The DNA encoding ESRRA protein without DNA-binding domain (DBD) sequence was subclone into pcDNA4 expression vector to construct ERSRRA-ΔDBD expressing plasmid (+565 bp - +1272 bp). The predicted promoter regions of *Spp1* and *Leptin* were cloned into pGL3-basic luciferase reporter vector, generating pGL3-*Spp1*-WT-luc and pGL3-*Leptin*-WT-luc. Wild-type *Spp1* promoter contains region1 and region2, and the truncated *Spp1* promoter contains only region 1 (−597 bp - +44 bp) (indicated as ΔR2-luc). Serial constructs include a promoter fragment containing site 1–3 (pGL3-Leptin-S123-luc), a promoter fragment containing site 1–2 (pGL3-Leptin S12-luc), and a promoter fragment containing site 1 (pGL3-Leptin -S1-luc). The putative ESRRA binding site 1 was mutated using PCR-based site-directed mutagenesis and QuikChange Site-Directed Mutagenesis Kit (Stratagene #200518) to obtain mutated constructs pGL3-*Leptin*-mut 1-luc.

For transfection, 3T3-L1 cells were transiently transfected with promoter-luciferase reporters and along with several expression constructs as indicated in each figure using Lipofectamine 3000 transfection reagent (Invitrogen #L3000015) and Opti-MEM reduced serum medium (Gibco #11058021). For the *Leptin* promoter reporter assay, 3T3-L1 were transfected with *Leptin* promoter reporters and either *Esrra, Pgargc1a* or control vector, together with C29 or DMSO for 24 h. For the *Spp1* promoter reporter assay, the transfected cells were cultured in phenol red-free DMEM (Gibco #A1048801) supplemented with 10% charcoal-stripped FBS for 24 h. Cells were treated with either *Esrra, Ppargc1a, Esr1* or control vector, supplemented with or without 17β-estradiol (E2,10 nM) for another 24 h, and were then exposed to adipogenic medium for 48 h. The cells were lysed for dual luciferase reporter gene assay kit (Promega #E2980), and the transfection efficiency was normalized to Renilla. Primers used to generate plasmid constructs are listed in Supplementary Table 1.

## ChIP assay

ChIP assays were performed using a ChIP assay kit (Cell Signaling Technology #9005) per the manufacturer's instructions. Briefly, cells were fixed with 1% formaldehyde, quenched with glycine solution, and then washed with PBS. Nuclei were extracted and micrococcal nuclease digestion was performed, followed by sonication to obtain 200–500 bp genomic DNA fragments. Chromatin complexes were immunoprecipitated with primary antibody as below and incubated with protein G magnetic beads (Cell Signaling Technology #9006) overnight. To determine the repression effect of ESRRA on the *Spp1* promoter, BMSCs from *Esrra*[fl/fl] and *Esrra*[AKO] mice were infected with adenovirus expressing ESRRA (50 pfu) or control GFP for 24 h, or 3T3-L1 cells were transiently transfected with ESRRA and/or ESR1 expression plasmids for 12 h, and cells were further stimulated with adipogenic induction medium in phenol red-free DMEM supplemented with 10% charcoal stripped FBS with or without E2 (10 nM). After 4 days, cells were subjected to ChIP assay with primary antibody against ESR1 (2 µl/IP, Abcam #ab32063), or ESRRA (10 µl/IP, Cell Signaling Technology #13826). Mock immunoprecipitations were performed by utilizing normal rabbit IgG (1 µg/IP, Cell Signaling Technology #2729). To determine physical binding of ESRRA on the *Lepin* promoter, 3T3-L1 cells were subjected to adipogenic induction for 4 days with treatment of 20 µM C29 or DMSO, or with addition of adenovirus expressing ESRRA or control as described above. These cells were then subjected to ChIP assays with primary antibody against ESRRA or normal rabbit IgG. The precipitated DNA was purified and used as a template for PCR using primers specifically designed to amplify a segment of 150–250 bp covering the putative ESRRA or ESR1 binding sites. Primer sequences used for ChIP-qPCR are listed in Supplementary Table 2. The amount of immunoprecipitated DNA in each sample was normalized to IgG-associated DNA and presented as relative fold enrichment.

## RNA isolation and qRT-PCR

Total RNA was extracted from homogenized gWAT or cells using AG RNAex Pro Reagent (Accurate Biotechnology #AG21101). Reverse transcription was performed from 2 µg of total RNAs using HiScript III 1st Strand cDNA Synthesis Kit (Vazyme #R312). qRT-PCR was employed to quantify the mRNA levels. RealStar Green Fast Mixture (Genstar #A301-10) was used as the detection reagent, and the primers listed in Supplementary Table 3 were applied. The quantity of mRNA was calculated using the 2^−ΔΔCt method. All reactions were performed at least four independent experiments.

## RNA-seq and gene set enrichment analysis

Total RNA was extracted using RNA isolation kit (Vazyme #RC112) from BMSCs from *Esrra*[fl/fl] and *Esrra*[AKO] mice, which were subjected to adipogenic induction for 7 days. The total RNA quantity and purity were analyzed by using Bioanalyzer 2100 and RNA 6000 Nano LabChip Kit (Agilent, CA, USA, 5067-1511). RNA-seq libraries were constructed and then sequenced using an Illumina Novaseq™ 6000 (LC-Bio Technology, China) following the vendor's recommended protocol. The difference expression of genes was analyzed using DESeq2 software and then subjected to enrichment analysis of GO functions and KEGG pathways. Significantly enriched GO terms and KEGG pathways were selected as follows: $p$ value < 0.05. FPKM (fragments per million reads per kilobase mapped) method was used to normalize gene expression, and lowly expressed genes (average FPKM < 10) were filtered in each sample. Then, heat map analysis was performed to visualize the $Z$-score calculation values for relative expression of secreted factors, oxidative phosphorylation, tricarboxylic acid cycle, lipid metabolism and glycolysis genes, which were identified across two experimental group comparisons ($p$ value < 0.05).

## Immunoblot analysis

Cells or tissues were collected and lysed in RIPA buffer (Beyotime #P0013B) containing a cocktail of protease inhibitors and phosphatase inhibitors. The protein concentration was quantified by BCA Assay (Thermo Fisher #23227). The lysate was mixed with SDS loading buffer and heated at $100\,°C$ for 10 min. Subsequently, $30–60\,\mu g$ of proteins were loaded onto gel for SDS-PAGE analysis. After SDS-PAGE analysis, proteins were transferred onto a PVDF membrane (Merck Millipore #IPFL00010) and blocked with 5% skimmed milk-PBST. Then, the membrane was incubated overnight with primary antibodies that were dissolved in 5% BSA. The antibodies used for immunoblot analysis were anti-ESRRA (1:1000), anti-tubulin (1:5000, BPI #AbM9005-37B-PU), anti-leptin (1:500), anti-SPP1 (1:1000) or anti-GAPDH (1:5000, Proteintech #60004-1-Ig). The immunocomplexes were incubated with corresponding secondary antibodies: HRP-goat anti-mouse IgG (EarthOx Life #E030110-02), HRP-goat anti-rabbit IgG (EarthOx Life #E030120-02) or HRP-rabbit anti-goat IgG (ABclonal #AS029). They were then detected with ECL luminescence reagent (Millipore #WBLUR0500). Images of the samples were captured using ChemiDoc XRS chemiluminescence imaging system (Bio-Rad).

## Statistical analysis

Sample and animal distribution among groups were randomized. All data distribution was assumed to be normal, but this was not formally tested. Differences across more than two groups were analyzed with one-way ANOVA followed by Bonferroni's post hoc tests or two-way ANOVA with Fisher's LSD post hoc multiple comparisons tests, as described in the figure legends. Unpaired, two-tailed Student's $t$ test was used for comparisons between two groups. Data are presented as mean $\pm$ SD. $p$ value $< 0.05$ was considered statistically significant. GraphPad Prism 8.0 and SPSS Statistics 26.0 were applied for statistical analyses.

## Reporting summary

Further information on research design is available in the Nature Portfolio Reporting Summary linked to this article.

## Data availability

The RNA sequencing data generated in this study have been deposited in the Gene Expression Omnibus (GEO) database under the accession code GSE248799. All other data generated in this study are provided in the Supplementary Information/Source Data file. Any additional information is available upon request to the corresponding author (M.G., min.guan@siat.ac.cn) Source data are provided with this paper.

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

## Acknowledgements

This work was supported in part by grants from the National Key Research and Development Program of China (2023YFB3810200 and 2018YFA0703100), National Natural Science Foundation of China (82372464, 82072493, 81770882, 81570532 and 81972071), Guangdong Basic and Applied Basic Research Foundation (2022A1515010528, 2023A1515110143), Shenzhen Medical Research Fund (D2301004), Shenzhen Science and Technology Program (JCYJ20210324101800002), Fund of State Key Laboratory of Phytochemistry and Plant Resources in West China (P2022-KF12).

## Author contributions

T.H. and M.G. conceived, designed the work, analyzed results, and wrote the manuscript. T.H. carried out most of the experiments. Z.L., Z.W., L.G. and J.G. helped with some experiments. N.Z. and L.C. helped with bioinformatics analysis. C.W. proofread the manuscript. M.G., X.Z., H.W., C.G., H.P., K.W.K.Y. and W.L. secured funding. All authors reviewed the manuscript.

## Competing interests

The authors declare no competing interests.
