## [Peer Review File · Nature Communications]

Targeting adipocyte ESRRA promotes osteogenesis and vascular formation in adipocyte-rich bone marrowREVIEWER COMMENTS

Reviewer #1 (Remarks to the Author):

Targeting adipocyte ESRRA promotes osteogenesis and vascular formation in adipocyte-rich bone marrow

General Comments

Huang *et al.* provide evidence that deletion/down regulation of *Esrra* expression influences bone metabolism, particularly in mice subjected to ovariectomy (OVX) or fed a high fat diet (60% energy from fat) to achieve diet-induced obesity (DIO). While *Esrra* deletion/depletion appears to have positive effects on cancellous bone volume fraction and bone formation and a negative effect on bone marrow adipose tissue in OVX'd and DIO mice, the putative mechanisms (redirection of differentiation from adipocytes to osteoblasts due to lower levels of leptin and increased *Spp1* expression in adipocytes) are primarily based on associations and not demonstration of causation. Additionally, there is a wealth of evidence, not considered by the authors, that does not support their interpretations of the results.

Specific Comments

Abstract

1. Nomenclature: It is not clear why the authors are capitalizing the letters in leptin and adiponectin.
2. Abstract line 32: “*Ectopic bone marrow adipocytes.*” The term ectopic generally refers an abnormal place or position. It is not clear why the authors believe bone marrow adipose tissue is abnormal.
3. Abstract lines 32-33: “... (*BMAds*) accumulation occurring under diverse pathophysiological conditions leads to bone deterioration.” While the authors may hypothesize that this is the case, the reviewer is unaware of any evidence that causally links bone marrow adipocytes to bone deterioration. Indeed, the very high levels of adipose tissue in caudal vertebra of mice (compared to DIO associated increases in femur, does not lead to bone deterioration. Please revise.
4. Abstract lines 34-35: “*Here, we show that adipocyte-specific ESRRA deficiency rescues osteogenesis and vascular formation in adipocyte-rich bone marrow due to estrogen deficiency or obesity.*” While the authors present evidence that ESRRA deficiency increases bone formation and vascular formation, this statement overinterprets the findings because associations, not causality are reported.
5. Abstract lines 35-40: “*Mechanistically, adipocyte ESRRA interferes with E2/ESR1 signaling resulting in transcriptional repression of secreted phosphoprotein 1 (Spp1); and positively modulates Leptin expression by binding to its promoter. ESRRA abrogation results in enhanced SPP1 and decreased LEPTIN secretion from both visceral adipocytes and BMAds, concertedly dictating bone marrow stromal stem cell fate commitment and restoring type H vessel formation, constituting a feed-forward loop for bone formation.*” The authors’ data provide limited support for these assertions. Also, the authors have not acknowledged or addressed literature that does not support the proposed mechanism. Briefly, most recent literature supports the following:
 - a) While there is often a reciprocal relationship between osteoblasts and bone marrow adipocytes, there are many exceptions, and osteoblasts and bone marrow adipocytes can be independently regulated by factors such as genetic manipulation, drugs, hormones, and housing temperature. It is notable that the authors are performing studies in growing OVX'd mice, in which cancellous bone loss occurs because of increased osteoclast-mediated bone resorption, not reduced bone formation. Despite increases in white adipose tissue (WAT) and bone marrow

adipose depots, OVX results in larger bones in growing rodents due to increased radial bone growth (osteoblasts) and longitudinal bone growth (growth plate chondrocyte hypertrophy and proliferation).

Excess marrow adipose tissue did not influence the magnitude of cancellous bone loss induced by skeletal disuse in *ob/ob* mice whereas absence of adipose tissue in *W/W^v* exaggerated bone loss.

Absence of bone marrow adipose tissue in *W/W^v* mice had no effect on cancellous bone loss following OVX.

The following are representative publications: (DOI: [10.1016/0378-5122\(96\)01015-8](https://doi.org/10.1016/0378-5122(96)01015-8) ; DOI: [10.1210/edrv-15-3-275](https://doi.org/10.1210/edrv-15-3-275); DOI: [10.1152/ajpendo.00646.2011](https://doi.org/10.1152/ajpendo.00646.2011); DOI: [10.1172/JCI6730](https://doi.org/10.1172/JCI6730)). DOI: [10.1038/s41598-019-45587-0](https://doi.org/10.1038/s41598-019-45587-0)

- b) Most studies performed during the last 2 decades report that plasma leptin has positive effects on bone formation by increasing the differentiation and activity of osteoblasts. Leptin treatment consistently reduces WAT mass and has been shown to decrease bone marrow adipose tissue. This evidence includes (1) leptin stimulation of osteoblast differentiation and matrix mineralization of bone marrow stromal cells *in vitro*, (2) evaluation of the skeletal phenotype of leptin-deficient *ob/ob* and leptin receptor-deficient *db/db* mice, (3) evidence that the osteopenic and osteopetrotic skeletal phenotypes of these mice are not due to obesity, (4) dose response studies demonstrating that leptin increases bone formation at levels having little or no effect on energy metabolism, and (5) adoptive transfer of leptin receptor-deficient cells from *db/db* mice into WT mice recapitulates the low bone formation phenotype of *db/db* mice without influencing energy metabolism.

The following are representative publications: (DOI: [10.1016/j.regpep.2006.04.013](https://doi.org/10.1016/j.regpep.2006.04.013) ; DOI: [10.1097/MCO.0b013e3282f795cf](https://doi.org/10.1097/MCO.0b013e3282f795cf); DOI: [10.1016/j.bone.2003.11.020](https://doi.org/10.1016/j.bone.2003.11.020); DOI: [10.1002/jbmr.36.7](https://doi.org/10.1002/jbmr.36.7); DOI: [10.1530/JOE-16-0484](https://doi.org/10.1530/JOE-16-0484); <https://doi.org/10.1210/endo.140.4.6637>; DOI: [10.1530/JOE-14-0224](https://doi.org/10.1530/JOE-14-0224); DOI: [10.1002/jbmr.1812](https://doi.org/10.1002/jbmr.1812)); DOI: [10.1002/jbmr.406](https://doi.org/10.1002/jbmr.406); DOI: [10.3389/fendo.2022.959743](https://doi.org/10.3389/fendo.2022.959743)).

6. Abstract lines 42-43: “*Thus, our findings highlight a therapeutic approach via targeting adipocyte ESRRA to preserve bone formation especially in detrimental adipocyte-rich bone milieu.*” There is evidence that inhibiting the expression of ESRRA will not necessarily increase bone formation. For example, ethanol suppresses *Esrra* expression. This did not lead to increased *Spp1* expression and bone formation, nor decreased accrual of bone marrow adipose tissue. Instead, the opposite was found (doi: [10.1111/acer.14185](https://doi.org/10.1111/acer.14185)). Additionally, increasing *Spp1* expression is known to lead to undesirable side effects (see last comment for details).

Introduction

7. Adipocytes express kit ligand (stem cell factor), which, as noted by the authors, plays an important role in regulation in hematopoietic cell lineage differentiation and function. However, there is also strong evidence that kit receptor/kit ligand signaling is important for bone marrow adipocyte differentiation. The critical findings relevant to the manuscript under review but not addressed by the authors are:
- a) Mice deficient in kit signaling have abnormalities in fat metabolism, including decreased fat accrual in bone marrow. Despite lack of bone marrow adipose tissue, kit receptor-deficient mice do not differ from WT mice in OVX-induced bone loss and have an exaggerated negative response to the hindlimb unloading model for skeletal disuse.

- b) Pharmacological blockade of kit signaling with the receptor tyrosine kinase antagonist gleevec decreases bone marrow adipose tissue in rats, but also decreases bone formation.
- c) Compared to WT mice, bone marrow from kit receptor-deficient *W/W^v* mice has a drastically reduced capacity to support adipocyte differentiation *in vitro*.
- d) Kit receptor-deficient *W/W^v* and kit ligand-deficient *Sl/Sl^d* mice have no adipocytes in bone marrow of their long bones or lumbar vertebra.
- e) *Sl/Sl^d* mice, despite having no bone marrow adipocytes, have reduced bone formation *in vivo* (not increased as predicted by the authors) and have a reduced capacity to form osteoblasts from bone marrow *in vitro*.
- f) *Sl/Sl^d* mice are unable to produce membrane kit ligand but produce soluble kit ligand. Administration of soluble kit ligand to *Sl/Sl^d* mice does not rescue their inability to form bone marrow adipocytes. This finding suggests that cells with kit receptor must directly associate with cells with membrane kit ligand to form bone marrow adipocytes.
- g) Kit receptor-deficient *W/W^v* mice have reduced expression of *Esrra* and no bone marrow adipocytes. However, bone formation is not universally increased in these mice.
- h) Adoptive transfer of purified hematopoietic stem cells into kit receptor-deficient *W/W^v* mice normalizes kit receptor expression in bone marrow. However, *Esrra* expression remains low in these mice, and they remain unable to form bone marrow adipocytes.

The following are representative publications: (DOI: [10.1371/journal.pone.0131192](https://doi.org/10.1371/journal.pone.0131192); DOI: [10.1007/s11914-018-0422-3](https://doi.org/10.1007/s11914-018-0422-3); DOI.org/[10.1002/ajh.2830080404](https://doi.org/10.1002/ajh.2830080404); DOI: [10.1667/RR15164.1](https://doi.org/10.1667/RR15164.1); DOI: [10.1016/j.bone.2012.11.034](https://doi.org/10.1016/j.bone.2012.11.034)).

Methods

8. Please report the temperature that the mice were housed at. ESRRA-null mice are cold intolerant and unable to maintain their core body temperature due to impaired adaptive thermogenesis (DOI: [10.1073/pnas.0607696104](https://doi.org/10.1073/pnas.0607696104); DOI: [10.1016/j.isci.2018.03.005](https://doi.org/10.1016/j.isci.2018.03.005)). In contrast to humans and larger rodents (e.g., rats) who are homeotherms, mice are facultative heterotherms. This is relevant because increased adaptive thermogenesis induced by housing mice below their thermoneutral zone (~32°C) results in rapid cancellous bone loss from the femur (DOI: [org/10.1007/s00198-016-3634-3](https://doi.org/10.1007/s00198-016-3634-3); DOI: [org/10.1002/ajpa.23684](https://doi.org/10.1002/ajpa.23684); DOI.org/[10.1038/s41574-020-00424-7](https://doi.org/10.1038/s41574-020-00424-7)). Thus, it is plausible that the stimulatory effect of ESRRA deficiency on bone is due to their impaired thermogenesis. In other words, these mice do not experience reduced bone formation and increased BMAT associated with upregulated thermogenesis when housed at room temperature.
9. The description of the statistical analysis is vague but the methods seem deficient in rigor. It is not clear whether the assumptions underlying the statistical tests were assessed. The authors did not test whether OVX differed from Sham surgery, yet make statements in the test (e.g., bone loss) as if they had done so. While by t-test BV/TV in OVX mice with ESRRA deletion differed from OVX control mice, the statistical analysis did not test if the response to OVX differed between the two genotypes. A two-way ANOVA to evaluate the independent effects of genotype and treatment (OVX) and possible genotype x treatment interaction would be more informative and rigorous.

Experimental Design and Interpretation of Results

10. Figure 1 is titled: “*Adipocyte-specific ESRRA ablation augments bone formation and inhibits MAT expansion in DIO osteopenia mice.*”

The following should be considered:

- a. The authors did not include mice fed a control diet and therefore have no basis for concluding that the mice fed the high fat diet were osteopenic. The skeletal influence of diet-induced obesity in mice is highly variable and may depend more on the diet than obesity (<https://doi.org/10.1016/j.bone.2023.116888>). However, in the absence of comorbidities, bone mass is generally positively associated with body mass and leptin levels in rodents and humans (doi: [10.1002/jbmr.3648](https://doi.org/10.1002/jbmr.3648)).
 - b. The K_D for leptin binding to leptin receptor is reported to be 0.1- 0.2 nM (doi:10.1371/journal.pone.0094843, DOI: [10.1016/j.ab.2004.01.022](https://doi.org/10.1016/j.ab.2004.01.022)). Thus, the circulating level of a WT mouse fed a normal diet is well above the K_D for leptin binding to its receptor. The high circulating levels of leptin in both groups in the present study would render the reported changes in plasma leptin levels Fig 1H unlikely to influence bone parameters. This is because the lower leptin value in the ESRRA-ablated mice (~15 ng/ml) remains 50-100 times the leptin/leptin receptor K_D . The levels would have to drop below 1.6 ng/ml (5-10 times K_D to have a meaningful effect), and this effect would likely be negative. This is supported by dose-response studies performed in *ob/ob* mice showing that the apparent EC_{50} for leptin actions on peripheral target tissues (e.g., bone and growth plate) occur at leptin levels of <0.5 ng/ml and for central target tissues (e.g., hypothalamus) of ~2 ng/ml (doi: [10.1530/JOE-16-0484](https://doi.org/10.1530/JOE-16-0484)). The absence of an effect is also supported by the finding that increasing leptin levels in the hypothalamus in normal rats using rAAV-leptin gene therapy reduces body weight and leptin levels, and reduces bone marrow adipose tissue without influencing bone metabolism (doi: [10.1530/JOE-15-0280](https://doi.org/10.1530/JOE-15-0280); doi: [10.3390/ijms22136789](https://doi.org/10.3390/ijms22136789)).
11. Figure 2 is titled: “*ESSRA deficiency in adipocytes favors bone formation and attenuates marrow adiposity in OVX-induced osteoporosis.*” The reviewer has several concerns regarding the design, reporting and interpretation of the experiment shown here.
- a. As mentioned, the statistical analysis lacks rigor. The authors did not perform analyses to support their claims that OVX resulted in bone loss that was attenuated by ESRRA. To support this claim, it is necessary to demonstrate a significant interaction where ESRRA deficiency influences the skeletal response to OVX.
 - b. Panels K and L should report data for intact mice, not just OVX mice.
 - c. A manuscript titled: “*Failure to generate bone marrow adipocytes does not protect mice from ovariectomy-induced osteopenia*” reported that OVX resulted in a comparable magnitude of bone loss in kit receptor-deficient *W/W^v* mice who have no bone marrow adipose tissue as in WT mice (DOI: [10.1016/j.bone.2012.11.034](https://doi.org/10.1016/j.bone.2012.11.034)). Whereas bone marrow adipose tissue increased in WT mice, the increase had no detrimental impact on bone formation or magnitude of bone loss. How do the authors reconcile their interpretation of the data presented in the manuscript with these earlier findings?
12. Figure 8 is titled: “*Pharmacological inhibition of ESRRA protects DIO mice from bone loss and MAT expansion.*” The results of this study are used to support “*a therapeutic approach via targeting adipocyte ESRRA to preserve bone formation especially in detrimental adipocyte-rich*

bone milieu.” In contrast to an earlier study (Figure 1) the authors include a control group fed a normal mouse diet. However, the BV/TV in the B6 mice (Figure 8) fed the DIO diet is drastically different from the control *Esrra*-expressing mice fed the same DIO diet in Figure 1. Does this mean that their genetic manipulation (floxing) influences the skeletal phenotype?

In interpreting their results, the authors should discuss the following:

- a) The 60% fat DIO diet used in this study is effective in inducing weight gain but does not recapitulate a human diet and as mentioned earlier most studies report positive effects of weight on bone in humans.
- b) Increased SPP1 may not be desirable as an approach to reduce marrow adipose tissue in that it is associated with increased insulin resistance, oxidative stress, inflammation and poorer cancer survival.

The following are representative publications: DOI:10.1038/s41598-017-02848-0; DOI: 10.1210/en.2007-1312; DOI:10.1371/journal.pone.0013959; DOI: [10.2337/db09-0404](https://doi.org/10.2337/db09-0404); DOI:org/10.1002/jso.25078.

REVIEWER COMMENTS

Reviewer #2 (Remarks to the Author):

The manuscript by Huang et al. reports the effect of ESRRRA adipocyte knock down on osteoblast differentiation and bone density. The authors use *in vivo*, *ex vivo* and *in vitro* methods to document the effect of the receptor. ESRRRA is inactivated using a genetic approach (adipoQ-Cre mediated knock down) as well as a pharmacological approach (using the C29 ESRRRA inhibitor). Strikingly, the main conclusion of the authors is that ESRRRA mainly act in adipocytes to regulate the expression of osteopontin (negatively) and leptin (positively) that both act in a paracrine manner on pre-osteoblasts and reduce their differentiation. The paper is well written, the results are very interesting and for the most parts, adequately controlled. I have a couple of question that could be useful to consider.

- The Karsenty lab showed (Ducy et al. Cell 2000) that Leptin inhibits bone formation. These authors also showed that there is no leptin signaling in osteoblast, pointing out to a centralized signaling. These papers and further developments on the thematic should be commented by the authors.

- The absence of ESRRRA, either a complete KO (Teyssier et al., PlosONE 2009) or a conditional KO in maturing osteoblasts (Gallet et al., PlosONE 2013), has been shown to result in increased osteoblast differentiation and activity during animal ageing or upon ovariectomy. The phenotype is very similar to what reported in the current manuscript (although the effect of ageing without any other challenge such as ovx in the present ms is unclear to me). These published papers should be commented (at least in the Discussion), in particular, the fact that OB-specific KO appear to result in a bone phenotype, which seems contradictory with the current manuscript conclusions.

- The authors show that ESRRRA is appropriately deleted in adipocytes. They should show that no depletion occurs in osteoblasts (precursors or mature) to reinforce the hypothesis of adipocyte-mediated effect on bone.

- Did the authors detect ESRRRA binding on the *Spp1* promoter (ChIP assays)? Does this binding vary according to E2 status? How can the authors distinguish binding on region 1 or 2 (which are 50bp apart) by ChIP-qPCR? Binding on region 1 and 2 should be detected with the same efficiency by qPCR. What is the y-scale of the ChIP data (relative to IgG detection? expressed as percent input?). Same remarks for ChIP on leptin promoter Figure 6.

- ESRRRA activates *SPP1* expression in osteoblasts in opposite to what is here shown in adipocytes. Is this a cell-specific difference? Effect of ESRRRA on *SPP1* expression in osteoblasts should be used as a control.

- Authors state that *SPP1* exert a proangiogenic effect, but what about VEGF which has been shown as an ESRRRA target (see Zou et al., J Pathol 2014). Is VEGF expression dysregulated in ESRRRA KO animals?

- On fig6 the authors show that ESRRRA requires PGC1alpha to stimulate Leptin promoter. The effect of ESRRRA on the *SPP1* promoter should be studied with and without PGC1a, it could indeed be possible that ESRRRA functions as a repressor only in the absence of PGC1a. Also what is the effect of ESRRRA on the leptin promoter without PGC1a?

- Leptin expression is reduced in the adipocyte-specific ESRRRA KO *in vivo*. Could this be due to altered levels of food intake in KO versus wild type animals (and thus not necessarily due to a direct effect of ESRRRA on leptin promoter)?

- The authors focus the second part of the ms on the effect of ESRRRA on Leptin. In FigS8, they show that ESRRRA reduces the expression of AdipoQ and PPARg in gWAT as potently as the expression of leptin. AdipoQ and PPARg are important for WAT differentiation and functions, thus why did the authors focus solely on leptin? A sentence should be added in the text to explain this motivated choice. In this respect, no change in PPARg CEBPa and FABP4 were detected in gWAT on Fig4D (wt versus ESRRRA KO, I assume?), whereas reduced expression is detected in BMSC-gWATCM induced (Fig7C). Can the authors explain?

Minor comments :

- Verify the spelling of LEPTIN everywhere (some occurrence of LEPIN or LEPITN)

- Define FFA on line 67

- FigS1: define gWAT, iWAt and mWAT in Suppl Fig Legend

Reviewer #3 (Remarks to the Author):

Huang et al. investigated the role of ERRa in adipocyte in DIO and OVX induced bone loss models using ERRa Adipoq-cre mice. Adipocyte ERRa deficiency protected mice from DIO or OVX-induced bone loss. ERRa is shown to oppositely regulate the expression of LEPTIN and *SPP1*, affecting MAT expansion and bone formation. This study uncovers the mechanisms by which ERRa in adipocytes contributes to bone formation and angiogenesis. In addition, this manuscript sheds light into the mechanisms by which MAT expansion is regulated and the effect of the fat-bone axis on pathological bone diseases.

Major points:

1. Figure 1. Fig. 1J. The authors should show normal diet controls. Fig. 1P. The visualization of ERRa in bone sections by immunofluorescence is necessary to support the effect of ERRa on MATs. In addition, *SPP1* signals in ERRa-deficient bone

increased, implicating the indirect effect of increased SPP1 from ERRa-deficient adipocytes on SPP1 expression of other cells. *in situ* hybridization or RNA scope of *Spp1* mRNA can clarify the pattern of *Spp1* in BMSCs. Fig. 1H. BMAs are actively secreting cytokines and adipokines. Metabolic inflammation in adipose tissue in obese mice and human subjects has been observed. Moreover, OPN plays an important role in inflammatory disorders. The authors should analyze if the secretion of inflammatory factors differs between control and ERRa-deficient mice.

2. Figure 3. The authors should provide more thorough analysis of RNA seq. It remains unclear how many genes are differentially regulated by ERRa deficiency. MATs express both BAT and WAT-marker genes. The authors should show the expression of BAT-marker genes to determine any phenotypic changes in MATs by ERRa. Although WAT has few mitochondria, mitochondrial function is important for the differentiation of adipocytes (especially in obese states). ERRa plays a crucial role in regulating mitochondrial function. It would be interesting to know whether ERRa regulates genes related to oxidative phosphorylation or other metabolic pathways in adipocytes.

3. Figure 4. Fig. 4F: SPP1 is diminished by estrogen deficiency in control mice. The authors should show the time-course expression of SPP1 and Leptin in both control and ERRa-deficient WAT in the same gel. Determining the level of ERRa will be useful in elucidating the intercorrelation between ERRa and Oestrogen during osteoporosis induced bone loss. Fig. 4L: The link between OPN and ERRa has been established. It has been shown that ERRa binds to the promoter of human OPN and increases the expression of OPN (PMID:286382130). The authors should discuss the tissue-specific regulation of OPN by ERRa in the discussion sections.

4. Figure 7. KO BMAds CM enhanced osteogenesis or decreased adipogenesis. However, in Figure 3C, there is no difference in osteogenesis or adipogenesis between WT and KO BMSCs, suggesting additional defects in the response to SPP1 or LEPTIN in ERRa KO BMSCs. The authors should discuss about this discrepancy.

Minor points

1. The authors should show the expression of ERRa in Figures 4J and K.
2. Figure S6 supports Figure 4L. The authors should move Figure S6 into main figure 4.

Response to comments from Reviewer #1:

General Comments

Huang et al. provide evidence that deletion/down regulation of *Esrra* expression influences bone metabolism, particularly in mice subjected to ovariectomy (OVX) or fed a high fat diet (60% energy from fat) to achieve diet-induced obesity (DIO). While *Esrra* deletion/depletion appears to have positive effects on cancellous bone volume fraction and bone formation and a negative effect on bone marrow adipose tissue in OVX'd and DIO mice, the putative mechanisms (redirection of differentiation from adipocytes to osteoblasts due to lower levels of leptin and increased *Spp1* expression in adipocytes) are primarily based on associations and not demonstration of causation. Additionally, there is a wealth of evidence, not considered by the authors, that does not support their interpretations of the results.

Response: We appreciate the constructive comments. We are sorry there were some vague descriptions that we have corrected and updated according to the reviewer's suggestions. Instead of involving redirection of differentiation from adipocytes to osteoblasts, we proposed the underlying mechanism that secreted factors leptin and *Spp1* from adipocytes are directly regulated by adipocyte *ESRRA*; enhanced *SPP1* and decreased leptin secretion could dictate bone marrow stromal stem cell fate commitment toward osteoblasts. Our findings are quite evidenced by *in vivo*, *ex vivo* and *in vitro* studies. We are grateful that the reviewer provides the following references and suggestions, helping us improve this manuscript.

Specific Comments

Abstract

1. Nomenclature: It is not clear why the authors are capitalizing the letters in leptin and adiponectin.

Response: Thank you for this kind reminder. We have corrected the gene and protein nomenclature according to guidelines (HGNC: <https://www.genenames.org/about/guidelines>; MGI: <https://www.informatics.jax.org/mgihome/nomen/gene.shtml>) throughout the manuscript. For example, the guidelines for the symbols of mouse genes and proteins (*Lep* and LEP; *Esrra* and *ESRRA*) and human genes and proteins (*LEP* and LEP; *ESRRA* and *ESRRA*) differ in their capitalization and italicization schemes. And we corrected the full name such as "Leptin" or "leptin" in the sentence.

2. Abstract line 32: “Ectopic bone marrow adipocytes.” The term ectopic generally refers an abnormal place or position. It is not clear why the authors believe bone marrow adipose tissue is abnormal.

Response: Thanks for this comment. We checked the Merriam-Webster's Dictionary about the term “ectopic” as “occurring in an abnormal position or in an unusual manner or form”. We agree with the reviewer that the medical definition of “ectopic” usually refers an abnormal place or position. We changed the word “ectopic” to “excessive” in the Abstract and throughout the manuscript.

3. Abstract lines 32-33: “... (BMAds) accumulation occurring under diverse pathophysiological conditions leads to bone deterioration.” While the authors may hypothesize that this is the case, the reviewer is unaware of any evidence that causally links bone marrow adipocytes to bone deterioration. Indeed, the very high levels of adipose tissue in caudal vertebra of mice (compared to DIO associated increases in femur, does not lead to bone deterioration. Please revise.

Response: As suggested, we updated the sentence as “Excessive bone marrow adipocytes (BMAds) accumulation often occurs under diverse pathophysiological conditions associated with bone deterioration.”

4. Abstract lines 34-35: “Here, we show that adipocyte-specific ESRRA deficiency rescues osteogenesis and vascular formation in adipocyte-rich bone marrow due to estrogen deficiency or obesity.” While the authors present evidence that ESRRA deficiency increases bone formation and vascular formation, this statement overinterprets the findings because associations, not causality are reported.

Response: We revised the sentence as “Here, we show that adipocyte-specific ESRRA deficiency preserves osteogenesis and vascular formation in adipocyte-rich bone marrow upon estrogen deficiency or obesity.”

5. Abstract lines 35-40: “Mechanistically, adipocyte ESRRA interferes with E2/ESR1 signaling resulting in transcriptional repression of secreted phosphoprotein 1 (Spp1); and positively modulates Leptin expression by binding to its promoter. ESRRA abrogation results in enhanced SPP1 and decreased LEPTIN secretion from both visceral adipocytes and BMAds, concertedly dictating bone marrow stromal stem cell fate commitment and restoring type H vessel formation, constituting a feed-forward loop for bone formation.” The authors’ data provide limited support for these assertions. Also, the authors have not acknowledged or addressed literature

that does not support the proposed mechanism. Briefly, most recent literature supports the following:

Response: For Abstract lines 36-40, our conclusion is based on the evidences by ChIP assays, luciferase assays, WB and qPCR analysis at the molecular levels *in vitro* and *ex vivo* in cell line and primary cells isolated from conditional ESRRRA knockout mice and controls. In the revised manuscript, we provided more evidences to address the direct regulation of ESRRRA on *Spp1* and *leptin* (**Fig.4** and **Fig.6**).

a) While there is often a reciprocal relationship between osteoblasts and bone marrow adipocytes, there are many exceptions, and osteoblasts and bone marrow adipocytes can be independently regulated by factors such as genetic manipulation, drugs, hormones, and housing temperature. It is notable that the authors are performing studies in growing OVX'd mice, in which cancellous bone loss occurs because of increased osteoclast-mediated bone resorption, not reduced bone formation. Despite increases in white adipose tissue (WAT) and bone marrow adipose depots, OVX results in larger bones in growing rodents due to increased radial bone growth (osteoblasts) and longitudinal bone growth (growth plate chondrocyte hypertrophy and proliferation). Excess marrow adipose tissue did not influence the magnitude of cancellous bone loss induced by skeletal disuse in ob/ob mice whereas absence of adipose tissue in W/W^v exaggerated bone loss. Absence of bone marrow adipose tissue in W/W^v mice had no effect on cancellous bone loss following OVX.

The following are representative publications: (DOI: 10.1016/0378-5122(96)01015-8 ; DOI: 10.1210/edrv-15-3-275; DOI: 10.1152/ajpendo.00646.2011; DOI: 10.1172/JCI6730). DOI: 10.1038/s41598-019-45587-0

Response: C57BL/6J mice reach peak bone mass at the age of about 8-week-old. A paper published in J Bone Miner Res 2007 (PMID: 17488199) assessed age-related changes in bone architecture in male and female C57BL/6J mice. Their findings clearly demonstrated that trabecular bone volume (BV/TV) was greatest at 6-8 week of age and declined steadily thereafter, particularly in the metaphyseal region of long bones, which is more pronounced at the femoral metaphysis than in the vertebrae, and is greater in females than males. In the paper provided by the reviewer in Question 10a (doi.org/10.1016/j.bone.2023.116888) also mentioned that “The present study was initiated when the mice were 8 weeks old. This age corresponds to peak cancellous bone volume fraction in the distal femur metaphysis in male B6 mice housed at room temperature.”

In the present OVX study, we initiated OVX operation in female C57BL/6J mice at the age of 10-wk-old for a period of 8 weeks. Female C57BL/6J mice at the age of 10-wk-old already experienced a rapid decrease in trabecular bone mass. Hence, we do not perform our study in bone growing mice. It has been suggested that bone loss after ovariectomy or during aging is largely associated with decreases in bone formation rate.

Consistent with other reports ((Sci Adv 2020 PMID: 33208358; JCI Insight. 2017 PMID: 29202453; J Bone Miner Res. 2017 PMID: 27391172), a significant reduction of bone formation rate induced by OVX was observed in the femur samples in present study.

These two review papers published at 1994 and 1996 referred by the reviewer (DOI: 10.1016/0378-5122(96)01015-8; DOI: 10.1210/edrv-15-3-275) both emphasized that estrogen affects the skeleton in all species studied especially bone resorption. They also mentioned that estrogen might have direct or indirect effect on osteoblasts which were uncertain. Recently, the positive effects of estrogen on osteogenesis and bone formation have been demonstrated (Cell Metabolism 2019 PMID: 30661929; Bone Research 2022, PMID: 34992221).

The paper referred by the reviewer (DOI: 10.1152/ajpendo.00646.2011) focused on chondrocyte proliferation and apoptosis in the growth plate which is incomparable with the present study.

The paper referred by the reviewer (DOI: 10.1172/JCI6730) showed that ER β is essential for the pubertal feminization of the cortical bone in female mice but is not required for the protective effect of estrogens on trabecular BMD. However, our data were based on adult mice under the pathological conditions including 8 weeks OVX and a long-term high-fat diet feeding.

In the paper (DOI: 10.1038/s41598-019-45587-0) the authors performed studies on obese leptin-deficient *ob/ob* mice. *ob/ob* mice is a genetic model with whole-body deficiency of leptin leading to a serious and complex metabolism disorder. In the present study, we put the mice on a high-fat diet which is incomparable with *ob/ob* mice. Moreover, we generated the conditional knockout mice which showed no significant changes in body weight, white adipose tissue mass, glucose tolerance test and other metabolic traits compared to control mice under normal condition (**revised Fig 1-2, Supplementary Fig. 1-3**).

b) Most studies performed during the last 2 decades report that plasma leptin has positive effects on bone formation by increasing the differentiation and activity of osteoblasts. Leptin treatment consistently reduces WAT mass and has been shown to decrease bone marrow adipose tissue. This evidence includes (1) leptin stimulation of osteoblast differentiation and matrix mineralization of bone marrow stromal cells in vitro, (2) evaluation of the skeletal phenotype of leptin-deficient *ob/ob* and leptin receptor-deficient *db/db* mice, (3) evidence that the osteopenic and osteopetrotic skeletal phenotypes of these mice are not due to obesity, (4) dose response studies demonstrating that leptin increases bone formation at levels having little or no effect on energy metabolism, and (5) adoptive transfer of leptin receptor-

deficient cells from db/db mice into WT mice recapitulates the low bone formation phenotype of db/db mice without influencing energy metabolism.

The following are representative publications: (DOI: 10.1016/j.regpep.2006.04.013 ; DOI: 10.1097/MCO.0b013e3282f795cf; DOI: 10.1016/j.bone.2003.11.020; DOI: 10.1002/jbmr.36.7; DOI:10.1530/JOE-16-0484; <https://doi.org/10.1210/endo.140.4.6637>; DOI: 10.1530/JOE-14-0224; DOI: 10.1002/jbmr.1812); DOI: 10.1002/jbmr.406; DOI: 10.3389/fendo.2022.959743).

Response: In contrast to the findings of these papers referred by the reviewer, Karsenty group reported that leptin inhibits bone formation through a hypothalamic relay in *ob/ob* mice. In the last two decades, there are debates in the effects of leptin on bone since the original concept was raised by the Karsenty group in 2000 (Cell 2000, PMID: 10660043). A review paper published in 2013 (J Bone Miner Res. 2013, PMID: 23188700) have made a comprehensive commentary on the opposite observations in *ob/ob* mice of leptin effects from those of Karsenty group and Turner group.

Most references in 5b) referred by the reviewer and Karsenty group performed the studies in *ob/ob* or *db/db* mice as a result of loss-of-function mutations in the genes for leptin and the leptin receptor which are rare in humans. In our current study, we employed a long-term high-fat diet strategy to establish an obese C57BL//6 mice model with increased leptin level, but not leptin deficiency as in *ob/ob* mice. The diet-induced obese (DIO) rodent model is well-described to mimic hyperleptinemia in human obesity, although the discrepancy cannot be avoided between the two species. Leptin/LepR signaling in the adult bone marrow might be more responsive for BMSCs fate dictation in the context of hyperleptinemia of DIO model (Cell stem cell 2016, PMID: 27053299; Cell reports 2019 PMID: 31091445).

Different from the paper referred by the reviewer (DOI: 10.1210/endo.140.4.6637) by using a marrow stromal cell line for lepin treatment, we performed conditional medium treatment in primary BMSCs. Furthermore, we cultured BMSCs in conditioned medium supplemented with osteogenic/adipogenic mixed induction medium to verify the effects of leptin, SPP1 and potential secreted factors from adipocytes.

As suggested by the reviewer, we updated the Discussion for this point (**Page 14 line 414-424**).

“Leptin was originally identified as a circulating hormone involved in feeding behavior and energy homeostasis. The role of leptin in bone formation is heavily debated as evidences for central nervous system and peripheral pathways have both been presented.^{46,47} However, analysis of bone phenotypes in these studies are complicated by the fact observed in genetic mutant *ob/ob* and *db/db* mice with global defects of *Lep* and *LepR*, respectively. *ob/ob* and *db/db* mice exhibit severe metabolic disorders,

including not only obesity and diabetes but also muscle hypoplasia, hypogonadism and hypercortisolism, accompanied with hypometabolic and hypothermic states, risk factors that can independently influence bone formation. Different from *ob/ob* and *db/db* mice with the abolished leptin/LepR signaling, DIO mice on 60% kcal fat develop hyperleptinemia, irrespective of genotype, which are similar to most obese humans. Leptin has been observed to act on LepR⁺ BMSCs mediating diet-induced fat accumulation in adult bone marrow at the expense of osteogenesis⁷.”

6. Abstract lines 42-43: “Thus, our findings highlight a therapeutic approach via targeting adipocyte ESRRA to preserve bone formation especially in detrimental adipocyte-rich bone milieu.” There is evidence that inhibiting the expression of ESRRA will not necessarily increase bone formation. For example, ethanol suppresses *Esrra* expression. This did not lead to increased *Spp1* expression and bone formation, nor decreased accrual of bone marrow adipose tissue. Instead, the opposite was found (doi: 10.1111/acer.14185). Additionally, increasing *Spp1* expression is known to lead to undesirable side effects (see last comment for details).

Response: Several studies reported the roles of ESRRA in bone formation and resorption. We updated the Discussion and added the related references (**Page 16 line 478-486**). Please also refer the detailed response for Question 2 of Reviewer#2.

The paper (doi:10.1111/acer.14185) referred by the reviewer showed that ethanol treatment decreased the mRNA level of *Esrra* which was measured in femur not in adipocyte. In current study, we focus on SPP1 expression regulated by ESRRA in adipocytes but not in either osteoblasts or osteocytes. It is well-described that ethanol affects other signal pathways involved in bone remodeling and necrosis. In the revised manuscript, our findings indicated that ESRRA as a transcriptional repressor or activator in a cell context-dependent manner. We also have discussed this point in the revised Discussion (**Page 16 line 465-476**).

Introduction

7. Adipocytes express kit ligand (stem cell factor), which, as noted by the authors, plays an important role in regulation in hematopoietic cell lineage differentiation and function. However, there is also strong evidence that kit receptor/kit ligand signaling is important for bone marrow adipocyte differentiation. The critical findings relevant to the manuscript under review but not addressed by the authors are:

a) Mice deficient in kit signaling have abnormalities in fat metabolism, including decreased fat accrual in bone marrow. Despite lack of bone marrow adipose tissue,

kit receptor-deficient mice do not differ from WT mice in OVX-induced bone loss and have an exaggerated negative response to the hindlimb unloading model for skeletal disuse. b) Pharmacological blockade of kit signaling with the receptor tyrosine kinase antagonist gleevec decreases bone marrow adipose tissue in rats, but also decreases bone formation.

c) Compared to WT mice, bone marrow from kit receptor-deficient W/W^v mice has a drastically reduced capacity to support adipocyte differentiation in vitro.

d) Kit receptor-deficient W/W^v and kit ligand-deficient Sl/Sld mice have no adipocytes in bone marrow of their long bones or lumbar vertebra.

e) Sl/Sld mice, despite having no bone marrow adipocytes, have reduced bone formation in vivo (not increased as predicted by the authors) and have a reduced capacity to form osteoblasts from bone marrow in vitro.

f) Sl/Sld mice are unable to produce membrane kit ligand but produce soluble kit ligand. Administration of soluble kit ligand to S/Sl mice does not rescue their inability to form bone marrow adipocytes. This finding suggests that cells with kit receptor must directly associate with cells with membrane kit ligand to form bone marrow adipocytes.

g) Kit receptor-deficient W/W^v mice have reduced expression of Esrra and no bone marrow adipocytes. However, bone formation is not universally increased in these mice.

h) Adoptive transfer of purified hematopoietic stem cells into kit receptor-deficient W/W^v mice normalizes kit receptor expression in bone marrow. However, Esrra expression remains low in these mice, and they remain unable to form bone marrow adipocytes.

The following are representative publications: (DOI: 10.1371/journal.pone.0131192;

DOI: 10.1007/s11914-018-0422-3; DOI.org/10.1002/ajh.2830080404;

DOI:10.1667/RR15164.1; DOI: 10.1016/j.bone.2012.11.034).

Response: There might be a misunderstanding. In this manuscript, we NEVER mean that the mice with no bone marrow adipocytes will have an increased bone mass. On the contrary, normal marrow adipocytes are important for bone microenvironment and bone homeostasis as mentioned in the manuscript.

The paper referred by the reviewer (DOI: 10.1016/j.bone.2012.11.034) showed that global deficiency of kit receptor-deficient W/W^v mice displayed no marrow adipocytes. Indeed, these mice with no marrow adipocytes showed lower cancellous BV/TV compared to WT mice.

The tyrosine kinase (TK) inhibitor imatinib (Gleevec) is not a specific antagonist of c-KIT. Actually, it targets BCR-ABL1 and exerts off-target effects including stem cell growth factor receptor (c-KIT), platelet-derived growth factor receptors (PDGF-R), and colony-stimulating factor-1 receptor (c-fms) that are involved in bone metabolism (DOI: 10.1371/journal.pone.0131192 referred by reviewer). We think the potential influences of imatinib on bone metabolism exceed the scope of this manuscript.

We agree with the reviewer that c-kit receptor and m-kit ligand are both important. Loss of function mutations of them result in anemia, mast cell deficiency, altered body composition, and skeletal abnormalities (DOI: 10.1007/s11914-018-0422-3 referred by the reviewer). This indicates that there are complex problems in these mice in addition to no bone marrow adipocytes. Different from the mice models (Kit receptor-deficient W/W^v and kit ligand-deficient Sl/Sld mice) accompanied with such symptoms, our findings clearly showed that adipocyte-specific ESRRA ablation in both male and female mice have no significant changes in metabolic and bone phenotypes under normal physiological condition. As we mentioned in the Introduction, ESRRA is a key regulator responding to stress-induced challenges, such as fasting, calorie restriction, cold exposure or overnutrition. We observed that adipocyte-ESRRA deficiency leads to significant inhibition of MAT expansion induced by DIO and OVX.

Furthermore, some studies refereed by reviewers were executed in young growing rodents. For examples, 4-week-old male Wistar rats were used in the paper DOI: 10.1371/journal.pone.0131192 and 4-week-old female mice were used in the paper DOI: 10.1016/j.bone.2012.11.034. Here, we performed the study in adult mice post OVX surgery or fed with a long-term high-fat diet.

Collectively, we updated the Discussion as follows:

“BMADs and their precursors LepR⁺ stromal cells in adult bone marrow, are the major sources of NGF, KitL and VEGF contributing to sustain nerves and promote haematopoietic and vascular regeneration after myeloablation^{17,69}. These emerging evidences suggest marrow adipocytes homeostasis are essential, especially in impaired bone marrow environment after irradiation or chemotherapy. Furthermore, the absence of marrow adipocytes induced by global reduction in c-kit receptor function and m-kit ligand function in mice result in exacerbated bone loss, demonstrating that normal marrow adipocytes and hematopoietic lineage cells are required for maintaining bone homeostasis^{70,71,72}. It needs to be determined whether these factors play essential roles in the expanded MAT induced by different pathological conditions. On the other hand, MAT volume varies between age, gender and species. Specifically, fold changes of MAT in rodents models due to metabolic diseases such as obesity generally exceeds that observed in humans⁹. Excessive MAT correlation with bone, vascularization, hematopoiesis or metabolism in humans in these pathophysiological conditions need to be further investigated.” (page 17 line 491-502)

Methods

8. Please report the temperature that the mice were housed at. ESRRA-null mice are cold intolerant and unable to maintain their core body temperature due to impaired adaptive thermogenesis (DOI: 10.1073/pnas.0607696104; DOI: 10.1016/j.isci.2018.03.005). In contrast to humans and larger rodents (e.g., rats) who are homeotherms, mice are facultative heterotherms. This is relevant because increased adaptive thermogenesis induced by housing mice below their thermoneutral zone (~32°C) results in rapid cancellous bone loss from the femur (DOI: org/10.1007/s00198-016-3634-3; DOI: org/10.1002/ajpa.23684; DOI.org/10.1038/s41574-020-00424-7). Thus, it is plausible that the stimulatory effect of ESRRA deficiency on bone is due to their impaired thermogenesis. In other words, these mice do not experience reduced bone formation and increased BMAT associated with upregulated thermogenesis when housed at room temperature.

Response: This is an important suggestion. As suggested, we have added a sentence “All mice were maintained in a specific-pathogen-free facility at $24 \pm 2^\circ\text{C}$ with $60\% \pm 5\%$ humidity under 12 h light/dark cycles which was defined as normal conditions.” in the first paragraph of Methods (**Page 18 line 527-529**). Additionally, there were no differences in rectal temperature between *Esrra*^{AKO} and control mice when they were housed in the normal conditions (**revised Supplementary Fig. 1f**). We agree that this is a very interesting question, but it is outside of the scope of the current study. We demonstrated the effect of ESRRA on marrow adipocytes and bone formation in this study, we do not rule out the possibility that adaptive thermogenesis might also have influences on the bone remodeling under certain stress condition in current mice models. We added the following sentences in the revised Discussion: “Furthermore, ESRRA regulates numerous genes in different tissues such as intestine, kidney and liver that are involved in bone metabolism, of which the multifaceted roles on bone need to be clarified in a cell or tissue-specific context. Moreover, ESRRA also mediates adaptive thermogenesis of BAT^{67,68}. Whether ESRRA deficiency affects bone metabolism through regulating BAT or the beige-like characteristics of MAT especially in cold stress condition need to be further investigated.” (**Page 17 line 484-489**).

9. The description of the statistical analysis is vague but the methods seem deficient in rigor. It is not clear whether the assumptions underlying the statistical tests were assessed. The authors did not test whether OVX differed from Sham surgery, yet make statements in the test (e.g., bone loss) as if they had done so. While by t-test BV/TV in OVX mice with ESRRA deletion differed from OVX control mice, the statistical analysis did not test if the response to OVX differed between the two genotypes. A two-way ANOVA to evaluate the independent effects

of genotype and treatment (OVX) and possible genotype x treatment interaction would be more informative and rigorous.

Response: We are grateful to the reviewer for this important suggestion. We updated the statistical analysis throughout the manuscript as stated in the **revised Methods section (page 26 line 765-771)**: “Sample and animal distribution among groups were randomized. All data distribution was assumed to be normal, but this was not formally tested. Differences across more than two groups were analyzed with one-way ANOVA followed by Bonferroni’s post hoc tests or two-way ANOVA with Fisher’s LSD post hoc multiple comparisons tests, as described in the figure legends. Unpaired, two-tailed Student’s t-test was used for comparisons between two groups. Data are presented as mean \pm SD. P value <0.05 was considered statistically significant. GraphPad Prism 8.0 and SPSS Statistics 26.0 were applied for statistical analyses.” For OVX mice study, multiple group comparisons were conducted by two-way ANOVA with Fisher’s LSD post hoc multiple comparisons tests as shown in **revised Fig.2 and legend**.

Experimental Design and Interpretation of Results

10. Figure 1 is titled: “Adipocyte-specific ESRRA ablation augments bone formation and inhibits MAT expansion in DIO osteopenia mice.” The following should be considered:

a. The authors did not include mice fed a control diet and therefore have no basis for concluding that the mice fed the high fat diet were osteopenic. The skeletal influence of diet-induced obesity in mice is highly variable and may depend more on the diet than obesity (<https://doi.org/10.1016/j.bone.2023.116888>). However, in the absence of comorbidities, bone mass is generally positively associated with body mass and leptin levels in rodents and humans (doi: 10.1002/jbmr.3648).

Response: Thank you for the important point. Following the reviewer’s suggestion, we included normal chow diet (NCD) groups, **reorganized Fig.1**, and updated the text, Methods especially statistical analysis. By using two-way ANOVA with Fisher’s LSD post hoc multiple comparisons tests as described in the **revised Fig.1 legend**, we determined a significant decrease in bone mass in male C57BL/6 mice fed with a high-fat diet for 16 weeks as shown in the **revised Fig.1f, g**. Our findings is consistent with a paper referred in the manuscript (J Bone Miner Res. 2018, PMID 29444341). In this paper, they fed the male C57BL/6 with the same diet as ours (60% kcal from fat, Research Diet #D12492) for 12 weeks and revealed that high-fat diet-induced obesity decreased trabecular and cortical bone mass significantly. They also demonstrated that BMAT expansion in response to HFD exerts a deleterious effect on the skeleton.

We agree with the reviewer that the skeletal influence of diet-induced obesity in mice is highly variable, depending on not only the diet composition but also mice strain, gender and age, as well as the feeding period. However, in the paper (doi: 10.1002/jbmr.3648), they focused on the effect of a high-fat diet on cortical bone in female and male LG,SM AI mice, that is “Large-by-Small advanced intercross (LG,SM AI) line” which was created from an initial cross of inbred strains chosen for large and small extremes of body size, respectively. Hence, their conclusion was about cortical bone traits in the LG, SM AI mice line fed with a high-fat diet which is incomparable with the current study.

We fed the male C57BL/6 mice with a high-fat diet (60% kcal from fat, Research Diet #D12492) for 16 weeks in **Fig.1** and 18 weeks in **Fig.8**. In the paper referred by the reviewer (doi.org/10.1016/j.bone.2023.116888), the authors fed the male C57BL/6 with a 46% kcal for 9 weeks. In this case, they also reported that mice consuming the high fat 'Western' diet exhibited a tendency for lower cancellous bone volume fraction and connectivity density, and had lower osteoblast-lined bone perimeter (an index of bone formation) and higher bone marrow adiposity than low fat controls. This result supported our findings and we have added this **reference (#28)** in the revised manuscript.

b. The K_D for leptin binding to leptin receptor is reported to be 0.1- 0.2 nM

(doi:10.1371/journal.pone.0094843, DOI: 10.1016/j.ab.2004.01.022). Thus, the circulating level of a WT mouse fed a normal diet is well above the K_D for leptin binding to its receptor. The high circulating levels of leptin in both groups in the present study would render the reported changes in plasma leptin levels Fig 1H unlikely to influence bone parameters. This is because the lower leptin value in the ESRRRA-ablated mice (~15 ng/ml) remains 50-100 times the leptin/leptin receptor K_D. The levels would have to drop below 1.6 ng/ml (5-10 times K_D to have a meaningful effect), and this effect would likely be negative. This is supported by dose-response studies performed in ob/ob mice showing that the apparent EC₅₀ for leptin actions on peripheral target tissues (e.g., bone and growth plate) occur at leptin levels of <0.5 ng/ml and for central target tissues (e.g., hypothalamus) of ~2 ng/ml (doi: 10.1530/JOE-16-0484). The absence of an effect is also supported by the finding that increasing leptin levels in the hypothalamus in normal rats using rAAV-leptin gene therapy reduces body weight and leptin levels, and reduces bone marrow adipose tissue without influencing bone metabolism

(doi: 10.1530/JOE-15-0280; doi: 10.3390/ijms22136789).

Response: We titled the **Fig 1** as “Adipocyte-specific ESRRRA ablation augments bone formation and inhibits MAT expansion in DIO osteopenia mice.” that was based on the

evidences from the conditional adipocyte knockout mice. The above information pointed by the reviewer is interesting but not relevant to our data.

The KD reported by these two papers were both analyzed *in vitro* for the binding of concentrated recombinant leptin receptor Fc (LepRec-Fc) chimera to recombinant leptin by either surface plasmon resonance or Bia-core analysis. None of these authors claimed that this kinetics data was reflective of the affinity of leptin binding to leptin receptor *in vivo*.

In the paper referred by the reviewer (doi: 10.1530/JOE-16-0484), the authors performed the subcutaneous leptin infusion for 12 days in *ob/ob* mice and found bone formation parameters increased at EC50 infusion of 7–17 ng/hour (cannot find the data about leptin levels of <0.05ng/ml in this paper). The other two papers performed hypothalamic leptin gene therapy by AAV that is not in the scope in current study.

11 Figure 2 is titled: “ESRRA deficiency in adipocytes favors bone formation and attenuates marrow adiposity in OVX-induced osteoporosis.” The reviewer has several concerns regarding the design, reporting and interpretation of the experiment shown here.

a. As mentioned, the statistical analysis lacks rigor. The authors did not perform analyses to support their claims that OVX resulted in bone loss that was attenuated by ESRRA. To support this claim, it is necessary to demonstrate a significant interaction where ESRRA deficiency influences the skeletal response to OVX.

Response: We are grateful to the reviewer for raising this concern. We have updated the statistical analysis. Please refer to the detailed response for Question 9. The result as shown in **revised Fig.2** clearly revealed that OVX operation resulted in bone loss that was attenuated by adipocyte-specific ESRRA ablation.

b. Panels K and L should report data for intact mice, not just OVX mice.

Response: As suggested, we included sham mice of two genotypes as shown in the **revised Fig. 2k, l**.

c. A manuscript titled: “Failure to generate bone marrow adipocytes does not protect mice from ovariectomy-induced osteopenia” reported that OVX resulted in a comparable magnitude of bone loss in kit receptor-deficient W/W^v mice who have no bone marrow adipose tissue as in WT mice (DOI: 10.1016/j.bone.2012.11.034). Whereas bone marrow adipose tissue increased in WT mice, the increase had no detrimental impact on bone formation or magnitude of bone loss. How do the

authors reconcile their interpretation of the data presented in the manuscript with these earlier findings?

Response: Please refer our response for Question 7c

12. Figure 8 is titled: “Pharmacological inhibition of ESRRA protects DIO mice from bone loss and MAT expansion.” The results of this study are used to support “a therapeutic approach via targeting adipocyte ESRRA to preserve bone formation especially in detrimental adipocyte-rich bone milieu.” In contrast to an earlier study (Figure 1) the authors include a control group fed a normal mouse diet. However, the BV/TV in the B6 mice (Figure 8) fed the DIO diet is drastically different from the control *Esrra*-expressing mice fed the same DIO diet in Figure 1. Does this mean that their genetic manipulation (floxing) influences the skeletal phenotype?

Response: Thanks very much to point out this discrepancy. We checked the original microCT data of **Fig.8** experiment and changed the threshold value in the step "Segmentation" with CTAn software to be consistent with **Fig.1** microCT analysis. Then we analyzed and recalculated the bone parameters as shown in the **revised Fig.8i**. The BV/TV in DIO-Veh mice in **Fig.8** is still lower than *Esrra*^{fl/fl} DIO mice. Since *Esrra*^{fl/fl} mice has been backcrossed to wild-type C57BL/6J for at least six generations in our lab, we think there might be two possibilities for this discrepancy: 1) The HFD feeding period is different in two experiments. In **Fig.8**, all mice at the age of 7-week-old were fed on a high-fat diet for 18 weeks. However, *Esrra*^{AKO} and *Esrra*^{fl/fl} mice at the age of 9-week-old were fed on a high-fat diet for 16 weeks. 2) All mice in **Fig.8** were administered vehicle which was composed of 10% vitamin E-TPGS, 20% PEG400, and 70% water, or C29 dissolved in Veh through oral gavage for 4 weeks. We addressed this point and changed “Control” mice group to “NCD + Veh” in the **revised Fig.8 and Methods. (Page19 line 543-552)**.

In interpreting their results, the authors should discuss the following:

a) The 60% fat DIO diet used in this study is effective in inducing weight gain but does not recapitulate a human diet and as mentioned earlier most studies report positive effects of weight on bone in humans.

Response: We agree with the reviewer that rodent model of obesity including DIO model, *ob/ob* mice are different from obese humans. Obesity leads to a controversial effects in humans. In present study, our findings are limited in mice models. We updated the following in the Discussion (**Page 17 line 497-502**): “It needs to be determined whether these factors play essential roles in the expanded MAT induced by different pathological conditions. On the other hand, MAT volume varies between age,

gender and species. Specifically, fold changes of MAT in rodents models due to metabolic diseases such as obesity generally exceeds that observed in humans⁹. Excessive MAT correlation with bone, vascularization, hematopoiesis or metabolism in humans in these pathophysiological conditions need to be further investigated.”

b) Increased SPP1 may not be desirable as an approach to reduce marrow adipose tissue in that it is associated with increased insulin resistance, oxidative stress, inflammation and poorer cancer survival.

The following are representative publications: DOI:10.1038/s41598-017-02848-0; DOI: 10.1210/en.2007-1312; DOI:10.1371/journal.pone.0013959; DOI: 10.2337/db09-0404; DOI:org/10.1002/jso.25078.

Response (also for Q6): As suggested, we updated this point in the Discussion (**Page 16 line 474-476**).

Response to comments from Reviewer #2:

The manuscript by Huang et al. reports the effect of ESRRA adipocyte knock down on osteoblast differentiation and bone density. The authors use *in vivo*, *ex vivo* and *in vitro* methods to document the effect of the receptor. ESRRA is inactivated using a genetic approach (adipoQ-Cre mediated knock down) as well as a pharmacological approach (using the C29 ESRRA inhibitor). Strikingly, the main conclusion of the authors is that ESRRA mainly act in adipocytes to regulate the expression of osteopontin (negatively) and leptin (positively) that both act in a paracrine manner on pre-osteoblasts and reduce their differentiation. The paper is well written, the results are very interesting and for the most parts, adequately controlled. I have a couple of question that could be useful to consider.

Response: We appreciate the positive feedback and constructive comments.

- The Karsenty lab showed (Ducy et al. Cell 2000) that Leptin inhibits bone formation. These authors also showed that there is no leptin signaling in osteoblast, pointing out to a centralized signaling. These papers and further developments on the thematics should be commented by the authors.

Response: This is an important point. The Karsenty lab (Ducy et al. Cell 2000) treated leptin in primary osteoblast cultures from calvaria. These primary osteoblasts are deficient of leptin receptor (Ob-Rb transcript). So they pointed out that there is no leptin signaling in osteoblasts. Actually, the effects of leptin on bone have been debated for two decades (JBMR 2013, PMID: 23188700). Recently, leptin receptor-expressing (LepR⁺) stromal cells are well accepted as common progenitors for osteoblasts and adipocytes in adult bone marrow. New evidences point out that Leptin/LepR signaling regulates adipogenesis and osteogenesis by BMSCs in adult bone marrow in response to diet and adiposity (Cell stem cells 2016, PMID: 27053299).

Please referred the detailed answer of Reviewer1 Q5b. We also copied below:

“Leptin was originally identified as a circulating hormone involved in feeding behavior and energy homeostasis. The role of leptin in bone formation is heavily debated as evidences for central nervous system and peripheral pathways have both been presented.^{46,47}. However, analysis of bone phenotypes in these studies are complicated by the fact observed in genetic mutant *ob/ob* and *db/db* mice with global defects of *Lep* and *LepR*, respectively. *ob/ob* and *db/db* mice exhibit severe metabolic disorders, including not only obesity and diabetes but also muscle hypoplasia, hypogonadism and hypercortisolism, accompanied with hypometabolic and hypothermic states, risk factors that can independently influence bone formation. Different from *ob/ob* and

db/db mice with the abolished leptin/LepR signaling, DIO mice on 60% kcal fat develop hyperleptinemia, irrespective of genotype, which are similar to most obese humans. Leptin has been observed to act on LepR⁺ BMSCs mediating diet-induced fat accumulation in adult bone marrow at the expense of osteogenesis⁷.” (Page 14 line 414-424)

- The absence of ESRRA, either a complete KO (Teyssier et al., PlosONE 2009) or a conditional KO in maturing osteoblasts (Gallet et al., PlosONE 2013), has been shown to result in increased osteoblast differentiation and activity during animal ageing or upon ovariectomy. The phenotype is very similar to what reported in the current manuscript (although the effect of ageing without any other challenge such as ovx in the present ms is unclear to me). These published papers should be commented (at least in the Discussion), in particular, the fact that OB-specific KO appear to result in a bone phenotype, which seems contradictory with the current manuscript conclusions.

Response: This is an excellent point. Actually, previously global or conditional ESRRA knockout mice showed the contradictory bone phenotypes (J Bone Miner Res. 2013, PMID: 23212690).

Teyssier *et al.* (Plos ONE 2009, PMID 19936213) reported that global ESRRA knockout (KO) male mice at the age of 14 weeks and 24 weeks showed no bone phenotypes. However, lower trabecular bone mass was observed in 14-week-old female KO mice compared to the same age female mice, whereas there were no significant difference between two genotypes of female mice at the age of 24-week-old. They also observed no changes in KO OVX mice compared to KO sham mice, while both KO-OVX mice and KO-sham mice displayed decreased bone mass compared to WT sham mice. However, they claimed that ESRRA negatively regulated osteoblast differentiation by *ex vivo* experiments.

Gallet *et al.* from the same lab (Plos ONE 2013, PMID 23359549) generated a conditional knockout mice Col1a-Cre; ERRa^{fl/fl} which inactivated ESRRA only in mature osteoblasts but not in mesenchymal stem cells or early committed osteoblast progenitors. They observed no changes of bone volume in cKO female mice at the age of 14 weeks and 24 weeks compared to the same age of control mice. However, they found cKO female mice were protected from OVX-induced bone loss. They isolated pre-osteoblasts from Col1a-Cre; ERRa^{fl/fl} mice calvaria and demonstrated that the absence of ESRRA promoted late osteoblast differentiation without impacting on early commitment and differentiation steps.

Furthermore, Delhon *et al.* (Endocrinology 2009, PMID 19608650) observed modest increase in femoral cancellous bone mineral density in whole-body ESRRA-deficient

female 4-month-old mice. Wei et al. (Cell Metabolism 2010 PMID 20519122; Cell metabolism 2016, PMID 26777690) reported that global ESRRA knockout male mice displayed high bone volume due to decreased bone resorption.

On the contrary, Rajalin et al. (Biochemical and Biophysical Research Communications 2010, PMID 20417614) isolated BMSCs from global ESRRA knockout mice and observed that ESRRA-deficiency inhibited osteoblastic differentiation of BMSCs. They claimed that ESRRA positively regulated BMSCs osteogenesis. Consistently, Auld et al (Journal of Molecular Endocrinology 2012, PMID 22333182) demonstrated that ESRRA silence inhibited and over-expression promoted osteogenesis in hMSCs via Wnt/b-catenin signaling. Previously, we also reported that ESRRA promotes hMSCs osteogenesis in vitro (Stem cells 2017, PMID: 27501743).

Collectively, we followed the Reviewer's suggestion and updated this part in the Discussion (**Page 16 line 478-486**):

“The direct regulatory role of ESRRA on bone cells remains obscure. Global ESRRA knockout mice displayed no changes in bone mass in male mice but a protective effect on aged bone in female mice^{60, 61, 62}; whereas other literature reported high bone mass in knockout male mice due to suppressed osteoclastogenesis and bone resorption^{63,64}. Moreover, several studies support a positive regulatory role for ESRRA in osteoblast differentiation *in vitro* by using human or murine MSCs^{65,66}. On the contrary; the absence of ESRRA in mice calvaria cells was demonstrated to promote osteoblast differentiation^{60,61}. These discrepancies might be due to differences in age, gender or mice strain. Furthermore, ESRRA regulates numerous genes in different tissues such as intestine, kidney and liver that are involved in bone metabolism, of which the multifaceted roles on bone need to be clarified in a cell or tissue-specific context. ”

- The authors show that ESRRA is appropriately deleted in adipocytes. They should show that no depletion occurs in osteoblasts (precursors or mature) to reinforce the hypothesis of adipocyte-mediated effect on bone.

Response: This is a good point. We followed the reviewer's suggestion. As shown in **Fig. 3a**, ESRRA was deficiency during adipogenic differentiation of *Esrra*^{AKO}-derived BMSCs. As expected, we observed no changes of ESRRA and SPP1 expression between *Esrra*^{fl/fl} and *Esrra*^{AKO}-derived BMSCs upon osteogenic differentiation at 0, 4, 8, 12 and 18 days, indicating that adipocyte-specific ESRRA knockout by using AdipoQ-Cre had no influence on ESRRA expression in osteoblasts. (**Supplementary Fig. 4a**). We noticed that ESRRA and SPP1 expression were induced during osteogenic differentiation of either *Esrra*^{fl/fl} or *Esrra*^{AKO} BMSCs (**Supplementary Fig. 4a**), which indicated that ESRRA might play a positive regulation on SPP1 expression

during BMSCs osteogenesis.

- Did the authors detect ESRRA binding on the *Spp1* promoter (ChIP assays)? Does this binding vary according to E2 status? How can the authors distinguish binding on region 1 or 2 (which are 50bp apart) by ChIP-qPCR? Binding on region 1 and 2 should be detected with the same efficiency by qPCR. What is the y-scale of the ChIP data (relative to IgG detection? expressed as percent input?). Same remarks for ChIP on leptin promoter Figure 6.

Response: We thank the reviewer for asking such important questions. As suggested, we performed ChIP assays for detection of ESRRA binding on the *Spp1* promoter. As shown in **revised Fig. 4j** (data copied below), ESRRA could bind to the *Spp1* promoter which was further enhanced by ESRRA overexpression. However, E2 treatment reduced the binding efficiency of ESRRA on *Spp1* promoter. Additionally, we constructed a DNA binding domain (DBD)-deletion ESRRA expression vector (ESRRA- Δ DBD). By using luciferase assays, we observed that ESRRA- Δ DBD lost the repression effect on E2/ESR1-driven *Spp1* promoter activation (**revised Fig. 4g**, data copied below). These evidences demonstrated that ESRRA executes a transcriptional repressor by binding to *Spp1* promoter. We updated the description in the revised text. (**Page 9 line 254-256, page 9 line 262-265, page 23 line 681-683, page 24 line 710-717**) We also updated the y-scale of the ChIP data in **revised Fig.4** and **Fig.6**.

Thanks for raising this concern. To further define ESR1 transcriptional activity on region 1 and 2, we constructed a region 2-deletion *Spp1* promoter (*Spp1*- Δ R2-luc) and performed luciferase assays as shown in **revised Fig. 4i** (data copied below). Deletion of R2 blocked the transcriptional effects of E2/ESR1 and ESRRA on *Spp1* promoter. This result indicated that R2 was more responsive for the modulation of E2/ESR1 with interruption of ESRRA which is consistent to that of ChIP assays. We also updated this part in the text (**page 9 line 260-262**) and Methods (**page 23 line 685-686**).

i

- ESRRR activates SPP1 expression in osteoblasts in opposite to what is here shown in adipocytes. Is this a cell-specific difference? Effect of ESRRR on SPP1 expression in osteoblasts should be used as a control.

Response: Yes, we found that ESRRR might positively regulated SPP1 expression during osteogenesis of BMSCs. We added the data in **revised Supplementary Fig. 4a**. ESRRR and SPP1 have similar expression pattern during osteogenic differentiation of BMSCs. These evidence indicated that ESRRR regulates SPP1 expression as a transcriptional activator or repressor in a cell context-dependent manner. We included the following in the Discussion (**Page 16 line 465-474**), “Previous studies reported that ESRRR regulates SPP1 expression as a transcriptional activator or repressor in a cell context-dependent manner^{54,55,56}. ESRRR positively regulates SPP1 gene in mouse osteoblastic (MC3T3-E1) cell lines or in non-osteoblastic (HeLa) human cell line^{54,55}. On the other hand, ESRRR also exerts repression effect on *SPP1* promoter transactivation in human osteosarcoma cell lines SaOs-2 and U2-OS by cross-talking with another nuclear receptor NR4A2⁵⁶. Most of these studies were performed in cell lines *in vitro*. Here, we revealed that ESRRR acts as a repressor on *Spp1* promoter in adipocytes by interrupting with E2/ESR1 signaling and further confirmed the bona fide binding sites by luciferase assays, as well as by ChIP assays in ESRRR-deficiency primary cells which may confer a truly physiological response.”

- Authors state that SPP1 exert a proangiogenic effect, but what about VEGF which has been shown as an ESRRR target (see Zou et al., J Pathol 2014). Is VEGF expression dysregulated in ESRRR KO animals?

Response: Thanks for raising this important concern. We checked the RNA-seq data and found that *Vegf* was down-regulated in *Esrra*^{AKO} group compared to ctrl *Esrra*^{fl/fl}. We updated this gene in the **revised Fig.3g**. Moreover, we performed qPCR analysis and confirmed that the mRNA level of *Vegf* was indeed reduced in *Esrra*^{AKO} BMAds (**revised Supplementary Fig. 4g**). However, *Vegf* FPKM in *Esrra*^{AKO} group is 10 which is much lower than *Spp1* FPKM (**revised Supplementary Fig. 4f**). It is reasonable that highly up-regulated SPP1 is a key factor for type H vessels formation in the context of this study. We also discuss the following in the revised manuscript:

“BMAds and their precursors LepR⁺ stromal cells in adult bone marrow, are the major sources of NGF, KitL and VEGF contributing to sustain nerves and promote haematopoietic and vascular regeneration after myeloablation^{17,69}. These emerging evidences suggest marrow adipocytes homeostasis are essential, especially in impaired bone marrow environment after irradiation or chemotherapy. Furthermore, the absence of marrow adipocytes induced by global reduction in c-kit receptor function and m-kit ligand function in mice result in exacerbated bone loss, demonstrating that normal marrow adipocytes and hematopoietic lineage cells are required for maintaining bone homeostasis^{70,71,72}. It needs to be determined whether these factors play essential roles in the expanded MAT induced by different pathological conditions.”

- On fig6 the authors show that ESRRA requires PGC1alpha to stimulate Leptin promoter. The effect of ESRRA on the SPP1 promoter should be studied with and without PGC1a, it could indeed be possible that ESRRA functions as a repressor only in the absence of PGC1a. Also what is the effect of ESRRA on the leptin promoter without PGC1a?

Response: As suggested, we performed luciferase assays and found that E2/ESR1-driven *Spp1* promoter activation can be repressed by either ESRRA, PPARGC1A or ESRRA+PPARGC1A (**revised Supplementary Fig. 5c**). It is possible that PPARGC1A could induce ESRRA expression as a coactivator since ESRRA has ERREs on its own promoter (J Biol Chem 2003, PMID: 12522104; Proc Natl Acad Sci. 2004, PMID: 15184675).

- Leptin expression is reduced in the adipocyte-specific ESRRA KO in vivo. Could this be due to altered levels of food intake in KO versus wild type animals (and thus not necessarily due to a direct effect of ESRRA on leptin promoter)?

Response: Actually we measured the food take and found no differences between two groups. We added this data in **revised Supplementary Fig. 2a**. Furthermore, *in vitro* data as shown in **Fig. 6** and **Fig. 8a-8e** demonstrated that ESRRA directly regulated leptin expression in adipocytes which was independent on food intake.

- The authors focus the second part of the ms on the effect of ESRRA on Leptin. In FigS8, they show that ESRRA reduces the expression of AdipoQ and PPARg in gWAT as potently as the expression of leptin. AdipoQ and PPARg are important for WAT differentiation and functions, thus why did the authors focus solely on leptin? A sentence should be added in the text to explain this motivated choice. In this respect, no change in PPARg CEBPa and FABP4 were detected in gWAT on Fig4D (wt

versus ESRRA KO, I assume?), whereas reduced expression is detected in BMSC-gWATCM induced (Fig7C). Can the authors explain?

Response: In Fig.S8 (**revised Supplementary Fig. 7**), we detected gene expression levels in gWAT from *Esrra*^{fl/fl}-DIO and *Esrra*^{AKO}-DIO mice. We observed that mRNA levels of *Adipoq* and *Pparg* were lower in *Esrra*^{AKO}-DIO mice compared to *Esrra*^{fl/fl}-DIO mice. In Fig4D (**revised Fig 4c**), we detected gene expression levels in gWAT from *Esrra*^{fl/fl}-OVX and *Esrra*^{AKO}-OVX mice. We observed no differences of *Pparg*, *Cebpa* and *Fabp4* expression between two OVX groups. However, we observed that leptin expression in either WAT or BMAdS was decreased by ESRRA deletion in both DIO and OVX studies. Furthermore, blood leptin levels were significantly reduced in *Esrra*^{AKO}-DIO and *Esrra*^{AKO}-OVX mice compared to the corresponding control mice, meanwhile there were no significant changes in WAT weight (**revised Fig.1c, 1e, Fig.2b-2d**). It is also well-known that Lepin/LepR signaling is important to dictate BMSCs fate commitment. Hence, these evidences prompted us to focus on the role of leptin. We updated this point in the text, “Compared to DIO or OVX controls, ESRRA deficient mice displayed declined levels of circulating leptin despite no differences in peripheral WAT mass, as well as repressed expression in both WAT-adipocytes and BMAdS (**Fig. 1e, 2d, 3j, 4b-4d and Supplementary Fig. 7**), suggesting that *Lep* may be a target gene of ESRRA.” (**Page 11 line 309-312**)

In **Fig.7c** and **Fig.7g**, we measured mRNA levels of well-established osteogenic markers *Runx2*, *Sp7* and *Bglap*; and adipogenic markers *Pparg*, *Cebpa* and *Fabp4* to determine the fate commitment of wild-type BMSCs. Similar to previously reported methods as shown in **Fig.7a** (Nat. Methods 2010 PMID 20676108; Cell metabolism 2019 PMID 30773468), we cultured wild-type BMSCs in medium containing both adipogenic and osteogenic cues to rigidly test BMSCs lineage commitment with an additional treatment of conditioned medium (CM) collected from either *Esrra*^{AKO}-gWAT/BMAdS or corresponding control CM. The reduced expression levels of *Pparg*, *Cebpa* and *Fabp4* indicated that wild-type BMSCs treated with CM from *Esrra*^{AKO} exhibited a significant decrease in adipogenic potential.

Minor comments :

- Verify the spelling of LEPTIN everywhere (some occurrence of LEPIN or LEPITN)

Response: Thanks for this kind reminder. We checked the manuscript and corrected all the typos.

- Define FFA on line 67

Response: We changed it to “free fatty acid (FFA)” in this sentence.

- FigS1: define gWAT, iWAt and mWAT in Suppl Fig Legend

Response: We defined these in the **revised Supplementary Fig. 1d** legend.

Response to comments from Reviewer #3:

Huang et al. investigated the role of ERRa in adipocyte in DIO and OVX induced bone loss models using ERRa Adipoq-cre mice. Adipocyte ERRa deficiency protected mice from DIO or OVX-induced bone loss. ERRa is shown to oppositely regulate the expression of LEPTIN and SPP1, affecting MAT expansion and bone formation. This study uncovers the mechanisms by which ERRa in adipocytes contributes to bone formation and angiogenesis. In addition, this manuscript sheds light into the mechanisms by which MAT expansion is regulated and the effect of the fat-bone axis on pathological bone diseases.

Response: We appreciate the positive feedback and constructive comments.

Major points:

1. Figure 1. Fig. 1J. The authors should show normal diet controls. Fig. 1P. The visualization of ERRa in bone sections by immunofluorescence is necessary to support the effect of ERRa on MATs. In addition, SPP1 signals in ERRa-deficient bone increased, implicating the indirect effect of increased SPP1 from ERRa-deficient adipocytes on SPP1 expression of other cells. *in situ* hybridization or RNA scope of *Spp1* mRNA can clarify the pattern of *Spp1* in BMSCs. Fig. 1H. BMAs are actively secreting cytokines and adipokines. Metabolic inflammation in adipose tissue in obese mice and human subjects has been observed. Moreover, OPN plays an important role in inflammatory disorders. The authors should analyze if the secretion of inflammatory factors differs between control and ERRa-deficient mice.

Response: We thank the reviewer for this important suggestion. We updated the control normal chow diet (NCD) groups and reorganized the whole **Fig. 1** and presented this part in the text (**page 5 line 130-143**).

There are different types of cells in the bone marrow some of which could be recognized with specific markers in bone sections by immunofluorescence. However, ESRRA as a transcriptional factor is ubiquitously expressed in different cell types. As expected, it's difficult to tell the differences of ESRRA expression in bone sections (data below).

Instead of ESRRRA immunofluorescence in bone section and in situ hybridization of *Spp1* mRNA in BMSCs, we performed the *in vivo* study by using an Adipoq-Cre to specific knockout ESRRRA in most BMAd and executed *ex vivo* experiments similar to previously reported studies. Additionally, we analyzed SPP1 expression during adipogenesis or osteogenesis of BMSCs from *Esrra*^{fl/fl} and *Esrra*^{AKO} mice in **revised Fig. 3a, j** and **Supplementary Fig. 4a**. These results revealed that adipocyte-specific ESRRRA ablation by using Adipoq-Cre enhanced SPP1 expression in adipogenic differentiated BMSCs, whereas no effect in osteogenic differentiated BMSCs. These evidenced prompted us to analyze secreted factors such as SPP1 or leptin from mature differentiated BMAd or WAT on wild-type BMSCs fate determination.

Thanks for this valid point. We agree with the reviewer that metabolic inflammation is often associated with obese. As suggested, we measured two key inflammatory factors TNFa and IL6 in two DIO studies as shown in **revised Fig. 1e** and **Fig. 8g** (data copied below). Indeed, the blood levels of TNFa and IL6 were significant increased in *Esrra*^{fl/fl}/*Esrra*^{AKO} DIO mice as well as wild-type DIO mice compared to NCD groups. In **revised Fig.1e** we did not observe significant differences of TNFa and IL6 between *Esrra*^{fl/fl} and *Esrra*^{AKO} DIO mice. However, blood levels of TNFa and IL6 were significant reduced in C29-treated DIO group as shown in **revised Fig.8g**. We updated the results as follows: “In comparison to NCD vehicle mice, circulating levels of Lepin, TNFa and IL6 were dramatically enhanced in DIO vehicle mice which were significantly reduced by oral gavage of C29, indicating repressed systemic inflammation due to reduced ESRRRA activity (**Fig. 8g**).” (**Page 13 line 372-374**).

2. Figure 3. The authors should provide more thorough analysis of RNA seq. It remains unclear how many genes are differentially regulated by ERRa deficiency. MATs express both BAT and WAT-marker genes. The authors should show the expression of BAT-marker genes to determine any phenotypic changes in MATs by ERRa. Although WAT has few mitochondria, mitochondrial function is important for the differentiation of adipocytes (especially in obese states). ERRa plays a crucial role in regulating mitochondrial function. It would be interesting to know whether ERRa regulates genes related to oxidative phosphorylation or other metabolic pathways in adipocytes.

Response: Thanks for the valuable suggestions. We have deposited RNA-seq datasets in Gene Expression Omnibus (GEO) database (GSE248799, hyperlink: <https://www.ncbi.nlm.nih.gov/geo/query/acc.cgi?acc=GSE248799> token: yzsfkwiitnglnwd) and added the information in “Data availability” section. As suggested, we performed a more thorough analysis of RNA-seq, and differentially expressed genes regulated by ESRRA deficiency have been shown in **revised Fig. 3d**. We also updated RNA-seq analysis related to oxidative phosphorylation, TCA cycle and other metabolic pathways including glycolysis and lipid metabolism in **revised Supplementary Fig. 4b-e**. We presented the following sentences in text (**page 7 line 210-213**): “A total of 4217 genes were quantified, of which 1977 and 1448 were respectively up- and down-regulated in the ESRRA knockdown cells (**Fig. 3d**). As expected, the expression profile of genes associated with energy metabolism, including oxidative phosphorylation, tricarboxylic acid cycle, lipid metabolism and glycolysis were modestly affected (**Supplementary Fig. 4b-e**).”

It has been proposed that MAT has beige-like characteristics. However, whether beige adipocyte precursors reside in the bone marrow and can differentiate into functional cells is unclear, as is the unlikely possibility that the precursor of marrow adipocytes can also give rise to beige adipocytes in mice (Sci. Rep. 9, 17427 (2019), PMID: 31758074; Cell Metab. 2016, PMID: 27238639). We measured the RNA expression levels of the BAT maker genes *Ucp1* and *Prdm16* by qRT-PCR analysis and found they were down-regulated by ESRRA deficiency in BMAds (data below). However, the mRNA expression levels of *Ucp1* and *Prdm16* detected by qPCR were quite low (Ct > 30 cycles).

We then checked the RNA-seq data and found that *Prdm16* *PFKM* < 1 and *Ucp1* *PFKM* < 0.02 which were nearly undetectable. We speculated that this result might be due to rosiglitazone (Rosi) treatment in our *ex vivo* study to mimic the excessive BMAds accumulation in bone. Another possibility is that beige-like adipocytes resided in the bone marrow are a small sub-population of MAT, and they might be

induced upon some stress challenges such as cold exposure. We are sorry that this current research is limited on studying on beige-like adipocytes within MAT. We added the following in the Discussion: “Moreover, ESRRA also mediates adaptive thermogenesis of BAT^{67,68}. Whether ESRRA deficiency affects bone metabolism through regulating BAT or the beige-like characteristics of MAT especially in cold stress condition need to be further investigated.” (Page 17 line 487-489)

3. Figure 4. Fig. 4F: SPP1 is diminished by estrogen deficiency in control mice. The authors should show the time-course expression of SPP1 and Leptin in both control and ERRa-deficient WAT in the same gel. Determining the level of ERRa will be useful in elucidating the intercorrelation between ERRa and Oestrogen during osteoporosis induced bone loss. Fig. 4L: The link between OPN and ERRa has been established. It has been shown that ERRa binds to the promoter of human OPN and increases the expression of OPN (PMID: 286382130). The authors should discuss the tissue-specific regulation of OPN by ERRa in the discussion sections.

Response: Thanks for the valuable suggestions. We performed the western blotting analysis in revised Fig.4d. The data showed the time-course expression of SPP1, Leptin and ESRRA in both control and ESRRA-deficient gWAT in the same gel (data copied below).

Thanks for this kind reminder. Actually, previous studies have reported that ESRRA acts as either an activator or a repressor in a cell context-dependent manner. As suggested, we added this part in Discussion section as follows (Page 16 line 465-474):

“Previous studies reported that ESRRA regulates SPP1 expression as a transcriptional activator or repressor in a cell context-dependent manner^{54,55,56}. ESRRA positively regulates SPP1 gene in mouse osteoblastic (MC3T3-E1) cell lines or in non-osteoblastic (HeLa) human cell line^{54,55}. On the other hand, ESRRA also exerts repression effect on SPP1 promoter transactivation in human osteosarcoma cell lines SaOs-2 and U2-OS by cross-talking with another nuclear receptor NR4A2⁵⁶. Most of

these studies were performed in cell lines *in vitro*. Here, we revealed that ESRRRA acts as a repressor on *Spp1* promoter in adipocytes by interrupting with E2/ESR1 signaling and further confirmed the bona fide binding sites by luciferase assays, as well as by ChIP assays in ESRRRA-deficiency primary cells which may confer a truly physiological response.”

4. Figure 7. KO BMAds CM enhanced osteogenesis or decreased adipogenesis. However, in Figure 3C, there is no difference in osteogenesis or adipogenesis between WT and KO BMSCs, suggesting additional defects in the response to SPP1 or LEPTIN in ERRa KO BMSCs. The authors should discuss about this discrepancy.

Response: We apologized for this confusion. We corrected “wild-type BMSCs” to “BMSCs” in the **revised Fig 3b**. In this experiment, BMSCs were isolated from *Esrra^{fl/fl}* and *Esrra^{AKO}* mice respectively and then differentiated for 14 days. We observed a reduction in adipogenesis of BMSCs from *Esrra^{AKO}* mice in adipogenic induction medium with an additional treatment of Rosi which mimics adipocyte expansion under pathological conditions (**Fig 3c**). These results indicate that loss of ESRRRA during adipocyte expansion under pathological conditions might exert a secondary or paracrine effect on BMSCs differentiation. These evidences prompted us to evaluate the effects of secreted factors from mature adipocytes on wild-type BMSCs fate determination. *Esrra^{fl/fl}*-CM and *Esrra^{AKO}*-CM were collected from either minced gWAT or fully differentiated BMAds with the treatment of Rosi. We updated “adipogenic induction” to “adipogenic induction (+ Rosi)” in **revised Fig. 7a**. To rigidly test BMSCs lineage commitment, BMSCs from wild-type adult mice were cultured in medium containing both adipogenic and osteogenic cues as previously reported by others (Nat. Methods 2010 PMID 20676108; Cell metabolism 2019 PMID 30773468). We observed that wild-type BMSCs treated with *Esrra^{AKO}*-derived CM exhibited an increase in osteogenic capability, meanwhile displayed a significant decrease in adipogenic potential in comparison with that of *Esrra^{fl/fl}* mice.

Minor points

1. The authors should show the expression of ERRa in Figures 4J and K.

Response: As suggested, *Esrra* overexpression was confirmed in **revised supplementary Fig. 5d-f**.

2. Figure S6 supports Figure 4L. The authors should move Figure S6 into main figure

4.

Response: As suggested, this figure is now included in the **updated Fig. 4m**. Additionally, we also added **Fig.6i** from supplementary data in the main figure.

REVIEWER COMMENTS

Reviewer #1 (Remarks to the Author):

Huang et al provided a comprehensive response to my review and addressed many of my concerns. However, there remain a few issues impacting interpretation of the authors results.

Issue 1

The authors insist that they are not performing their studies in growing mice. This is not the case. Longitudinal and radial bone growth in B6 mice slows with age, as does body weight gain, but continues until the mice are about 6 months old. Also, longitudinal bone growth accelerates in mice following ovariectomy. The Glatt paper referenced by the authors is not relevant here because: 1) linear bone growth, not cancellous BV/TV defines skeletal maturity and 2) the profound decrease in BV/TV in the mice in the Glatt study mice was subsequently shown to be due to cold stress induced by room temperature housing and is prevented by thermoneutral housing. The fact that the mice in the authors study were growing is relevant because the region of interest for the microCT measurements began 0.36 mm distal to the growth plate. This means that some (perhaps all) of the bone that they measured was formed by endochondral ossification or was primary spongiosa at the start of the study. It also means that the proportion of cancellous bone in the region of interest that was present at the start of the study (mature) will be less for ovariectomized mice than for ovary intact mice. Additionally, the mechanisms of action of sex steroids differ between mature and immature bone (see issue 2).

Issue 2

The reviewer respectfully but strongly disagrees with the authors view that bone loss following ovariectomy is due to reduced bone formation. The acute skeletal response to ovariectomy in skeletally mature primates (including humans) and rodents (including mice) is bone loss due to increased turnover (resorption is increased more than formation). In contrast, growing rodents develop cancellous osteopenia in their long bones due to increased resorption of calcified cartilage. It is also important for the authors to appreciate species differences. Although the effects of endogenously produced estrogens on the mouse skeleton appear to be like other mammals the effects of endogenous estrogens are unique to mice. This was beautifully described by M.R Urist (discover of bone morphogenic protein), D.J. Simmons and others in a series of papers published many decades ago. Urist and colleagues evaluated the actions of 11 natural and synthetic estrogens in 8 mammalian species. They found that administration of estrogen to mice and only mice resulted in an endosteal and cancellous osteosclerotic response which subsequently has been shown to be secondary to bone marrow failure. This skeletal response is not relevant humans and is probably not physiologically relevant to mice, since it requires non physiological hormone administration of very high levels of estrogen.

Issue 3

The reviewer is a little concerned that the authors disregard evidence that that does not support their viewpoint based on flimsy arguments. For example, the authors argue dose response leptin studies in ob/ob mice should in their view not be considered because ob/ob mice have many metabolic abnormalities. This is true, but the peripheral and central defects are shown to corrected by leptin replacement and peripheral but not central leptin signaling deficiency can be created in wild type mice by adoptive transfer of leptin receptor negative bone marrow cells. From the reviewer's perspective, mice fed a diet where 60% of their energy is obtained from fat is hardly physiological. Nevertheless, the authors results when published warrant respectful evaluation.

Reviewer #2 (Remarks to the Author):

The authors have satisfactorily answered to the comments I previously raised. Attention should however be taken to the reference numbering. I noted that for instance ref 60, 61 and 62 in the text on page 16 should actually be ref 57, 58 and 59 in ref list. I would suggest that all references could be checked in the text and ref list to insure appropriate correspondence.

Reviewer #3 (Remarks to the Author):

The authors have adequately addressed my comments in the revised manuscript.

Minor points

The authors should address the discrepancy in baseline ERRa levels between Figure 3a and Suppl Fig. 4a.

Response to comments

Reviewer #1 (Remarks to the Author):

Huang et al provided a comprehensive response to my review and addressed many of my concerns. However, there remain a few issues impacting interpretation of the authors results.

Response: We appreciate the Reviewer for constructive comments, helping us improve the revised manuscript.

Issue 1

The authors insist that they are not performing their studies in growing mice. This is not the case. Longitudinal and radial bone growth in B6 mice slows with age, as does body weight gain, but continues until the mice are about 6 months old. Also, longitudinal bone growth accelerates in mice following ovariectomy. The Glatt paper referenced by the authors is not relevant here because: 1) linear bone growth, not cancellous BV/TV defines skeletal maturity and 2) the profound decrease in BV/TV in the mice in the Glatt study mice was subsequently shown to be due to cold stress induced by room temperature housing and is prevented by thermoneutral housing.

The fact that the mice in the authors study were growing is relevant because the region of interest for the microCT measurements began 0.36 mm distal to the growth plate. This means that some (perhaps all) of the bone that they measured was formed by endochondral ossification or was primary spongiosa at the start of the study. It also means that the proportion of cancellous bone in the region of interest that was present at the start of the study (mature) will be less for ovariectomized mice than for ovary intact mice. Additionally, the mechanisms of action of sex steroids differ between mature and immature bone (see issue 2).

Response: We are sorry for our miscommunication, we meant that we did not perform our studies in puberty mice. Instead, our study were conducted in mice at the age of 18-25 weeks, which are usually described as “adult mice” in majority publications. We agree with the reviewer that longitudinal and radial bone keep growing in normal B6 mice until 6-month-old. As the reviewer mentioned, indeed even modest differences in housing temperature conditions could influence cancellous bone turnover balance in distal femur metaphysis in growing female mice according to recent published studies (PMID: 37007217; 27189604). The paper (Osteoporos Int 2016 PMID: 27189604) reported that the BV fraction in metaphysis in growing (4-week-old) female housed at 32 °C for up to 18 weeks was significantly increased compared to the mice housed at

22 °C at the same age, and similar to the BV fraction of mice housed at room temperature (22 °C) at 8 weeks of age. However, the authors of this paper did not compare the BV fraction of mice at different ages housed at 32 °C. Whether “the profound decrease in BV/TV in the mice in the Glatt study mice upon ageing was due to cold stress induced by room temperature housing and was prevented by thermoneutral housing” needs to be further evaluated by conducting an experiment to compare mice of different ages housed at 32 °C (or mice of different ages housed at 32 °C for a same period). Interestingly, we found a publication reporting that thermoneutral housing at 30°C also accelerates the onset of metabolic inflammation in adipose tissue and the vasculature (Cell metabolism 2016 PMID 26549485) which might also affect bone metabolism in our current genotypes. We are sorry we cannot prove all these aspects in our current study since we performed the experiments only at room temperature and all the mice of different groups in each figure were at the same age.

For adult mice, we analyzed microCT beginning at 0.36 mm distal to the growth plate, according to published guidelines (PMID: 20533309; PMID: 15005847) and the Book “Micro-computed Tomography (micro-CT) in Medicine and Engineering” (<https://link.springer.com/book/10.1007/978-3-030-16641-0>), on Page 62-63 in the book: “For trabecular bone the offset is around 0.2–0.4 mm for mice and about 0.4–1.5 mm for rats. The analyst should select the offset distance as the distance needed to move clear of most of the fine-textured primary spongiosal structures near the growth plate into a metaphyseal region dominated by more mature (and remodeled) secondary trabecular structures.” For young mice, the offset distance might need to be adjusted at around 0.2 mm. In our current study, we chose the distance starting at 0.36 mm under the growth plate for all the 18-25 week-old mice (both control and OVX/DIO mice), that is a metaphyseal region dominated by mature secondary spongiosa, and to make all the data comparable. We further confirmed that many other publications also performed microCT analysis in similar proportion of cancellous bone in the region of interest in both OVX and sham adult mice (For example: at 0.215mm in Bone research 2021 PMID: 34719673; 0.28mm in JCI 2013 PMID: 23945236 and Nat med 2018 PMID: 29785024; 0.3mm in JBMR 2017 PMID: 28745432 and Nature medicine 2014 PMID: 25108526; 0.4mm in Nat med 2018 PMID: 29662200). Hence, we believe that we measured an appropriate ROI of secondary spongiosa but not primary spongiosa in these adult mice at the age of 18-25 weeks.

We thank the Reviewer for raising this important question. In the revised manuscript, we added the following in the Discussion as well as updated the related references: “On the other hand, emerging evidences showed that thermoneutral housing has a broad influence in regulating bone turnover, metabolic and behavioral responses in mice which might better model human physiology and disease^{71,72,73}. Whether adipocyte-ESRRA deficiency mice can be modulated by thermoneutral housing should be paid attention to in future investigations.” (Page 17 line 493-496)

Issue 2

The reviewer respectfully but strongly disagrees with the authors view that bone loss following ovariectomy is due to reduced bone formation. The acute skeletal response to ovariectomy in skeletally mature primates (including humans) and rodents (including mice) is bone loss due to increased turnover (resorption is increased more than formation). In contrast, growing rodents develop cancellous osteopenia in their long bones due to increased resorption of calcified cartilage.

It is also important for the authors to appreciate species differences. Although the effects of endogenously produced estrogens on the mouse skeleton appear to be like other mammals the effects of endogenous estrogens are unique to mice. This was beautifully described by M.R Urist (discover of bone morphogenic protein), D.J. Simmons and others in a series of papers published many decades ago. Urist and colleagues evaluated the actions of 11 natural and synthetic estrogens in 8 mammalian species. They found that administration of estrogen to mice and only mice resulted in an endosteal and cancellous osteosclerotic response which subsequently has been shown to be secondary to bone marrow failure. This skeletal response is not relevant humans and is probably not physiologically relevant to mice, since it requires non physiological hormone administration of very high levels of estrogen.

Response: We thank the Reviewer for providing these valuable comments. We definitely agree with the reviewer that ovariectomy will induce dramatic bone resorption. Actually, we did observe a significantly augment of osteoclast-associated resorption in either DIO (Fig.1j and l) or OVX mice (Fig.2g and 2j) compared to control mice. However, there were no differences in bone resorption between the two genotypes. This phenomenon attracted our attention to bone formation since we observed that adipocyte ESRRA ablation in both DIO and OVX mice resulted in elevated bone formation rate and bone mass compared to corresponding DIO and OVX controls which suggests a protection effect of ESRRA ablation in mice against bone loss in these pathological conditions. We agree with the Reviewer that species differences are very important for either estrogen studies or establishing appropriate OVX models for many pharmacological conditions, especially in bone diseases.

Collectively, we updated the following in the revised Discussion:

“Moreover, although we observed adipocyte ESRRA inhibition leads to a significantly augmentation of osteoprogenitors associated with improved type H vessel formation in adipocyte-rich bone marrow, this study is limited without using elderly OVX mice or aged female mice at over 18 months of age to mirror bone turnover in elderly postmenopausal women and preclinical testing of potential therapies for osteoporosis.

Hence, considering ageing and species differences, whether adipocyte ESRRA deficiency is protective against bone loss in aged mice or non-human primates is to be further addressed in estrogen-deficiency condition associated with ageing, which is often accompanied by not only accelerated bone resorption but also decreased bone formation due to declined number in BMSCs pool and decreased capability in osteogenic differentiation. ” (Page 17 line 510-518)

Issue 3

The reviewer is a little concerned that the authors disregard evidence that that does not support their viewpoint based on flimsy arguments. For example, the authors argue dose response leptin studies in *ob/ob* mice should in their view not be considered because *ob/ob* mice have many metabolic abnormalities. This is true, but the peripheral and central defects are shown to corrected by leptin replacement and peripheral but not central leptin signaling deficiency can be created in wild type mice by adoptive transfer of leptin receptor negative bone marrow cells. From the reviewer’s perspective, mice fed a diet where 60% of their energy is obtained from fat is hardly physiological. Nevertheless, the authors results when published warrant respectful evaluation.

Response: Thanks very much for the reviewer’s kind reminder once again. We realized that we made a wrong description of high-fat diet that contains not “60% fat in kcal” but 35% fat (Research diet #D12492) which is often employed for diet-induced obese mice studies. We apologize for this careless mistake and have corrected it in revised manuscript (Page 14 line 421-422; Page 19 line 551).

The current study was conducted under the scenario of hyperleptinemia in DIO mice. Although we are not on the same page with the Reviewer regarding the leptin-deficient *ob/ob* mice or adoptive transfer of leptin receptor negative bone marrow cells in wild type mice, we reconsidered the Reviewer’s concerns and added the following discussion and updated related references provided by the Reviewer in the revised manuscript. “Exogenous leptin replacement by either a peripheral or central pathway in leptin-deficient *ob/ob* mice has been reported to play a positive influence in bone formation suggesting that leptin is essential for normal bone homeostasis and might play multifaceted roles not only in bone turnover and metabolic regulation but also under the different scenarios such as leptin-deficiency and hyperleptinemia^{48,49}.” (Page 15 line 423-426)

We highly appreciate the Reviewer’s comments. Since leptin exerts complicated effects through central, peripheral and even local signaling, our efforts might contribute to advance current knowledge.

Reviewer #2 (Remarks to the Author):

The authors have satisfactorily answered to the comments I previously raised. Attention should however be taken to the reference numbering. I noted that for instance ref 60, 61 and 62 in the text on page 16 should actually be ref 57, 58 and 59 in ref list. I would suggest that all references could be checked in the text and ref list to insure appropriate correspondence.

Response: We apologize for this mistake. We have corrected references in revised manuscript (ref 56-70) in the revised manuscript. Thank you very much for your careful checking and kind reminder.

Reviewer #3 (Remarks to the Author):

The authors have adequately addressed my comments in the revised manuscript.

Response: We thank the Reviewer for the valuable comments on our manuscript.

Minor points

The authors should address the discrepancy in baseline ERRa levels between Figure 3a and Suppl Fig. 4a.

Response: These two results represent ESRRa levels of BMSCs in different induction treatments, that is Fig. 3a showing ESRRa levels during adipogenic differentiation and Suppl. Fig. 4a displaying ESRRa levels during osteogenic differentiation. However, ESRRa levels in undifferentiated (0 d) BMSCs in either Figure 3a or Suppl Fig. 4a were generally very low.

REVIEWERS' COMMENTS

Reviewer #1 (Remarks to the Author):

The authors have addressed the reviewers concerns. The reviewer recognizes and thanks the authors for the considerable effort they made in preparing and revising this work.

Reviewer #3 (Remarks to the Author):

The authors have adequately addressed my comments in the revised manuscript.